# High Homogeneous Freezing Onsets of Sulfuric Acid Aerosol at Cirrus Temperatures

Julia Schneider[1], Kristina Höhler[1], Robert Wagner[1], Harald Saathoff[1], Martin Schnaiter[1], Tobias Schorr[1], Isabelle Steinke[2], Stefan Benz[3], Manuel Baumgartner[4,a], Christian Rolf[5], Martina Krämer[5,4], Thomas Leisner[1], and Ottmar Möhler[1]

[1]Institute of Meteorology and Climate Research, Karlsruhe Institute of Technology, Karlsruhe, Germany
[2]Atmospheric Sciences and Global Change Division, Pacific Northwest National Laboratory, Richland, USA
[3]Landesanstalt für Umwelt Baden-Württemberg, Karlsruhe, Germany
[4]Institute for Atmospheric Physics, Johannes Gutenberg University, Mainz, Germany
[5]Institute for Energy and Climate Research, IEK-7: Stratosphere, Research Center Jülich, Jülich, Germany
[a]now at: Deutscher Wetterdienst, Offenbach, Germany

**Correspondence:** Dr. Ottmar Möhler (ottmar.moehler@kit.edu)

**Abstract.** Homogeneous freezing of aqueous solution aerosol particles is an important process for cloud ice formation in the upper troposphere. There the air temperature is low, the ice supersaturation can be high, and the concentration of ice-nucleating particles is too low to initiate and dominate cirrus cloud formation by heterogeneous ice nucleation processes. The most common description to quantify homogeneous freezing processes is based on the water activity criterion (WAC) as proposed by Koop et al. (2000). The WAC describes the homogeneous nucleation rate coefficients only as a function of the water activity, which makes this approach well applicable in numerical models. In this study, we investigate the homogeneous freezing behaviour of aqueous sulfuric acid aerosol particles by means of a comprehensive collection of laboratory-based homogeneous freezing experiments conducted at the AIDA (Aerosol Interaction and Dynamics in the Atmosphere) cloud simulation chamber, which were conducted as part of 17 measurement campaigns since 2007. The most recent experiments were conducted during October 2020 with special emphasis on temperatures below $200\,\mathrm{K}$. Aqueous sulfuric acid aerosol particles of high purity were generated by particle nucleation in a gas flow composed of clean synthetic air and sulfuric acid vapor, which was added to the AIDA chamber. The resulting chamber aerosol had number concentrations from $30\,\mathrm{cm^{-3}}$ up to several thousand $\mathrm{cm^{-3}}$ with particle diameters ranging from about $30\,\mathrm{nm}$ to $1.1\,\mathrm{\mu m}$. Homogeneous freezing of the aerosol particles was measured at simulated cirrus formation conditions in a wide range of temperatures between $185\,\mathrm{K}$ and $230\,\mathrm{K}$ with a steady increase of relative humidity during each experiment. At temperatures between about $205\,\mathrm{K}$ and about $230\,\mathrm{K}$, the AIDA results agree well with the WAC-based predictions of homogeneous freezing onsets. At lower temperatures, however, the AIDA results show an increasing deviation from the WAC-based predictions towards higher freezing onsets. For temperatures between $185\,\mathrm{K}$ and $205\,\mathrm{K}$, the WAC-based ice saturation ratios for homogeneous freezing onsets increase from about 1.6 to 1.7, whereas the AIDA measurements show an increase from about 1.7 to 2.0 in the same temperature range. Based on the experimental results of our direct measurements, we suggest a new fit line to formulate the onset conditions of homogeneous freezing of sulfuric acid aerosol particles as an isoline for nucleation rate coefficients between $5\cdot 10^{8}\ \mathrm{cm^{-3}s^{-1}}$ and $10^{13}\ \mathrm{cm^{-3}s^{-1}}$. The potential significant impacts of the higher homogeneous freezing thresholds, as directly observed in the AIDA

experiments under simulated cirrus formation conditions, on the model prediction of cirrus cloud occurrence and related cloud radiative effects are discussed.

*Copyright statement.* TEXT

## 1 Introduction

For predicting the weather and climate on our planet, it is of particular importance to understand and describe the cloud processes in the atmosphere. The evolution of mixed-phase and cirrus clouds is strongly influenced by the presence of cloud ice (Lynch, 2002). Cirrus clouds generally form in the cold upper troposphere and occur in particular in tropical regions but also

in mid and high latitudes. By being able to scatter and absorb light, cirrus clouds are an important factor in the global radiation budget. Solid aerosol particles often act as promoter of cloud ice formation in the atmosphere via heterogeneous nucleation at low ice supersaturations. In regions with low concentrations of ice-nucleating particles, the homogeneous nucleation of aqueous solution aerosol particles becomes more relevant for cirrus clouds at conditions of lower temperatures and higher ice supersaturations. (Cziczo and Froyd, 2014; Cziczo et al., 2013; Lynch, 2002).

Koop et al. (2000) suggested the water activity criterion (WAC) to predict isolines for the homogeneous freezing rate coefficients of aqueous solutions. This approach (hereafter called Koop2000) was deduced from measurements of the homogeneous freezing temperature observed for different aqueous solutions in dependence on their composition. According to their observations, the ice nucleation behavior only depends on the water activity of the solute, which in thermodynamic equilibrium equals the relative humidity of the surrounding environment (Koop, 2015). Thus, the parameterization of the homogeneous nucleation

rate coefficients and related freezing onsets does not depend on the type of the solute, which makes this approach very suitable for atmospheric models.

As aqueous sulfuric acid solution ($H_2SO_4$/$H_2O$) aerosol particles are frequently present in the upper troposphere and lower stratosphere, these particles are of particular importance for freezing processes at these altitudes. Therefore, the experiments in this study focus on this aerosol type. Homogeneous freezing onsets of sulfuric acid available from literature are compiled

in Table 1 and plotted in Fig. 1 in the water activity-temperature-space. In cases where water activities or relative humidities were not given by the authors, we used the Extended AIM Aerosol Thermodynamics Model (E-AIM), Inorganic Model I on http://www.aim.env.uea.ac.uk/aim/aim.php (Clegg et al., 1992; Carslaw et al., 1995; Massucci et al., 1999; Wexler and Clegg, 2002; Clegg and Brimblecombe, 2005) to transfer given $H_2SO_4$ concentrations into water activities, which we assume to be equal to relative humidities. It needs to be considered that for the low temperature range, the model predictions are based on

extrapolations of physiochemical properties towards low temperatures, which remain uncertain (Koop, 2004). For calculating the melting point curve (solid line in Fig. 1), which is the saturation pressure over liquid water in equilibrium with the ice phase, the parameterizations for the saturation pressure over liquid water and ice suggested by Murphy and Koop (2005) have been used. Also shown are homogeneous freezing onset lines calculated according to Koop2000 for nucleation rate coefficients

of $J_V = 10^{13}$ cm$^{-3}$s$^{-1}$ ($\Delta a_w = 0.32$, dashed line) and $J_V = 5 \cdot 10^8$ cm$^{-3}$s$^{-1}$ ($\Delta a_w = 0.3$, dotted line) (Möhler et al., 2003) and corrected according to the revised version of the homogeneous freezing lines by Koop and Zobrist (2009).

The earliest freezing experiments with aqueous sulfuric acid solutions at variable solute concentrations and temperatures were performed with macroscopic samples (Ohtake, 1993; Beyer et al., 1994; Song, 1994). The observed onsets deviate significantly from Koop2000 with low water activities close to the melting point curve. More recent flow tube experiments, where Fourier transform infrared (FTIR) spectroscopy was used to derive the concentration of freezing sulfuric acid aerosol (Bertram et al., 1996; Clapp et al., 1997), reported lower water activities especially for the freezing onsets at low temperatures ($T <$ 220 K). However, the majority of the compiled freezing onsets in Fig. 1 agree with the Koop2000 predictions considering their measurement uncertainties. Homogeneous freezing experiments performed in the AIDA (Aerosol Interaction and Dynamics in the Atmosphere) cloud simulation chamber (Möhler et al., 2003; Mangold et al., 2005) show a tendency of freezing onsets towards higher water activities compared to Koop2000 at low temperatures ($T <$ 200 K). For T < 200 K, Koop2000 predicts homogeneous freezing thresholds ranging between ice saturation ratios of 1.6 and 1.7. However, in an aircraft campaign in 2004, enhanced ice saturation ratios up to 2.3 at about 187 K were observed in the upper troposphere (Jensen et al., 2005), clearly exceeding the Koop2000 freezing thresholds. Also in a later aircraft campaign in 2006, more than half of the relative humidity measurements showed values exceeding Koop2000 at upper tropospheric temperatures (Lawson et al., 2008). Other atmospheric observations at T < 200 K reported atmospheric relative humidities predominantly below Koop2000, but in a few cases the threshold was also exceeded (Krämer et al., 2009; Krämer et al., 2020). These atmospheric observations seemingly contradict Koop2000 if the assumption holds that ice saturation ratios are unlikely to exceed the homogeneous freezing thresholds in the atmosphere. To explain this discrepancy, the formation of metastable ice phases with higher saturation pressure than stable forms of ice was suggested by various authors (Malkin et al., 2012; Russo et al., 2014). Also, the presence of organic films on the aerosol particles has been discussed, which would slow down the water vapor uptake and thus the aerosol equilibration to the surrounding conditions (Jensen et al., 2005). It was also suggested that parameterizations for the saturation pressure over supercooled liquid water at low temperatures are at least 20% too low (Jensen et al., 2005). Other authors argued that ice nucleation in solutions is accompanied and influenced by the expulsion of ions from the ice lattice (Bogdan and Molina, 2010; Shafique et al., 2016), which form a highly concentrated residual solution. It was suggested that the formation of such mixed-phase cloud particles could produce higher ice supersaturations than expected according to Koop2000 (Bogdan and Molina, 2010).

There are only few in situ measurements of cirrus and their microphysical properties in the field, e.g. in aircraft campaigns (Krämer et al., 2020). These measurements are often limited, as the collection of cloud samples in the atmosphere usually takes place after the onset of ice formation, so that the specific conditions at freezing onset are rarely matched in aircraft studies. However, an improved understanding of atmospheric homogeneous freezing processes can be obtained from laboratory experiments under well controlled conditions. Prompted by the observed tendency of homogeneous freezing onsets towards higher ice saturation ratios than predicted by Koop2000 in previous AIDA studies at low temperatures (Möhler et al., 2003; Mangold et al., 2005), an extended set of homogeneous freezing AIDA experiments focused on the freezing of H$_2$SO$_4$/H$_2$O aerosol particles conducted since 2007 was re-analysed in this study. To quantify the observed trend and the potential deviation

from Koop2000 especially at low temperatures, the re-analysed experiments have been complemented by a more recent series
of AIDA measurements focusing on temperatures below about 200 K. With this extensive data set of $H_2SO_4/H_2O$ freezing
onsets we can provide valuable addition to existing data sets and contribute to a better understanding of homogeneous freezing
processes in the atmosphere.

## 2 Methods

### 2.1 Experimental setup

All experiments have been performed according to experimental procedures as described by Möhler et al. (2003), Benz et al.
(2005) and Wagner et al. (2008). Therefore, we only provide a brief summary of the experimental setup and procedures here.

The AIDA facility basically consists of a cloud chamber with $84\,m^3$ volume, which is situated in a thermal housing and
can be cooled to temperatures as low as 183 K. Before starting an experiment, the AIDA cloud chamber is carefully cleaned
by evacuating it to a pressure below about 0.1 hPa followed by flushing with dry and particle-free synthetic air several times
between 10 hPa and 1 hPa. Then, pure water vapor from a heated liquid water reservoir is added to renew the ice layer on the
inner chamber walls and the AIDA is filled with dry synthetic air to ambient pressure. In the cleaned and re-filled chamber, the
background aerosol particle concentration is usually lower than $0.1\,cm^{-3}$.

Fresh sulfuric acid aerosol particles were generated by passing synthetic air over a small amount of concentrated sulfuric
acid located in a heated glass tube. Passive cooling of the mixture in the inlet tube was sufficient to induce the homogeneous
nucleation of pure $H_2SO_4/H_2O$ particles. The number concentration and size of the particles was controlled by the setting of
the tube temperature and the air flow (Wagner et al., 2008). For experiments since 2017, an additional dilution flow of synthetic
air was added downstream of the heated glass tube in order to better control the number concentration of generated particles.
For example, a reservoir temperature of about 383 K, an air flow of about 0.75 standard $l\,min^{-1}$ and a dilution flow of about
14 standard $l\,min^{-1}$ resulted in a median diameter of about 350 nm for the aerosol particles added to the cloud chamber.

A homogeneous distribution of aerosol particles throughout the cloud chamber is achieved by a mixing fan located one
meter above the bottom of the chamber. A few minutes after the end of injection, the aerosol particle number concentration
and size distribution were measured with a Condensation Particle Counter (CPC 3010, TSI GmbH), an Aerodynamic Particle
Sizer (APS 3321, TSI Inc.), and a Scanning Mobility Particle Sizer (SMPS 3934, TSI Inc.), see Sect. 2.2 for details. A cloud
expansion experiment is then started by opening the valves to a vacuum pump operated at controlled pump speed. Immediately
after the pump starts, the pressure and thus the temperature inside the chamber start to drop. At the beginning of the experiment,
the gas temperature drop is close to adiabatic. However, the increasing temperature difference between the well-mixed chamber
interior and the chamber walls, which stay at almost constant temperature, causes an increasing heat flux from the walls to the
chamber volume and thereby causes a deviation from the adiabatic temperature profile towards higher gas temperatures. The
gas temperature was measured by five vertically and five horizontally aligned temperature sensors. These sensors are calibrated
to an accuracy of 0.1 K by direct comparison to a certified platinum resistance temperature sensor (Lake Shore Cryotronics Inc.,
sensor type PT-103-14L with calibration certificate). The sensor calibration runs are conducted in a chiller bath at temperatures

between 293 K and 197 K. We account for an additional uncertainty due to the temperature variability throughout the chamber. The analysis of measurement uncertainties is presented in more detail in Sect. 2.4.

A range of instruments was available for the measurement of gas phase and total water (Wagner et al., 2009). The water vapor pressure was measured inside AIDA by Tunable Diode Laser (TDL) absorption spectroscopy with a time resolution of 1 s using a laser wavelength in the near-infrared of 1370 nm (Fahey et al., 2014). The measurement is done by direct long path absorption in a plane 4.2 m above the bottom of the AIDA chamber. The accuracy of the water vapor measurement is better than $\pm 5\%$ (Fahey et al., 2014). The total water concentration (which is the sum of water vapor and the amount of condensed and frozen water contained in aerosol and/or cloud particles) was measured with a dew point mirror (373LX, MBW) by sampling air from the AIDA chamber via a heated stainless steel tube (accuracy $\pm 3\%$). By subtracting the gas phase water concentration from the total water concentration, the mass concentration of the condensed and frozen water can be calculated. The water vapor measurement by the AIDA TDL system has been compared with a range of other hygrometers during the AquaVIT-1 (intercomparison of atmospheric water vapor measurement techniques) campaign in 2007 and showed good agreement with established methods (Fahey et al., 2014). The ice saturation ratio, $S_{ice}$, in the AIDA chamber is calculated based on the gas temperature, the TDL water vapor measurements and the saturation vapor pressure over ice given by Murphy and Koop (2005).

For some experiments, in situ Fourier Transform Infrared spectroscopy (FTIR) was used to deduce the composition and mass concentration of the $H_2SO_4/H_2O$ aerosol particles before and during expansion cooling (Wagner et al., 2008). Laser light scattering and depolarization measurements with the SIMONE instrument (Schnaiter et al., 2012) allowed for a precise determination of the freezing onset of the $H_2SO_4/H_2O$ aerosol particles during expansion cooling, as described in Sect. 2.3. In the SIMONE set-up, a polarized laser beam with an emission wavelength of 488 nm is directed horizontally across the AIDA chamber. Photomultipliers detect the light scattered by the aerosol and/or cloud particles in the forward (2°) and backward (178°) direction. At 178°, the scattering intensities are measured polarization-resolved, enabling the determination of the back-scattering linear depolarization ratio, which is zero for the spherical $H_2SO_4/H_2O$ solution droplets but increases as soon as aspherical ice crystals are formed after the freezing onset.

The number size distribution of aerosol and cloud particles was continuously measured by two optical particle counters (OPC; model: welas, Palas GmbH), which were sampling via a vertical line from the bottom of the AIDA vessel. The size classification and measurement ranges of the two OPCs (named welas1 and welas2) depend on the particle shape and refractive index. For example, for spherical particles with a refractive index of 1.33, the size ranges of the two sensors are 0.7 to 46 μm (welas1) and 5 to 240 μm (welas2). In Sect. 2.3, it is described how the OPC records are analyzed to provide a second measure for the freezing onset of the $H_2SO_4/H_2O$ aerosol particles in addition to the SIMONE depolarization data.

## 2.2 Characterization of the $H_2SO_4/H_2O$ aerosol particles

The $H_2SO_4/H_2O$ aerosol particles were characterized after their injection into the AIDA chamber. A CPC continuously measured the number concentration of aerosol particles, $c_{aerosol}$ with diameters in the size from 10 nm up to 3 μm inside the chamber. The size distribution of the particles was measured two times after the injection by APS and SMPS. The SMPS detects aerosol particles in a size range between 0.014 and 0.82 μm (mobility diameter), whereas the APS only measures larger

particles with diameters between 0.523 and 19.81 μm (aerodynamic diameter). As aqueous aerosol particles are spherical, the dynamic shape factor to convert the mobility diameter to a geometric diameter was set to unity. To transform the aerodynamic diameter to a geometric diameter, we applied the same shape factor together with the particle density, which was calculated with Model I of the E-AIM model (Clegg et al., 1992; Carslaw et al., 1995; Massucci et al., 1999; Wexler and Clegg, 2002; Clegg and Brimblecombe, 2005) for the specific equilibrium $H_2SO_4/H_2O$ composition. The combination of APS und SMPS data provides particle size distributions over the whole relevant size range. For most of the experiments, the measured size distribution could be well described and fitted by an unimodal lognormal distribution with the parameters $N$ (total number concentrations), $D_m$ (median diameter) and $\sigma$ (geometric standard deviation). Figure 2 shows the lognormal fits of the size distributions for all AIDA experiments for which APS/SMPS size distribution measurements were available. The median diameter ranges from about 30 nm to 1.1 μm. The measured size distribution of experiment number 14 of the ICE01 campaign (grey dots) shows the typically good agreement between the measurement and the lognormal fit. For several experiments with a start temperature below about 210 K, we observed continuously increasing aerosol particle concentrations in the CPC measurements after the $H_2SO_4$ aerosol injection had been stopped. The injected $H_2SO_4$ vapor further nucleated small aerosol particles in the cold environment in the AIDA chamber, creating a second mode in the number size distribution at diameters between 10 nm and 30 nm. As these nanometer-sized particles from the occasional second nucleation mode retain a higher solute concentration due to the Kelvin effect and have a lower freezing probability due to their small volume compared to the $H_2SO_4/H_2O$ particles of diameters > 100 nm in the principal nucleation mode, they do not significantly contribute to the observed homogeneous ice nucleation modes during expansion cooling.

## 2.3 Determination of freezing onsets

The freezing onsets were determined based on ice crystal number concentration measurements by the OPCs and the depolarization ratio records by SIMONE. We illustrate our approach using the measured time series of experiment number 4 of the TROPIC04 campaign as an example, see Fig. 3. At time $t = 0$ the expansion started with pressure and temperature immediately decreasing (black and red lines, upper panel). The relative humidity with respect to ice computed for gas phase and total water increased accordingly (middle panel). The $H_2SO_4/H_2O$ aerosol particles took up water in order to maintain thermodynamic equilibrium with the environment, which is visible in the faster increase in total water in comparison with the gas phase. This observation was supported by the SIMONE light scattering measurements (bottom panel), in which the forward scattering intensity (2°, dark red) also continuously increased with the beginning of expansion cooling, which is due to the increasing size of the aerosol particles due to the water uptake. The depolarization ratio (red), however, remained constant at the zero background level in this initial time period, which indicates that the particles still had a spherical shape, meaning they were in liquid phase. After about 280 s of pumping, a sharp increase was observed in the depolarization ratio, which is indicative for the formation of aspherical ice crystals in the chamber (red dashed line). The forward scattering signal also showed a strong increase at the same time because the nucleated ice crystals rapidly grow to large sizes with much higher scattering cross sections compared to the unactivated $H_2SO_4/H_2O$ aerosol particles. A few seconds later, the activated fraction $N_{ice}/c_{aerosol}$, relating the number concentration of nucleated ice crystals, $N_{ice}$, to $c_{aerosol}$ from the CPC data, sharply increased, indicating

the detection of ice particles in the OPC measurements (blue dashed line). The temperature and ice saturation ratio at the times derived in this way denoted the ice onset conditions $T_{ice}$ and $S_{ice,fr}$.

For the derivation of $N_{ice}$ from the OPC measurements, it was feasible for most experiments to define a specific optical threshold size to distinguish between smaller-sized $H_2SO_4/H_2O$ aerosol particles and larger sized ice crystals. This procedure was applied to experiments at starting temperatures above about 200 K, where the ice crystals rapidly grew to much larger sizes in comparison to the aerosol population. The optical size threshold was typically set to values between 2 and 10 μm, depending on the aerosol particle size distribution. All particles larger than this size threshold were counted as ice, all smaller particles as $H_2SO_4/H_2O$ aerosol particles. For experiments at temperatures below about 200 K, ice crystal growth is comparatively slow and the sizes are much smaller than those of ice crystals formed at higher temperatures. In these low temperature experiments, $N_{ice}$ was therefore evaluated by analyzing the relative increase of the number concentrations in each size bin of the size distribution measurements instead of defining a size threshold. The ice number concentration and the resulting activated fraction derived from both methods show a sharp increase as soon as ice was formed, and we define the time of this sharp increase as the time of freezing onset. An estimate for the uncertainty of the freezing onset time is given in Sect. 2.4.

Regarding the SIMONE measurements, the onset of ice nucleation was defined as the start of the sharp increase in the back-scattering linear depolarization ratio and the overall back-scattering signals. As SIMONE measures the particle phase in situ, this method is not influenced by any possible artefacts induced by sampling. The time corresponding to the sharp increase of the depolarization ratio was evaluated visually from the individual experimental time series. Figure A1 shows a comparison of the ice onset conditions determined from the OPC and SIMONE measurements. Both the values for $T_{ice}$ and $S_{ice,fr}$ agree well for all analyzed experiments, underlining the consistency between the SIMONE measurement, where the nucleated ice crystals are detected in situ in the AIDA chamber, and the OPC measurement, where the ice crystals are sampled from the chamber.

In general, the freezing onsets determined in the AIDA chamber are sensitive to nucleation rate coefficients between $J_V = 5 \cdot 10^8\,\mathrm{cm}^3\,\mathrm{s}^{-1}$ and $J_V = 10^{13}\,\mathrm{cm}^3\,\mathrm{s}^{-1}$, as reported in Möhler et al. (2003). These nucleation rate coefficients were determined for sulfuric acid aerosol particles with diameters between 0.5 and 2 μm with an estimated freezing probability between 0.02 and 0.5. For the definition of the freezing probability, see Equ. 2 in Möhler et al. (2003). The time delay for the detection of the nucleation peak ranged between 1 and 10 s (Möhler et al., 2003).

## 2.4 Analysis of uncertainties

In this Section, we will discuss the sources of uncertainties for the determination of the freezing onset conditions $T_{ice}$ and $S_{ice,fr}$.

Regarding the freezing onset temperature $T_{ice}$, there are two major sources of uncertainties influencing its determination. The first source of uncertainty is related to the onset time determined from the OPC and SIMONE records, which is described in detail in Sect. 2.3. The comparison shown in Fig. A1 has illustrated that both methods to determine the freezing onset conditions agree very well. Moreover, given the steepness of the $N_{ice}$ and depolarization ratio curves at the onset of ice nucleation, the experimental error in the determination of the freezing onset time is assumed to be small. We determined an uncertainty of $\pm 3\,\mathrm{s}$ for the onset time. Based on estimates for the diffusion length of water molecules in the $H_2SO_4/H_2O$ aerosol particles

at low temperature, which are discussed in greater detail in Sect. 3, we assume that this uncertainty covers the time the initial ice nucleus needs to grow into the detection range of the OPCs. The variability of the gas temperature $T$ in this $6\,\mathrm{s}$ time interval is a first factor contributing to the uncertainty of $T_{ice}$. The second source of uncertainty in the determination of $T_{ice}$ is the temperature distribution inside the chamber. Five vertically and five horizontally distributed temperature sensors inside the AIDA chamber were considered to account for temperature inhomogeneity for each experiment. Prior to the experiment start, the deviation of the temperature measurements of all sensors is usually below $0.1\,\mathrm{K}$. In the course of the experiment, the measured temperatures start to deviate. Especially the deviation in the vertical is increasing, whereas the horizontal deviation remains comparably small. Depending on the spread in vertical temperatures in the $6\,\mathrm{s}$ time interval around the freezing onset, we account for an additional uncertainty in $T_{ice}$. As the deviation of the temperature sensors at the time of freezing onset varies from experiment to experiment, the uncertainty for every single experiment were analyzed individually. A typical deviation at the time of the freezing is about $\pm 0.5\,\mathrm{K}$.

Regarding the ice saturation ratio at freezing onset $S_{ice,fr}$, there are three major sources contributing to the uncertainty in its determination. The first source is the uncertainty in the determination of the freezing onset time from OPC and SIMONE measurements, which is assumed to be $\pm 3\,\mathrm{s}$, as described above. For calculating $S_{ice,fr}$, the measured water vapor pressure $p_w$ is divided by the water saturation vapor pressure with respect to ice $p_{sat,ice}(T)$. For $p_{sat,ice}$, the parameterization given by Murphy and Koop (2005) was used. As this parameterization depends on temperature, the temperature uncertainty including the temperature distribution inside the chamber in the time interval of $6\,\mathrm{s}$ affects the uncertainty of $S_{ice,fr}$ via this parameter and denotes the second factor contributing to the uncertainty in $S_{ice,fr}$. The third source of uncertainty is the direct measurement of $p_w$ by the TDL hygrometer, with a general measurement accuracy of $\pm 5\,\%$. It is assumed that the TDL has a slight tendency to underestimate water vapor content at higher temperatures and to overestimate at lower temperatures ($T <$ about $220\,\mathrm{K}$) (Fahey et al., 2014). For a reasonable correction of the TDL measurements especially at low temperatures, a specific time interval after pumping was stopped and a dense cloud persisted in the chamber was considered. During this time, the interior of the chamber is assumed to be saturated with respect to ice. We can therefore use this time period as a reference to correct our measurements to reasonable values. The highest correction factors were applied on the lowest temperature experiments, which resulted in a shift of up to $0.17\,S_{ice}$ towards lower values. For the highest temperatures, the correction factor was often near zero or only a few percent. Finally, we account for the uncertainty in $S_{ice,fr}$ by considering the three sources of uncertainty individually determined for each experiment.

## 2.5 Overview of AIDA campaigns and experiments

All experiments evaluated for this work are listed in Tables B1, B2 and B3, together with the initial pressure, $p_0$, and temperature, $T_0$, before the start of the expansion, as well as the temperature and ice saturation ratio, $T_{ice}$ and $S_{ice,fr}$, and the associated cooling rate, $cr$, and vertical velocity, $v$, at the derived homogeneous freezing onset. In Figure B1, six histograms show the range and frequency of experimental conditions for all considered AIDA experiments. Except for two experiments, all expansions were started at ambient pressure around $1000\,\mathrm{hPa}$. Note that in the atmosphere, cirrus clouds are formed at much lower pressures. This, however, is not expected to be crucial for investigating the freezing onset conditions in terms of

$T$ and $S_{ice}$. With various start temperatures between 190 K and 236 K, our experiments cover a broad range of temperatures relevant for homogeneous freezing processes. Most of the experiments started between 210 K and 215 K, but with the series of

TROPIC campaigns conducted in 2019 and 2020, we have significantly extended the data set at temperatures below 200 K. The majority of cooling rates determined at the associated freezing onset was around $1\,K\,min^{-1}$, which corresponds to a vertical velocity of about $1.67\,m\,s^{-1}$ assuming dry adiabatic conditions. Only a few particular experiments had higher cooling rates up to $5\,K\,min^{-1}$ (about $8.33\,m\,s^{-1}$). The injected $H_2SO_4/H_2O$ aerosol particle population also covered a broad range of characteristic parameters with mean diameters between 30 nm and 1.1 μm and number concentrations from $30\,cm^{-3}$ up to several

thousand $cm^{-3}$. Finally, the weight percent composition of the $H_2SO_4/H_2O$ solution particles, $wt\%$, at the beginning of each experiment ranged between 32 % and 41 %. As the aerosol particles are assumed to be in thermodynamic equilibrium with the environment before starting the expansion, Model I of the E-AIM model (Clegg et al., 1992; Carslaw et al., 1995; Massucci et al., 1999; Wexler and Clegg, 2002; Clegg and Brimblecombe, 2005) was used to calculate the equilibrium weight percent composition at the measured temperature and relative humidity conditions inside the chamber.

## 3 Results and Discussion

The homogeneous freezing onsets in AIDA were determined as described in Sect. 2.3 using the OPC ice count and SIMONE data. Figure 4 shows the freezing onsets color-coded for the respective AIDA campaign and the corresponding year. For comparison, the WAC-based temperature dependent Koop2000 homogeneous freezing onset lines for nucleation rate coefficients of $J_V = 5 \cdot 10^8\,cm^{-3}\,s^{-1}$ (dotted line) and $J_V = 10^{13}\,cm^{-3}\,s^{-1}$ (dashed line) are shown (Möhler et al., 2003), as well as the

liquid water saturation ratios computed using the parameterization by Murphy and Koop (2005) (solid black line) and the more recent parameterization by Nachbar et al. (2019) (solid blue line). The latter is suggested to be used only for temperatures $T > 200$ K. We found that the homogeneous freezing onsets of $H_2SO_4/H_2O$ particles at temperatures $T > 205$ K agree within the measurement uncertainties to Koop2000. With decreasing temperature, the AIDA measurements show an increasing deviation from Koop2000 towards higher ice saturation ratios. A comparison of the AIDA results to the homogeneous freezing

onsets from previous studies in the $a_w$-$T$-space shown in Fig. 1 can be found in Fig. C1.

The Koop2000 lines are based on homogeneous freezing experiments of 18 different aqueous solutions with a known and constant composition. The homogeneous freezing temperature was measured as a function of the concentration of the solute. Converting the solute concentration into water activity resulted in a close match of the freezing temperatures for the different solutes. It was therefore suggested to formulate the freezing nucleation rate coefficients of aqueous solution particles only

as function of the water activity and the temperature. More recent studies suggested that the homogeneous nucleation rate coefficients can be up to 2 orders of magnitude lower than those given by Koop et al. (2000) (Knopf and Rigg, 2011; Riechers et al., 2013). In terms of water activity, the experimentally derived freezing curve by Knopf and Rigg (2011) is about 0.01 lower than that predicted by Koop et al. (2000). This slightly delayed ice nucleation onset, however, does not account for the much larger deviation of the AIDA results to Koop2000. In the present study, the homogeneous freezing of water with $H_2SO_4$

as the solute was measured as a function of the steady increase of the $S_{ice}$ and the related increase of the water content or

the dilution of the aerosol particles during the AIDA experiments. Thus, the observed differences of the AIDA homogeneous freezing onsets and the Koop2000 derived values may be related to the conversion of the water activity $a_w$ of the solute used by Koop et al. (2000) to $S_{ice}$ used in our studies as the basic water concentration unit in the aerosol system, or the assumption of thermodynamic equilibrium composition of the aerosol particles at the time of freezing onset.

Therefore, the measurement uncertainties of the water vapor concentration and the gas temperature are the first key aspect to discuss and interpret the AIDA data in comparison to Koop2000. In the present experiments, the water vapor concentration was directly measured in situ with a TDL setup (see Sect. 2.1) which was not yet available during the previous AIDA experiments (Möhler et al., 2003; Mangold et al., 2005). For the latter experiments, gas phase water concentration was derived from the total water measurements with the Lyman-$\alpha$ hygrometer FISH (Meyer et al., 2015) by subtracting the aerosol water content,

which was calculated from the sulfuric acid aerosol mass concentration and the molar ratio of sulfuric acid and water in the aerosol particles at thermodynamic equilibrium composition (Möhler et al., 2003). The sulfuric acid mass concentration was determined by ion chromatography analysis of aerosol filter samples. The contribution of the aerosol water content to the total water content was below $1\,\%$ at temperatures around $235\,\mathrm{K}$ and increased to about $10\,\%$ at temperatures around $195\,\mathrm{K}$. Therefore, this correction had only a minor contribution to the overall uncertainty of $S_{ice,fr}$, which varied between $\pm 0.08$ and

$\pm 0.13$ (see Table 1 in Möhler et al. (2003)). The uncertainty of the $S_{ice,fr}$ values derived in this manner varies from about $\pm 0.06$ at the higher temperatures to about $\pm 0.13$ at the lower temperatures. Major contributions to this uncertainty come from the determination of the freezing onset time from OPC particle number and SIMONE scattering intensity and depolarization measurements, the direct measurements of the water vapor partial pressure, and the gas temperature uncertainty and chamber internal variability needed to calculate the water vapor saturation pressure with respect to the ice phase (see Sect. 2.4). Within

these uncertainties, the two data sets from previous and more recent AIDA cloud simulation chamber experiments agree well with each other, and show both a clear trend of increasing freezing onsets to lower temperatures in comparison to Koop2000.

The second important aspect for discussing the deviation of AIDA freezing onsets from Koop2000 is the assumption of thermodynamic equilibrium composition of the aerosol particles in the dynamic environment of changing temperature and relative humidity during the AIDA experiments. The process modeling results by Haag et al. (2003) showed that larger aerosol

particles may have a slightly enhanced sulfuric acid concentration, in particular during experiments at high cooling rates and low temperatures, but that the majority of smaller particles can well follow thermodynamic equilibrium in typical AIDA cloud expansion experiments. A similar result was obtained by Wagner et al. (2008) with a thorough analysis and application of the process model MAID (Bunz et al., 2008) for AIDA experiments with aqueous sulfuric acid aerosol particles performed at temperatures between $230\,\mathrm{K}$ and $205\,\mathrm{K}$. For lower temperatures, the viscosity of sulfuric acid increases with decreasing

temperature (Williams and Long, 1995) and it can even undergo a transition to a glassy state (Koop, 2004). An enhanced viscosity or a glassy state may reduce the water uptake of the aerosol particles and therefore particle growth and dilution. Such effects were discussed to e.g. inhibit homogeneous freezing of secondary organic aerosol particles at $T < 200\,\mathrm{K}$ (Fowler et al., 2020). For aqueous sulfuric acid solutions, Koop (2004) summarized the conditions for transitions to ultraviscous and glassy states in dependence on temperature and aerosol composition. To get information about the phase state of the $H_2SO_4/H_2O$

aerosol particles in the AIDA experiments, we used Model I of the E-AIM model (Clegg et al., 1992; Carslaw et al., 1995;

Massucci et al., 1999; Wexler and Clegg, 2002; Clegg and Brimblecombe, 2005) to calculate the sulfuric acid weight percentage for the temperature and humidity conditions before experiment start and at the conditions of the observed ice onsets (see Tables B1, B2 and B3). For all the experiments, the calculated weight percentages are above the values for a transition to ultraviscous or glassy particles according to Koop (2004) and references therein (see Fig. D1). Especially the glass transition occurs at significantly lower temperatures compared to the AIDA starting and ice onset conditions. This is also supported by the aerosol particle forward scattering intensity measurements with the SIMONE instrument, which showed no evidence of delayed water uptake that would be expected if initially glassy particles were gradually transformed to aqueous solution droplets. To calculate the viscosity of the $H_2SO_4/H_2O$ aerosol particles in the AIDA chamber we applied the parameterization by Williams and Long (1995). This parameterization is only valid for temperatures $> 200\,K$ and for a $H_2SO_4/H_2O$ weight percentage between 30 and $80\,wt\%$ and can therefore only be applied to the starting conditions of AIDA experiments with $T_0 > 200\,K$ (see Fig. D2, data points for experiments with $T_0 < 200\,K$ are displayed in a grey shaded box). The Stokes-Einstein-equation was then used to calculate the diffusion coefficient of water molecules in the aqueous sulfuric acid aerosol particles. From this, we then calculated the diffusion length of water molecules in the aerosol particles on a time scale of $6\,s$, which is much larger than the mean aerosol diameter. This indicates that the water diffusion is still fast enough to keep the aerosol particles in thermodynamic equilibrium.

In order to support our assumption that the aerosol particles freezing in AIDA cloud expansion experiments are at or at least close to thermodynamic equilibrium conditions, we performed an experiment starting at a gas temperature of about $197\,K$, as shown in Fig. 5a. This experiment was not performed with a constant pump rate as all other experiments discussed here, but with an intermediate reduction of the pump rate in order to maintain a constant relative humidity for a longer time period. The experiment was started as other experiments with decreasing temperature and increasing relative humidity, but after the relative humidity passed the Koop2000 line after about $80\,s$, the pump rate was reduced to keep the relative humidity at an almost constant value above the Koop2000 line for about five minutes to give the particles enough time for reaching thermodynamic equilibrium. The forward scattering intensity of the SIMONE measurements (dark red) shows a clear increase at the beginning of the experiments, indicating the growing of the aerosol particles due to water uptake with increasing relative humidity. As soon as the relative humidity is kept at a constant value, no further increase in the forward scattering intensity was observed. Fig. 5b shows the relative humidity and the forward scattering intensity at the beginning of the experiment. The forward scattering intensity clearly increases during the first two minutes of the experiment, which is caused by water uptake as a result of increasing relative humidity, and results in a decreasing viscosity of the particles. As soon as the relative humidity is controlled to an almost constant value by the reduction of the pump speed, no further increase in the forward scattering intensity is observed. The signal even follows the slight variations in the relative humidity, and a delayed water uptake is not indicated. This supports our assumption that the aerosol particles can well follow thermodynamic equilibrium conditions during an AIDA experiment at typical pump rates and related rates of cooling and relative humidity increase. As soon as the pump rate was increased again at about $450\,s$ experiment time, the freezing onset was observed at a relative humidity similar to other experiments with constant pump rate. The fact that no ice formation was observed during the long time period with almost constant relative humidity well above the Koop2000 line supports our assumption, that the observed shift of the

freezing onset to higher relative humidity is not caused by a delayed nucleation due to kinetic limitation of water diffusion to and uptake by the aerosol particles. The result of this experiment also indicates that the observed high freezing onsets may not be caused by a delayed growth of the pristine ice nucleus embedded in a highly concentrated solution layer as suggested in previous studies (Clapp et al., 1997; Bogdan et al., 2006; Bogdan, 2006; Bogdan and Molina, 2010; Bogdan et al., 2013).

In addition, the depolarization measurements with the SIMONE instrument and the signatures of the recorded FTIR spectra show no indication for the formation of sulfuric acid hydrates (Nash et al., 2001), which could occur under the experimental conditions in the AIDA chamber (Koop et al., 1997). An impact of hydrate formation on the observed ice onsets, as shown e.g. in the case of sulfuric acid tetrahydrate (Fortin et al., 2003), is therefore not expected.

Results from Wagner et al. (2008) and more recent measurements during the TROPIC04 campaign show that the mass

fraction of sulfuric acid in the $H_2SO_4/H_2O$ particles measured by FTIR extinction spectroscopy right before the homogeneous freezing onset tend to be slightly lower when compared to the composition of the freezing solutions shown in Koop et al. (2000), but still overlap in the range of uncertainties. This observation supports the assumption that the conversion from the $wt\%$-$T$-space into the $S_{ice}$-$T$-space is a potential explanation for the observed deviation to Koop2000. For this conversion, the water vapor saturation pressure with respect to the supercooled liquid water phase is needed, or more precisely, the ratio of

the saturation pressures with respect to the supercooled liquid and ice phases. The descriptions for the liquid water saturation pressures are rather uncertain (Koop, 2004), and existing parameterizations deviate from each other (e.g. Buck, 1981; Sonntag, 1994; Tabazadeh et al., 1997; Murphy and Koop, 2005; Nachbar et al., 2019). In this work, we are using parameterizations from Murphy and Koop (2005) to calculate both liquid water and ice saturation pressures, but in a recent study Nachbar et al. (2019) suggested a new parameterization for liquid water saturation conditions which indicates that Murphy and Koop (2005) may

increasingly underestimate the liquid water saturation pressure with decreasing temperature below about 220 K. Jensen et al. (2005) discussed that a shift of the liquid water saturation line to higher ice saturation ratios would also shift the homogeneous freezing onset in cirrus formation processes to higher ice saturation ratios. Higher liquid water saturation pressures would also reduce the difference between the AIDA and the Koop2000 freezing onsets for aqueous sulfuric acid aerosol particles. This is shown in Fig. 4, where Koop2000 is shown in combination with the liquid water saturation pressure parameterization of

Murphy and Koop (2005) (black dashed and dotted lines, MK2005) and additional in combination with the more recent line by Nachbar et al. (2019) (blue dashed and dotted lines, N2019). To fully explain the differences between the AIDA and the Koop2000 onsets, the liquid water saturation pressure would need to be even higher at low temperatures than suggested by Nachbar et al. (2019).

Finally, we do not have a solid explanation for the deviation of the $H_2SO_4/H_2O$ homogeneous freezing thresholds observed

in AIDA experiments from the WAC-based homogeneous freezing lines. This deviation may be related to uncertainties in the formulation of physicochemical properties at low temperatures, which are required for the conversion between the Koop2000 parameter space ($a_w$-$T$-space) and the AIDA parameter space ($S_{ice}$-$T$-space), as described above.

Based on the AIDA results and the discussion above, we provide a new fit line for homogeneous freezing of $H_2SO_4/H_2O$ aerosol particles directly measured as a function of $S_{ice}$ and $T$. When plotted the SIMONE and OPC freezing onsets in an

Arrhenius type diagram as $\ln(S_{ice})$ versus $1/T$ in Fig. 6a, the AIDA freezing onsets almost follow a straight line. Using an ordinary least square fit routine the data points are fitted by

$$\ln(S_{ice}) = a + b \cdot \frac{1}{T} \tag{1}$$

with fitting coefficients $a$ and $b$. The coefficients of the fit shown in Fig. 6a are $a = -1.40 \pm 0.05$ and $b = 390 \pm 10\,\mathrm{K}$. This fit transferred into the $S_{ice}$-$T$-space and compared to Koop2000 and the water saturation lines is shown in 6b (red line) with the range of fit uncertainty (red shaded area). The goodness of fit is $R^2 = 0.92$. Applying this new freezing threshold for cloud formation processes in the atmosphere, the homogeneous freezing onset in cirrus formation would be shifted to ice saturation ratios of about 2.0 at temperatures around $185\,\mathrm{K}$. A higher homogeneous freezing onset may explain the high ice saturation ratios occasionally reported for low temperatures in the upper troposphere in some field studies (Jensen et al., 2005; Lawson et al., 2008; Krämer et al., 2009; Krämer et al., 2020). However, it needs to be considered that the fit line only describes homogeneous freezing of $H_2SO_4/H_2O$ aerosol particles under laboratory conditions. Other aerosol species, which could be relevant for homogeneous freezing processes in the atmosphere, are not taken into account. Application to atmospheric conditions therefore needs to be done with caution.

As homogeneously formed cirrus clouds usually have a higher optical depth due to higher ice crystal number concentrations and ice water content, the reflectivity of solar radiation is increased in comparison to heterogeneously formed cirrus (Lohmann et al., 2016). However, homogeneously formed cirrus clouds also reduce the outgoing longwave radiation (OLR) from below to be emitted in space due to their higher optical depth (Lohmann et al., 2016). Consequently, a precise description of homogeneous freezing processes is crucial to understand cloud radiative effects in the present climate as well as in predictions of climate change. This may in particular be relevant for cirrus clouds in the cold tropical tropopause layer (TTL). TTL cirrus clouds have the highest occurrence frequency globally and the strongest radiative warming effect, which enhances their importance for the Earth's climate. The higher homogeneous freezing onsets would suppress the TTL cirrus cloud formation resulting in a decreasing cloud fraction (Jensen et al., 2005; Schoeberl et al., 2016), which would influence the global radiation budget by impacting the OLR (Mitchell and Finnegan, 2009) and reflected shortwave solar radiation.

Besides radiative effects, the presence of cirrus clouds in the tropopause region, especially in the TTL, are assumed to influence the water vapor mass flux from the troposphere to the stratosphere (Jensen et al., 1996; Hartmann et al., 2001; Corti et al., 2006). Air passing the TTL is dehydrated by crossing the cold point in the tropopause layer (Brewer, 1949), which then controls the stratospheric water vapor (Dinh and Fueglistaler, 2014). The supersaturations measured above the cold point tropopause suggest that the water exchange with the stratosphere is $10-20\,\%$ higher than expected (Rollins et al., 2016; Krämer et al., 2020), which might also be related to an enhanced freezing threshold (Schoeberl et al., 2016). To understand stratospheric water contents and related stratospheric chemistry, cirrus formation thresholds (Schoeberl et al., 2016), supersaturations and temperatures (Randel and Jensen, 2013; Randel and Park, 2019) in the tropopause region are of particular importance. Several studies have investigated the relation between the amount of stratospheric water vapor and the properties of the TTL in climate

change scenarios (Solomon et al., 2010; Riese et al., 2012; Randel and Jensen, 2013), which puts the importance of upper tropospheric cirrus formation parameterizations into a larger context.

## 4 Conclusions

In this study, we present and discuss a comprehensive set of homogeneous freezing measurements for $H_2SO_4/H_2O$ aerosol particles. The experiments were conducted at the AIDA cloud simulation chamber and covered a wide range of temperatures, cooling rates, aerosol sizes and number concentrations. At temperatures above $205\,\mathrm{K}$, the measured ice saturation ratios at homogeneous freezing onset $S_{ice,fr}$ agree with the water activity criterion (WAC) based predictions by Koop et al. (2000). Towards lower temperatures, however, the AIDA homogeneous freezing onset results show an increasing deviation from

Koop2000 towards higher ice saturation ratios. For temperatures between $205\,\mathrm{K}$ and $185\,\mathrm{K}$, the WAC-based ice saturation ratios increase from about 1.6 to 1.7, whereas the AIDA measurements show an increase from about 1.7 to about 2.0 in the same temperature range.

For the comparison with Koop2000, we assume the aerosol particles to be in thermodynamic equilibrium at freezing onset. This assumption is justified by previous measurements and process model results (Haag et al., 2003; Wagner et al., 2008). We

also demonstrate that the enhanced freezing onset is not caused by a delayed water uptake and by that a delayed dilution during the AIDA experiments with steady increase of $S_{ice}$. For this we conducted an AIDA experiment during which $S_{ice}$ was kept at an almost constant value well above the Koop2000 line for about 5 minutes. No ice formation or further water uptake of the aerosol particles was observed during this time period, but when increasing $S_{ice}$ in the cloud chamber to higher values, the freezing onset occurred in agreement to results from experiments with steady increase of $S_{ice}$.

Further, the water activities $a_w$ used by Koop2000 have to be converted into $S_{ice}$ directly measured during AIDA experiments for comparing AIDA results with Koop2000. This conversion requires a formulation for the saturation vapor pressures over supercooled liquid water $p_{sat,liq}$, which are rather uncertain at low temperatures (Murphy and Koop, 2005). A recent study by Nachbar et al. (2019) suggested a new parameterization for $p_{sat,liq}$, which, for temperatures below about $220\,\mathrm{K}$, increasingly deviates from Murphy and Koop (2005) to higher values with decreasing temperature. Higher $p_{sat,liq}$ would shift

the Koop2000 freezing onsets to higher ice saturation ratios and would therefore reduce the difference between the AIDA and the Koop2000 freezing onsets. It was already discussed in Jensen et al. (2005) that a shift of the $p_{sat,liq}$ line to higher ice saturation ratios would also shift the homogeneous freezing onset in cirrus cloud formation processes to higher ice saturation ratios. A higher homogeneous freezing onset as derived from our experiments may also explain field observations of high clear-sky supersaturation, which should not occur according to the freezing thresholds predicted by Koop2000 (Jensen et al.,

2005; Lawson et al., 2008; Krämer et al., 2009; Krämer et al., 2020). However, the discussed high freezing onsets only consider homogeneous freezing of $H_2SO_4/H_2O$ aerosol particles under laboratory conditions, without involving other atmospherically relevant aerosol species.

An empirical fit to the AIDA results with the form $\ln(S_{ice}) = a + \frac{1}{T} \cdot b$ with fit parameters $a = -1.40 \pm 0.05$ and $b = 390 \pm 10\,\mathrm{K}$ describes the observed homogeneous freezing onsets of $H_2SO_4/H_2O$ aerosol particles in the chamber. This fit

line is based on direct measurements of the freezing onset conditions at simulated cirrus formation conditions and provides an isoline for homogeneous freezing of sulfuric acid aerosol particles for nucleation rate coefficients between $5 \cdot 10^8$ cm$^{-3}$s$^{-1}$ and $10^{13}$ cm$^{-3}$s$^{-1}$. The application of this fit line to atmospheric conditions requires further work on the physical behavior of H$_2$SO$_4$/H$_2$O aerosol particles at low temperatures and on the involvement of other atmospheric aerosol particle types. Ongoing experiments in the AIDA cloud simulation chamber aim at investigating homogeneous freezing onsets of different solutes and at constraining the descriptions for liquid water saturation pressures to experimental results.

*Data availability.* The measurement data shown in this study are available via the KITopen data repository under https://doi.org/10.5445/IR/1000130863 (Schneider et al., 2021)

*Author contributions.* JS wrote this paper supported by OM, KH and RW. OM, SB, KH, RW, HS, MS, TS, IS and JS planned and conducted the experiments at the AIDA chamber and did the analysis and interpretation of the respective measurement data. MK, MB, CR and TL contributed to the discussion and the interpretation of the data.

*Competing interests.* The authors declare no competing interests.

*Acknowledgements.* We thank all the members of the AIDA staff for their continuous support during the measurement campaigns. This work has been funded by the Deutsche Forschungsgemeinschaft (DFG) through several projects in the years since 2007 (AIDA-HALO projects 47366677 within the HALO priority program SPP-1294, WaterIsotopes project 181901664, project 170852269 within the research unit INUIT FOR 1525, AWiCiT project 311095914, PIRE TropiC project 392369854). We would also like to thank the Gutenberg Research College of the University of Mainz and the Carl Zeiss Foundation for support of this project. We acknowledge support by the KIT-Publication Fund of the Karlsruhe Institute of Technology.

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

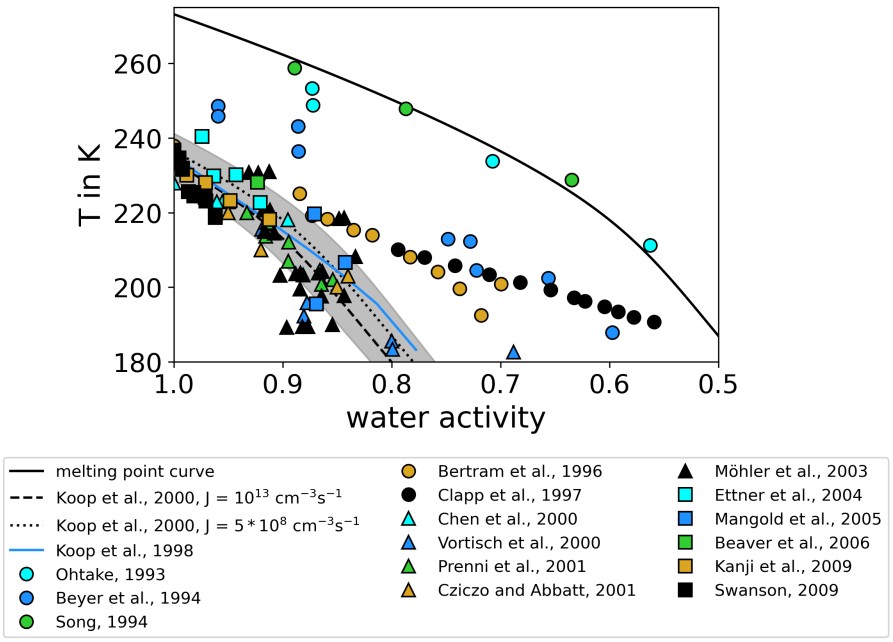

**Figure 1.** Review of homogeneous freezing measurements of $H_2SO_4/H_2O$ solutions. The homogeneous freezing onsets of sulfuric acid solution samples reported in different studies are shown and compared. Most of the studies report onset temperatures and weight percentage of $H_2SO_4$ in the solution samples. We used Model I of the E-AIM model (Clegg et al., 1992; Carslaw et al., 1995; Massucci et al., 1999; Wexler and Clegg, 2002; Clegg and Brimblecombe, 2005) to transfer this weight percentage data into water activity, which is assumed to be equal to the relative humidity (assumption of thermodynamic equilibrium). If ice saturation ratios were given, the parameterizations of Murphy and Koop (2005) for the saturation pressures of supercooled liquid water and ice were used to calculate water activities. Additionally, the melting point line according to Murphy and Koop (2005) (solid line) and the homogeneous freezing thresholds for two different nucleation rate coefficients according to Koop et al. (2000) (dashed and dotted lines) are shown. The grey shaded area is indicative for the uncertainties of these homogeneous freezing thresholds ($\pm 5\,\%$ in water activity) as given in Koop (2004).

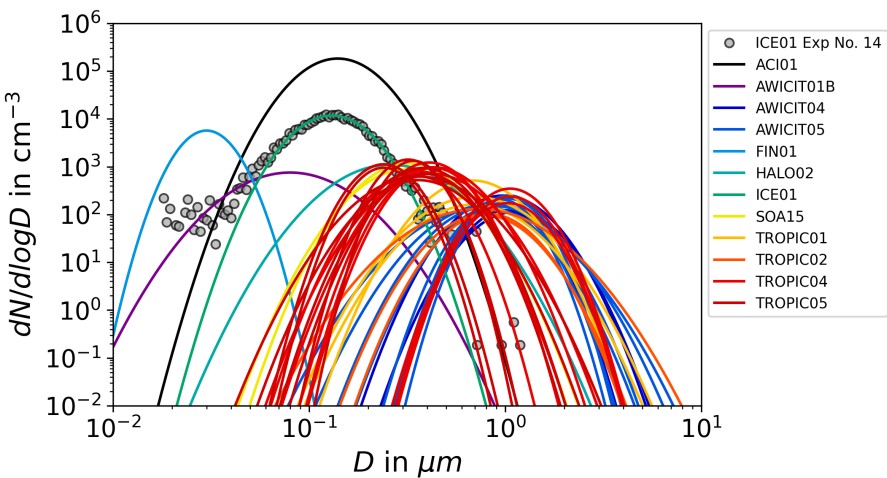

**Figure 2.** Number size distributions of $H_2SO_4/H_2O$ aerosol particles. We show lognormal fitted number size distributions for all AIDA experiments for which size information is available. The grey dots show an exemplary size distribution showing that a longnormal fit generally is adequate to accurately represent the measured size distribution (for details see Sect. 2.2).

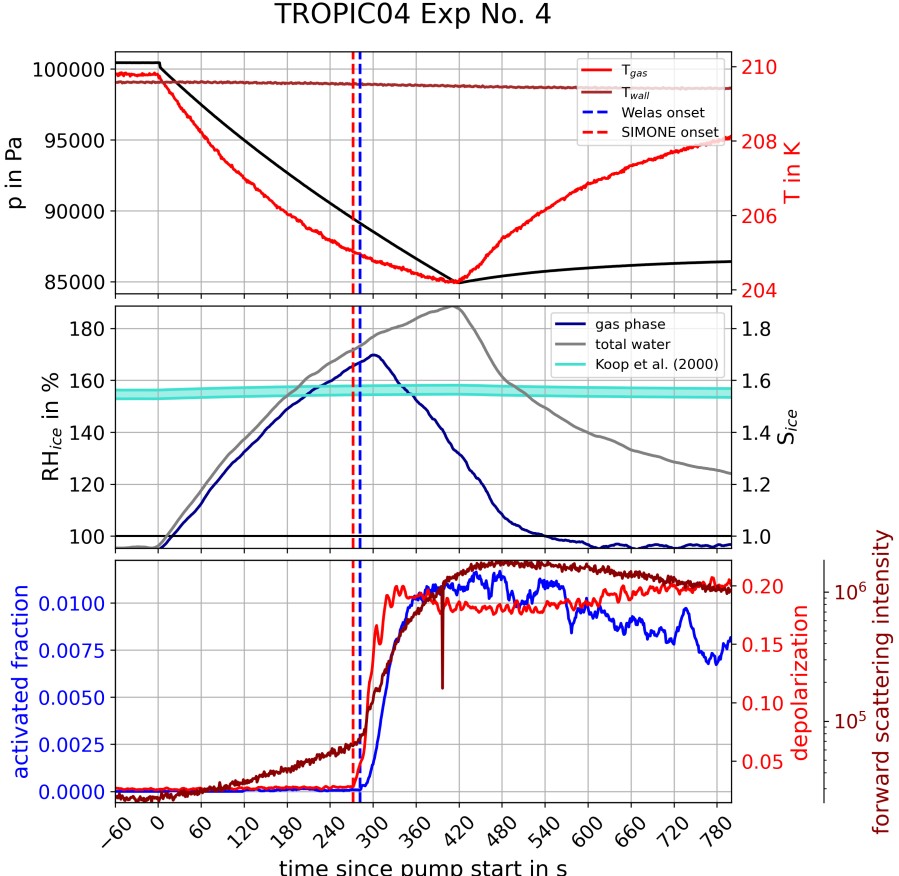

**Figure 3.** Time series of a typical AIDA homogeneous freezing experiment. The upper panel shows the course of pressure (black), as well as gas (red) and wall (dark red) temperature inside the AIDA chamber as a function of time since pump start at $t = 0$. Accordingly, the middle panel shows the course of relative humidity with respect to ice $RH_{ice}$ and ice saturation ratio $S_{ice}$ of gas phase (blue) and total water (grey). The turquoise area marks the range of freezing onset saturation ratios according to Koop et al. (2000). In the lower panel, the activated fraction of $H_2SO_4/H_2O$ particles determined by OPC data (blue) and the forward scattering intensity (dark red) as well as the back-scattering depolarization ratio (red) derived from SIMONE light scattering measurements are presented. The time of homogeneous freezing onsets derived from the activated fraction (blue) and from the SIMONE measurements (red) are marked by the vertical dashed lines.

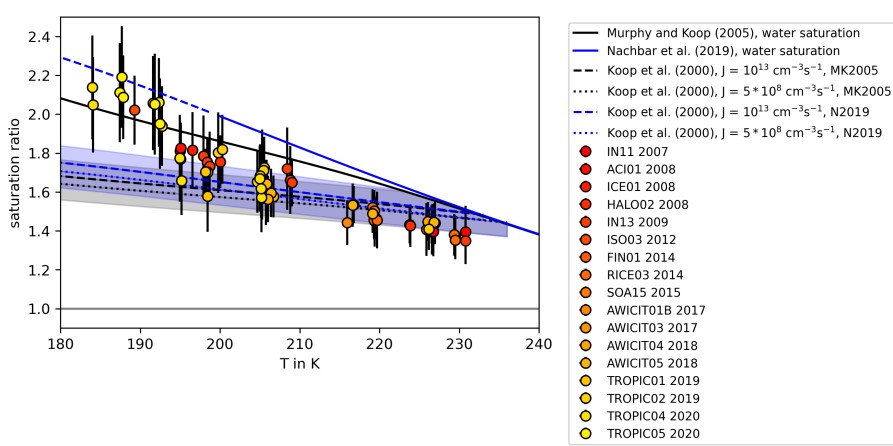

**Figure 4.** Homogeneous freezing onsets of $H_2SO_4$/$H_2O$ aerosol particles. The freezing onset conditions, $T_{ice}$ and $S_{ice,fr}$, are displayed in comparison with the homogeneous freezing thresholds suggested by the WAC-based predictions by Koop et al. (2000) (dashed and dotted lines) using two different parameterizations for the water saturation pressure with respect to supercooled liquid water from Murphy and Koop (2005) (MK2005, black) and Nachbar et al. (2019) (N2019, blue). The blue and grey shaded areas are indicative for the uncertainties of the WAC-based predictions ($\pm 5\,\%$ in water activity) as given in Koop (2004). The used water saturation pressures with respect to supercooled liquid water according to MK2005 (solid black line) and N2019 (solid blue line) are also shown. The colors of the measurement data points represent the different AIDA campaigns in the corresponding years. The oldest campaigns are presented in reddish, whereas the more recent campaigns are shown in yellowish colors.

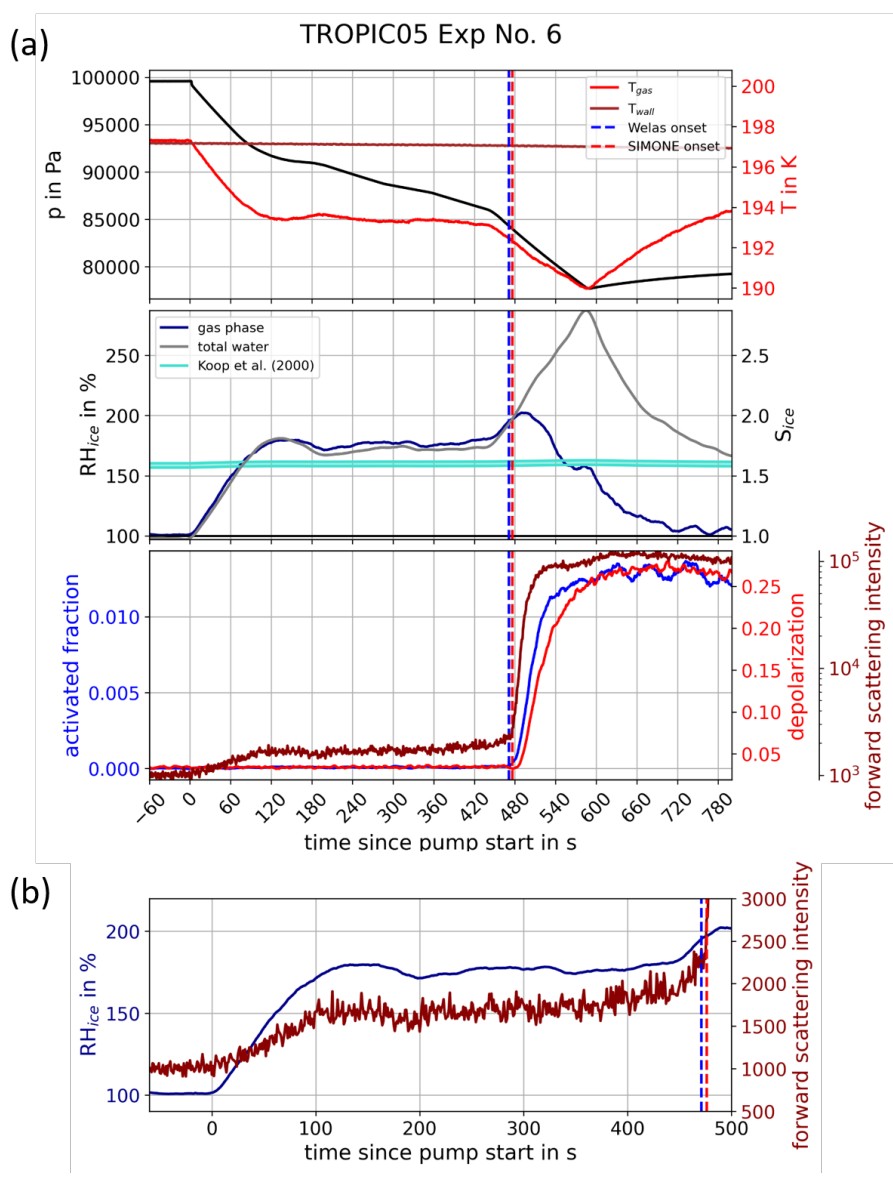

**Figure 5.** Investigation of kinetic limitations with respect to water uptake in the AIDA chamber. Panel (a): The figure is composed in the same way as Fig. 3. For the experiment shown, started at about 197 K, the pump rate was controlled in such a way that the relative humidity with respect to ice stayed relatively constant for about 5 minutes at about 170 %, hence above the homogeneous freezing threshold suggested by Koop et al. (2000). For details on this experiment, see Sect. 3. Panel (b): Enlarged view of the relative humidity and the forward scattering intensity at the beginning of the experiment.

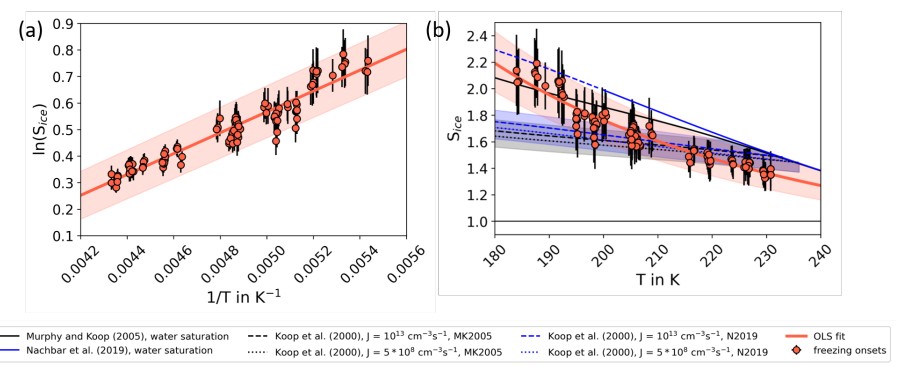

**Figure 6.** New fit line for homogeneous freezing onsets of $H_2SO_4/H_2O$ aerosol particles. Panel (a): The freezing onsets determined by OPC and SIMONE data (red dots) are shown in an Arrhenius plot and fitted by an ordinary least square (OLS) fit with the form $\ln(S_{ice}) = a + \frac{1}{T} \cdot b$. The parameters are $a = -1.40 \pm 0.05$ and $b = 390 \pm 10 \, \text{K}$ and the goodness of the fit is $R^2 = 0.92$. The shaded area is indicative for the uncertainty of the fit parameters. Panel (b): The OLS fit shown in panel (a) is transferred into the $S_{ice}$-$T$-space and compared to Koop2000 and the water saturation lines suggested by Murphy and Koop (2005) and Nachbar et al. (2019). The blue and grey shaded areas are indicative for the uncertainties of Koop2000 ($\pm 5\,\%$ in water activity) as given in Koop (2004).

**Table 1.** Previous measurements of homogeneous freezing of $H_2SO_4/H_2O$ solutions. The columns show the name of the publication, the used measurement device, the sample volume $V$, the cooling rate $cr$ (if available) and information about which parameters, weight percentage composition $wt\%$ or the relative humidity $RH$, were measured or derived to characterize the freezing onsets ('measured' = parameter was directly measured, 'given' = parameter was calculated e.g. using the E-AIM model or others).

| | device | V (µl) | cr (K/min) | wt% | RH |
|---|---|---|---|---|---|
| Ohtake (1993) | test tube | 10000 | 1 | measured | - |
| Beyer et al. (1994) | capillary | 5 | 2 | measured | - |
| Song et al. (1994) | test tube | 2000 | - | measured | - |
| Bertram et al. (1996) | flow tube | $4.2 \cdot 10^{-12}$ | - | measured | - |
| Clapp et al. (1997) | flow cell | $1.4 \cdot 10^{-11}$ | - | measured | - |
| Koop et al. (1998) | cold stage | $6.5 \cdot 10^{-11}$ - $4.2 \cdot 10^{-6}$ | 10 | measured | - |
| Chen et al. (2000) | CFDC | $6.5 \cdot 10^{-14}$ | - | - | measured |
| Vortisch et al. (2000) | levitated drop | $1.1 \cdot 10^{-4}$ | - | measured | - |
| Prenni et al. (2001) | flow tube | $8.7 \cdot 10^{-11}$ | - | given | measured |
| Cziczo and Abbatt (2001) | flow tube | $2.7 \cdot 10^{-10}$ | - | given | measured |
| Möhler et al. (2003) | AIDA | $6.5 \cdot 10^{-11}$ | 1 - 0.1 | given | measured |
| Ettner et al. (2004) | levitated drop | 0.3 - 5.6 | - | given | - |
| Mangold et al. (2005) | AIDA | $4.2 \cdot 10^{-12}$ - $1.4 \cdot 10^{-11}$ | 1 - 0.1 | - | measured |
| Beaver et al. (2006) | flow tube | - | - | measured | - |
| Kanji et al. (2009) | CFDC | $5.2 \cdot 10^{-13}$ | - | - | measured |
| Swanson et al. (2009) | free fall tube | $1.4 \cdot 10^{-8}$ | - | - | measured |
| this study | AIDA | $1.4 \cdot 10^{-14}$ - $7 \cdot 10^{-10}$ | 0.04 - 4.8 | given | measured |

## Appendix A: Comparison of OPC and SIMONE derived freezing onsets

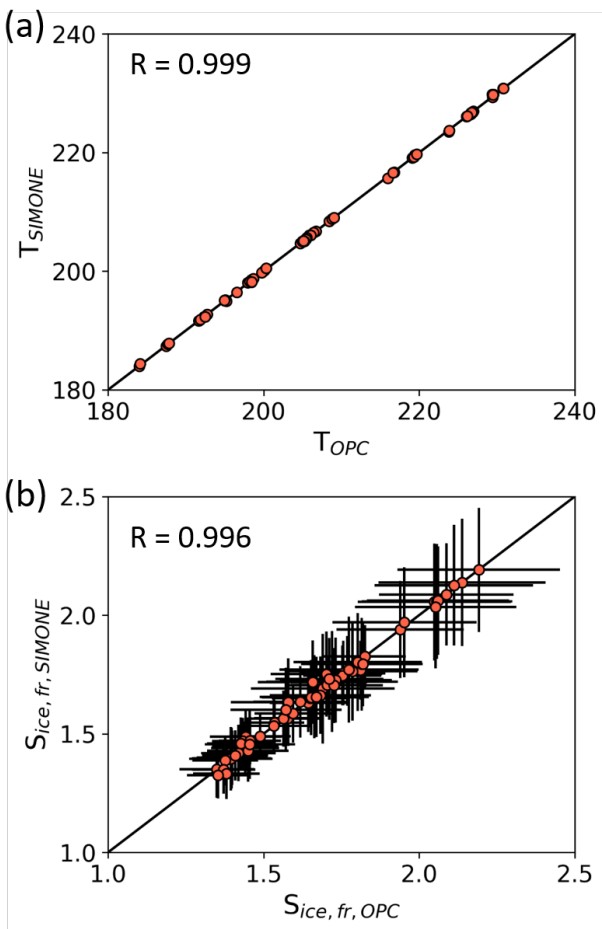

**Figure A1.** Comparison of the homogeneous freezing onset temperature ($T_{ice}$, panel a) and ice saturation ratio ($S_{ice,fr}$, panel b) derived from the SIMONE and OPC data. The respective Pearson's correlation coefficients are R = 0.999 and R = 0.996.

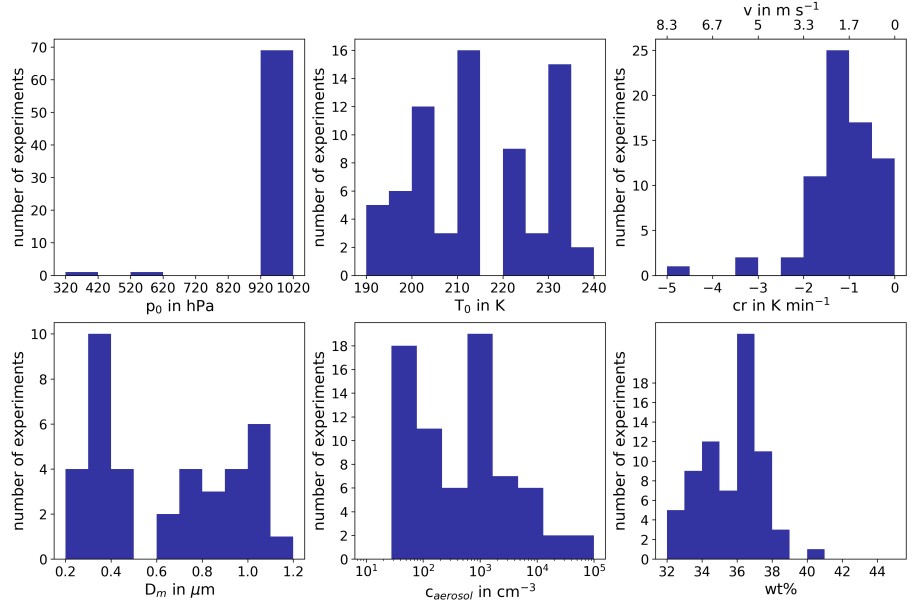

**Figure B1.** Overview of experimental conditions. The six histograms show the distribution and the parameter range of start pressure $p_0$, start temperature $T_0$, cooling rate $cr$, vertical velocity $v$, as well as the median aerosol particle diameter $D_m$, the aerosol particle number concentration $c_{aerosol}$ and weight percentage $wt\%$ of $H_2SO_4$ in the aqueous solution particles before experiment start.

## Appendix B: Overview of AIDA experiments

**Table B1.** Summary of $H_2SO_4/H_2O$ AIDA experiments part I. The considered experiments are listed together with the corresponding campaign, the experiment number, the date of the experiment, the start pressure $p_0$, the start temperature $T_0$, the start relative humidity with respect to ice $RH_{i,0}$ and liquid water $RH_{w,0}$, the freezing onset temperature $T_{ice}$, the ice saturation ratio at freezing onset $S_{ice,fr}$, the cooling rate $cr$ and the corresponding vertical velocity $v$ at the freezing onset.

| Campaign | Exp No. | date | $p_0$ (hPa) | $T_0$ (K) | $RH_{i,0}$ (%) | $RH_{w,0}$ (%) | $T_{ice}$ (K) | $S_{ice,fr}$ | cr (K min$^{-1}$) | v (m s$^{-1}$) |
|---|---|---|---|---|---|---|---|---|---|---|
| ACI01 | 20 | 2008-03-06 | 1016 | 200 | 95 | 51 | 195 | $1.81 \pm 0.07$ | 0.09 | 0.15 |
| | 21 | 2008-03-06 | 1016 | 200 | 93 | 50 | 195.1 | $1.83 \pm 0.07$ | 0.52 | 0.87 |
| | 22 | 2008-03-06 | 1015 | 200 | 94 | 50 | 195 | $1.77 \pm 0.08$ | 0.23 | 0.39 |
| | 24 | 2008-03-07 | 1000 | 225 | 90 | 57 | 219.6 | $1.46 \pm 0.07$ | 0.47 | 0.78 |
| AWICIT01B | 42 | 2017-04-12 | 1005 | 220 | 92 | 56 | 215.9 | $1.44 \pm 0.08$ | 1.15 | 1.92 |
| AWICIT03 | 15 | 2017-09-15 | 1000 | 211 | 92 | 53 | 205.7 | $1.60 \pm 0.11$ | 0.92 | 1.54 |
| AWICIT04 | 6 | 2018-01-15 | 994 | 231 | 97 | 65 | 226.6 | $1.44 \pm 0.07$ | 1.23 | 2.05 |
| | 7 | 2018-01-16 | 987 | 231 | 97 | 64 | 226.5 | $1.43 \pm 0.07$ | 1.52 | 2.54 |
| | 11 | 2018-01-17 | 995 | 221 | 96 | 59 | 216.8 | $1.54 \pm 0.06$ | 0.76 | 1.27 |
| | 37 | 2018-03-29 | 996 | 230 | 95 | 63 | 226.1 | $1.45 \pm 0.07$ | 1.2 | 2.01 |
| AWICIT05 | 2 | 2018-06-04 | 996 | 231 | 97 | 65 | 227 | $1.44 \pm 0.08$ | 1.32 | 2.2 |
| | 4 | 2018-06-05 | 997 | 221 | 96 | 59 | 216.6 | $1.53 \pm 0.07$ | 1.1 | 1.83 |
| | 7 | 2018-06-06 | 1000 | 211 | 95 | 54 | 205.8 | $1.64 \pm 0.07$ | 0.58 | 0.96 |
| | 8 | 2018-06-06 | 999 | 211 | 92 | 53 | 206.7 | $1.58 \pm 0.07$ | 0.81 | 1.34 |
| | 10 | 2018-06-07 | 1003 | 211 | 94 | 54 | 206.4 | $1.60 \pm 0.08$ | 0.75 | 1.24 |
| | 15 | 2018-06-08 | 998 | 224 | 95 | 59 | 219.1 | $1.49 \pm 0.07$ | 0.88 | 1.47 |
| | 16 | 2018-06-11 | 997 | 231 | 96 | 64 | 226.8 | $1.44 \pm 0.07$ | 1.28 | 2.13 |
| | 32 | 2018-06-18 | 1012 | 211 | 95 | 54 | 206 | $1.56 \pm 0.08$ | 1.02 | 1.7 |
| FIN01 | 25 | 2014-11-13 | 999 | 230 | 97 | 64 | 225.9 | $1.41 \pm 0.1$ | 1.48 | 2.46 |
| HALO02 | 10 | 2008-12-11 | 995 | 228 | 93 | 61 | 223.8 | $1.43 \pm 0.07$ | 0.97 | 1.62 |
| | 11 | 2008-12-11 | 998 | 228 | 92 | 60 | 223.8 | $1.43 \pm 0.08$ | 1.88 | 3.13 |
| | 15 | 2008-12-12 | 1010 | 205 | 94 | 52 | 200.1 | $1.76 \pm 0.08$ | 1.13 | 1.88 |
| ICE01 | 14 | 2008-06-27 | 1007 | 201 | 94 | 51 | 196.5 | $1.82 \pm 0.11$ | 1.01 | 1.69 |

**Table B2.** Summary of $H_2SO_4/H_2O$ AIDA experiments part II. The considered experiments are listed together with the corresponding campaign, the experiment number, the date of the experiment, the start pressure $p_0$, the start temperature $T_0$, the start relative humidity with respect to ice $RH_{i,0}$ and liquid water $RH_{w,0}$, the freezing onset temperature $T_{ice}$, the ice saturation ratio at freezing onset $S_{ice,fr}$, the cooling rate $cr$ and the corresponding vertical velocity $v$ at the freezing onset.

| Campaign | Exp No. | date | $p_0$ (hPa) | $T_0$ (K) | $RH_{i,0}$ (%) | $RH_{w,0}$ (%) | $T_{ice}$ (K) | $S_{ice,fr}$ | cr (K min$^{-1}$) | v (m s$^{-1}$) |
|---|---|---|---|---|---|---|---|---|---|---|
| IN11 | 8 | 2007-11-15 | 1008 | 231 | 91 | 61 | 226.6 | $1.40 \pm 0.08$ | 1.13 | 1.89 |
| | 9 | 2007-11-15 | 1008 | 231 | 91 | 61 | 226.7 | $1.40 \pm 0.08$ | 1.61 | 2.68 |
| | 10 | 2007-11-15 | 1008 | 231 | 91 | 61 | 226.7 | $1.42 \pm 0.08$ | 0.5 | 0.83 |
| | 11 | 2007-11-15 | 1009 | 231 | 89 | 60 | 226.7 | $1.40 \pm 0.09$ | 0.35 | 0.58 |
| | 46 | 2007-12-05 | 1005 | 236 | 91 | 63 | 230.8 | $1.39 \pm 0.1$ | 1.01 | 1.69 |
| | 51 | 2007-12-07 | 988 | 211 | 92 | 53 | 205.3 | $1.69 \pm 0.14$ | 0.25 | 0.42 |
| IN13 | 30 | 2009-03-31 | 585 | 235 | 89 | 62 | 230.8 | $1.35 \pm 0.09$ | 1.16 | 1.93 |
| | 36 | 2009-04-02 | 1003 | 214 | 95 | 56 | 208.4 | $1.72 \pm 0.12$ | 0.31 | 0.52 |
| | 37 | 2009-04-02 | 1001 | 214 | 93 | 55 | 208.8 | $1.66 \pm 0.11$ | 0.31 | 0.52 |
| | 38 | 2009-04-02 | 1001 | 214 | 92 | 54 | 209 | $1.65 \pm 0.07$ | 0.75 | 1.24 |
| | 39 | 2009-04-03 | 1005 | 204 | 92 | 51 | 197.9 | $1.79 \pm 0.12$ | 0.59 | 0.99 |
| | 40 | 2009-04-03 | 1005 | 204 | 92 | 50 | 198.4 | $1.75 \pm 0.07$ | 0.1 | 0.16 |
| | 41 | 2009-04-03 | 1005 | 204 | 93 | 51 | 198.7 | $1.73 \pm 0.1$ | 0.29 | 0.48 |
| ISO03 | 43 | 2012-10-25 | 320 | 195 | 95 | 50 | 189.3 | $2.02 \pm 0.09$ | 1.3 | 2.16 |
| RICE03 | 29 | 2014-12-11 | 1003 | 233 | 93 | 63 | 229.4 | $1.38 \pm 0.08$ | 1.34 | 2.23 |
| | 30 | 2014-12-11 | 998 | 233 | 96 | 65 | 229.5 | $1.37 \pm 0.07$ | 1.45 | 2.41 |
| | 31 | 2014-12-11 | 990 | 233 | 97 | 66 | 229.4 | $1.38 \pm 0.08$ | 1.31 | 2.19 |
| | 32 | 2014-12-11 | 1000 | 233 | 92 | 63 | 229.5 | $1.35 \pm 0.07$ | 1.51 | 2.52 |
| | 36 | 2014-12-15 | 996 | 223 | 95 | 59 | 219.4 | $1.45 \pm 0.09$ | 1.69 | 2.82 |
| | 37 | 2014-12-15 | 1003 | 224 | 93 | 58 | 219.3 | $1.46 \pm 0.07$ | 1.8 | 3 |
| | 38 | 2014-12-15 | 1003 | 224 | 96 | 60 | 219.7 | $1.46 \pm 0.1$ | 1.95 | 3.25 |
| SOA15 | 77 | 2015-11-26 | 1001 | 223 | 100 | 63 | 219.1 | $1.52 \pm 0.06$ | 1.01 | 1.69 |
| | 78 | 2015-11-26 | 1003 | 223 | 100 | 62 | 219.4 | $1.50 \pm 0.07$ | 1.31 | 2.18 |
| TROPIC01 | 27 | 2019-07-03 | 1006 | 203 | 99 | 54 | 198.4 | $1.70 \pm 0.11$ | 2.35 | 3.91 |
| | 28 | 2019-07-03 | 1006 | 203 | 95 | 52 | 198.2 | $1.70 \pm 0.07$ | 0.04 | 0.07 |
| | 29 | 2019-07-03 | 1006 | 203 | 93 | 51 | 198.5 | $1.58 \pm 0.12$ | 4.84 | 8.07 |

**Table B3.** Summary of $H_2SO_4/H_2O$ AIDA experiments part III. The considered experiments are listed together with the corresponding campaign, the experiment number, the date of the experiment, the start pressure $p_0$, the start temperature $T_0$, the start relative humidity with respect to ice $RH_{i,0}$ and liquid water $RH_{w,0}$, the freezing onset temperature $T_{ice}$, the ice saturation ratio at freezing onset $S_{ice,fr}$, the cooling rate $cr$ and the corresponding vertical velocity $v$ at the freezing onset.

| Campaign | Exp No. | date | $p_0$ (hPa) | $T_0$ (K) | $RH_{i,0}$ (%) | $RH_{w,0}$ (%) | $T_{ice}$ (K) | $S_{ice,fr}$ | $cr$ (K min$^{-1}$) | $v$ (m s$^{-1}$) |
|---|---|---|---|---|---|---|---|---|---|---|
| TROPIC02 | 4 | 2019-07-09 | 1004 | 230 | 94 | 63 | 226.2 | $1.41 \pm 0.08$ | 1.08 | 1.8 |
| | 14 | 2019-07-11 | 998 | 211 | 94 | 53 | 205.5 | $1.73 \pm 0.09$ | 1.07 | 1.78 |
| | 24 | 2019-07-16 | 1002 | 210 | 96 | 55 | 205.4 | $1.71 \pm 0.1$ | 0.97 | 1.61 |
| | 26 | 2019-07-17 | 999 | 205 | 97 | 54 | 199.8 | $1.80 \pm 0.11$ | 0.88 | 1.47 |
| | 27 | 2019-07-17 | 999 | 206 | 94 | 52 | 200.3 | $1.82 \pm 0.1$ | 1.62 | 2.7 |
| TROPIC04 | 2 | 2020-03-16 | 1004 | 210 | 94 | 53 | 204.7 | $1.65 \pm 0.1$ | 3.15 | 5.25 |
| | 3 | 2020-03-16 | 1004 | 210 | 94 | 54 | 205 | $1.68 \pm 0.1$ | 0.1 | 0.17 |
| | 4 | 2020-03-16 | 1004 | 210 | 95 | 54 | 205 | $1.67 \pm 0.1$ | 0.32 | 0.54 |
| | 5 | 2020-03-17 | 1014 | 210 | 93 | 53 | 205.2 | $1.57 \pm 0.11$ | 3.19 | 5.32 |
| | 6 | 2020-03-17 | 1014 | 210 | 94 | 53 | 205.1 | $1.62 \pm 0.09$ | 0.54 | 0.9 |
| | 15 | 2020-03-19 | 1006 | 200 | 87 | 47 | 195.2 | $1.66 \pm 0.11$ | 2.29 | 3.82 |
| | 16 | 2020-03-19 | 1006 | 200 | 88 | 48 | 195 | $1.77 \pm 0.13$ | 0.64 | 1.06 |
| | 26 | 2020-03-24 | 1015 | 190 | 95 | 48 | 184 | $2.14 \pm 0.13$ | 1 | 1.67 |
| | 28 | 2020-03-24 | 1012 | 190 | 84 | 43 | 184.1 | $2.05 \pm 0.12$ | 1.68 | 2.8 |
| TROPIC05 | 2 | 2020-10-12 | 1005 | 197 | 105 | 57 | 191.6 | $2.06 \pm 0.12$ | 1.05 | 1.76 |
| | 3 | 2020-10-12 | 1003 | 197 | 102 | 54 | 192.3 | $2.06 \pm 0.11$ | 1.88 | 3.14 |
| | 4 | 2020-10-13 | 999 | 197 | 99 | 52 | 191.8 | $2.05 \pm 0.13$ | 0.54 | 0.9 |
| | 5 | 2020-10-13 | 996 | 197 | 101 | 53 | 192.7 | $1.94 \pm 0.11$ | 0.98 | 1.63 |
| | 6 | 2020-10-13 | 996 | 197 | 101 | 53 | 192.5 | $1.95 \pm 0.12$ | 1.25 | 2.08 |
| | 7 | 2020-10-14 | 999 | 193 | 105 | 54 | 187.4 | $2.11 \pm 0.12$ | 1.27 | 2.11 |
| | 8 | 2020-10-14 | 998 | 193 | 95 | 49 | 187.7 | $2.19 \pm 0.12$ | 1.53 | 2.56 |
| | 10 | 2020-10-15 | 999 | 193 | 101 | 52 | 187.8 | $2.09 \pm 0.1$ | 0.63 | 1.05 |

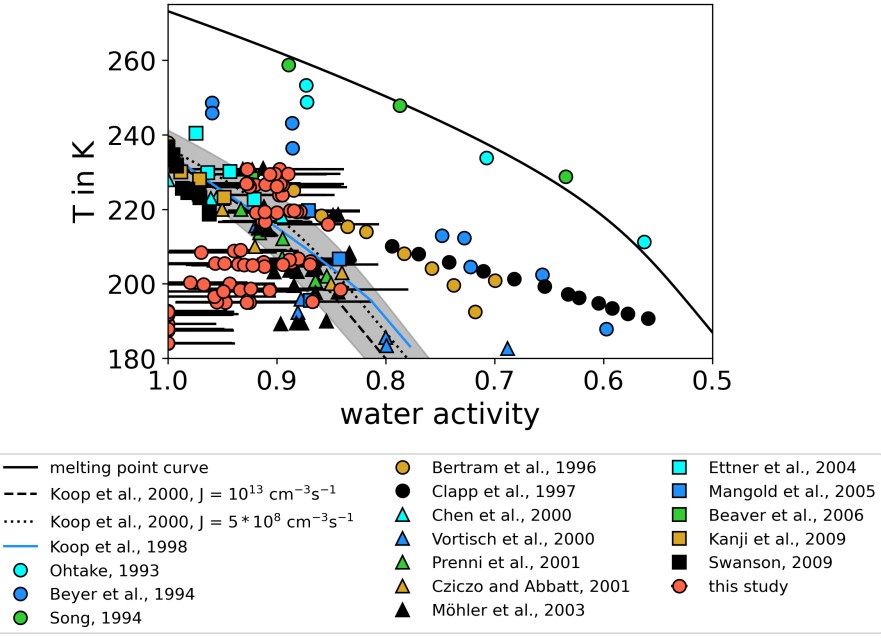

**Figure C1.** Summary of homogeneous freezing measurements of $H_2SO_4/H_2O$ solutions in the $a_w$-$T$-space. The homogeneous freezing onsets of sulfuric acid solution samples shown in Fig. 1 are complemented by the AIDA results from this study (red points). The results in the $S_{ice}$-$T$-space were converted into the $a_w$-$T$-space by assuming equilibrium conditions $a_w = RH_w$ and by using the parameterizations for the water vapor saturation pressures with respect to ice and supercooled liquid water given by Murphy and Koop (2005). The uncertainties of the calculated $a_w$ values vary between $\pm 0.036$ and $\pm 0.076$.

## Appendix C: AIDA results in $a_w$-$T$-space

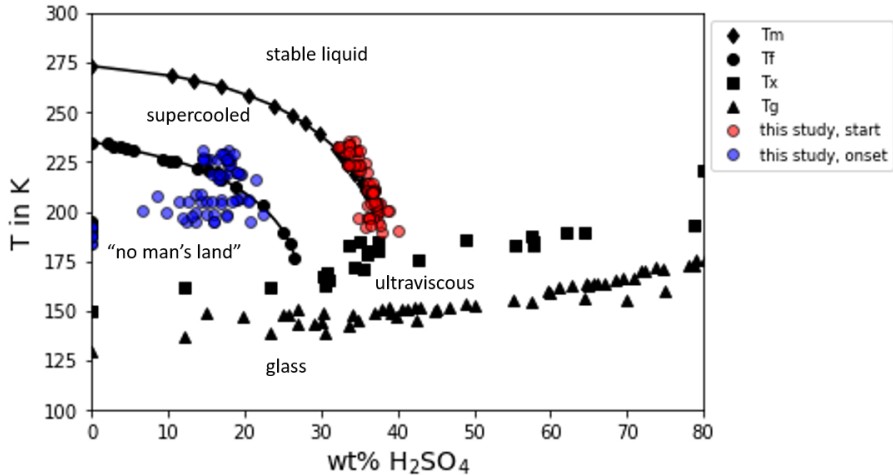

**Figure D1.** Phase diagram of sulfuric acid adapted from Koop (2004). The phase diagram of sulfuric acid as given by Koop (2004) and references therein is shown and complemented by the calculated weight percentage concentrations of the sulfuric acid aerosol particles before the start of the AIDA experiments of this study (red dots) and at the observed ice onset during the experiments (blue dots).

**Appendix D: Phase state of supercooled sulfuric acid aerosol particles**

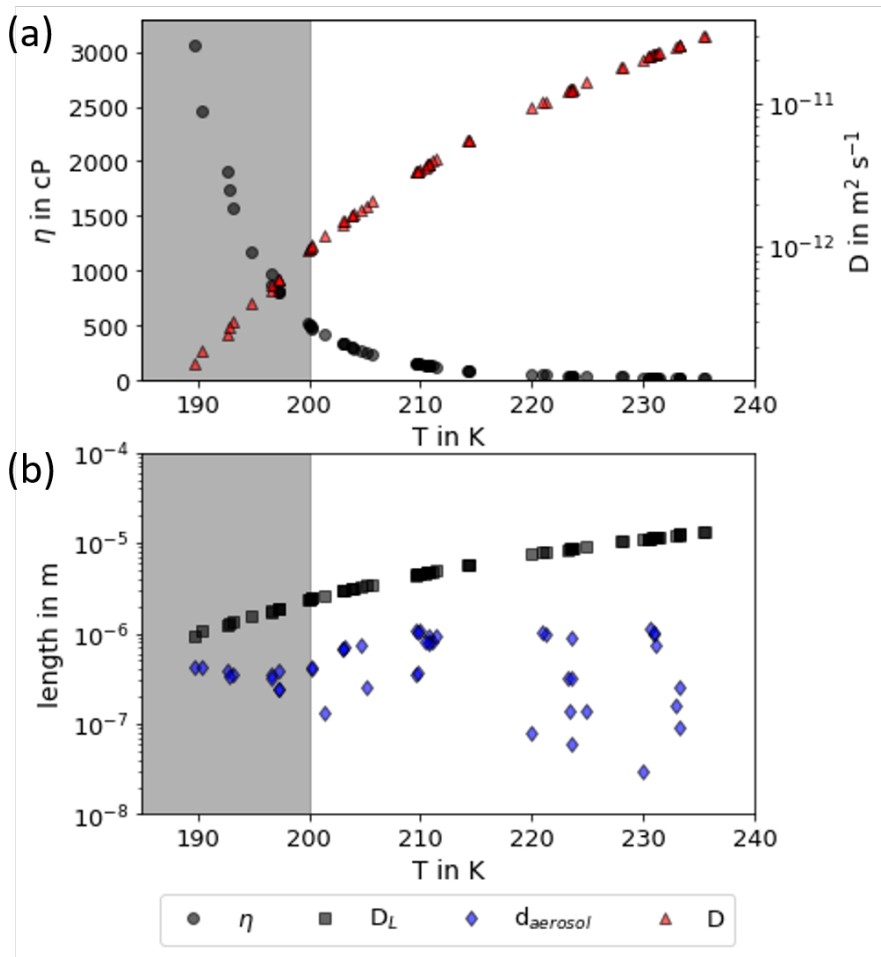

**Figure D2.** Viscosity and diffusion coefficients for the start conditions of the AIDA experiments. Panel (a): Viscosity (black dots) and diffusion coefficients (red triangles) calculated using the parameterization of Williams and Long (1995) and the Stokes-Einstein-equation. Panel (b): Comparison of the mean particle diameter of the sulfuric acid particle population (blue diamonds) to the diffusion length of water molecules (black squares) on a time scale of 6 s. Note that the shown data points represent the starting conditions of the AIDA experiments. As the parameterization of Williams and Long (1995) is only valid for T > 200 K, the data points of experiments at lower temperatures are displayed in a grey shaded box.