# Peer review of "High Homogeneous Freezing Onsets of Sulfuric Acid Aerosol at Cirrus Temperatures"

_Atmospheric Chemistry and Physics, 2021_

## Referee Comment (RC1)

**Review for "High Homogeneous Freezing Onsets of Sulfuric Acid Aerosol at Cirrus Temperatures" by Schneider et al.**

The manuscript submitted to *Atmospheric Chemistry and Physics* titled "High Homogeneous Freezing Onsets of Sulfuric Acid Aerosol at Cirrus Temperatures" by Schneider et al. presents new and reanalyzed data on homogeneous freezing of aqueous sulfuric acid aerosol in the AIDA chamber. Homogeneous ice nucleation studies at such large supercoolings is certainly relevant topic and important for atmospheric science. The authors report the onset of ice nucleation to be below water saturation and follow the water activity criterion (WAC) from Koop *et al.*[1]. However, their results deviate from Koop *et al.*[1] at 185-205 K. After a thorough uncertainty analysis and clearly stating their assumptions, they conclude that this deviation is significant. They claim that the deviation may be because water saturation curves with respect to ice and water are uncertain and suggest that the estimate by Nachbar *et al.*[2] instead of Murphy and Koop[3] would cause deviations to decrease. Finally, the authors make a claim about the consequence of their results to ambient cirrus clouds.

Overall, the intro, methods and results of this manuscript are well written. The methods are described well and the error analysis is sound. However the discussion needs great improvement. There are major comments that cast the authors conclusions in serious doubt. These have to do with the lack of discussion of the physical evidence for the WAC, the uncertainty for the WAC, the mixing time of the particles, and finally, their suggestion of treating homogeneous freezing for cirrus clouds as only from sulfuric acid. A few minor comments exist. Overall, I cannot recommend publication at this time without significant revision.

**Major comments**

1. There is a lack of any physical reasoning. The WAC is not an empirical parametrization of aqueous sulfuric acid onset freezing temperatures. It is a physical description of freezing of a variety of solutes at ambient pressure, and of pure water at high pressure[1,4]. In Koop[4], physical evidence is presented that freezing and melting temperatures of pure water at high pressure and highly concentrated aqueous solution at ambient pressure are similar and are the result of similar affects

on the water hydrogen bonding structure. To be fair and balanced, if the authors claim their data deviates from the WAC, then they must claim a physical reason for this and independent evidence to support their reasoning. For example, if WAC is solute independent (l. 38-39), why do they suspect sulfuric acid is such a special case? Does the hydrogen–hydrogen radial distribution function[4] of sulfuric acid aqueous solutions deviate from high-pressure water at the same water activity? Does it deviate from other solutes below 205 K at the same water activity, but not deviate at warmer temperatures? If they cannot explain their results physically or come up with a realistic quantitative measure, it is acceptable that the authors include a statement that they do not know a physical reason why such a deviation would occur.

2. There is uncertainty of the WAC that should be included. I appreciate the authors experimental uncertainty analysis, however, they lack the uncertainty analysis for the WAC. They must include the uncertainty in the WAC lines for a fair comparison. Koop[4] claims a uncertainty up to 5% in temperature from the freezing line, which translates to an error close to $\pm 15$ K in temperature and $\pm 0.08$ in water activity, $a_w$, at homogeneous freezing temperatures of 185 K. Please check this. In order for a fair comparison with WAC, the authors must calculate errors on the WAC lines at all temperatures and show them in their figures.

3. Mixing time of high concentrated aqueous solutions at low temperature. On l. 294, the authors claim their assumption that particles are well-mixed and in equilibrium with their humidified environment. Support for this assumption is given[5] for temperatures $> 205$ K and for the experiment performed by the authors at 194 K in Fig. 6. However, there remains some doubt about this assumption, and the particles may be highly viscous to the point of limiting mixing within them due to slow molecular transport.

Whether a particle is or is not well-mixed can depend on the entire relative humidity history, even before the start of experiments. If experiments began at a relative humidity with respect to ice, $RH_i$, and temperature in which aqueous aerosol particles were initially in a glassy state, it would take time for a glassy and inhomogeneously mixed particle to transition to mixed and satisfy the authors assumption. For example, Berkemeier *et al.*[6] has shown that for glassy organic aerosol, a humidity induced transition to well-mixed particles can take 1600 s starting at 215K

and cooling to 212 K and consequently humidity increasing from 60% to 87%. Although glassy inorganic solutions may behave differently than glassy organics, experiments by the authors here were 3 to 4 times faster and therefore, a kinetic limitation cannot be ruled out. If the authors began their experiment in Fig. 6 at a lower $RH_i$ as they did for their experiments in Fig. 3, would kinetic limitations be observed? Evidence for a kinetic limitation comes from the sulfuric acid phase diagram[4]. When $RH_i = 95\%$ and $T = 185$ K, the weight percent of sulfuric acid solution in equilibrium is roughly $50\%$[7-9] and this is exactly at the boundary of ultra-viscous solutions. The authors should include the starting $RH_i$ in the appendix tables.

The uncertainty here is large, of course, due to extrapolation and seen by the scatter in crystallization temperatures of glassy particles upon warming in Fig. 4 of Koop[4]. What I expect is that the authors include a value of viscosity, molecular diffusion coefficients, or mixing time scales at their exact experimental conditions measure in literature. The authors have not shown evidence for this assumption for temperatures down to 185 K. I do not know of existing viscosity or diffusion coefficient measurements in this temperature and humidity range. If they exist, what is the variability. I do not recommend extrapolating from common measurements. A lack of measurements would cast doubt on this assumption, and thus their conclusions.

4. The authors want their fitted line in atmospheric models and replace the WAC (l. 391-392). Inherent in this is that only sulfuric acid aerosol particles nucleate homogeneously to form cirrus clouds in models, is to discard the presence of other solutes such as secondary organic aerosol; nitrates or sea salts, is to treat homogeneous freezing only at the authors' measured onset, and is to not account for homogeneous ice nucleation rate coefficients (as function of water activity and temperature). As there is no given physical explanation for their data, this suggestion is a large leap backward for understanding atmospheric physics and chemistry. The authors certainly make a line go through data points, however it is not appropriate to use this line to predict the formation for cirrus clouds. Up scaling a purely empirical parametrization from the AIDA chamber to real atmospheric conditions is an extrapolation outside of their experimental conditions. If the authors want to replace Koop *et al.*[1], then more work needs to be done to quantify and understand the physics of homogeneous ice nucleation and apply that understanding to the range of temperature,

water activity and nucleation rate coefficients valid for their measurements and consistent with the over 20 years of observation supporting the WAC. Please remove any mention of suggesting to use this parametrization in atmospheric models on l. 344, l. 351-352, l. 384-385 and l. 391-392 and in the last 2 sentences of the abstract. These are the instances I have found.

**Minor comments**

1. l. 6. The WAC is not a function of aerosol particle size. Likewise, it is not a function of time either.

2. l. 23-24 and 386-407. The authors certainly review and discuss cirrus cloud formation and radiative effects, however, these are not conclusions. No cloud model or any calculations of radiative forcing were made here to support these statements. In the abstract I suggest the following rewrite or something similar, "Our results are discussed in the context of predicting the formation of cirrus clouds and related cloud radiative effects." In addition, these conclusions need to be moved to the results and discussion section.

3. l. 44-47. and 327-329. It was already stated by Koop[4] that thermodynamic models (to calculate water activity of solutions or saturation vapor pressures) extrapolated to these low temperatures can be large sources of errors. I suggest to add this reference here.

4. l. 46. What E-AIM model did the authors use? I suppose Model I[7–9]. Please check the correct references on the E-AIM website.

5. l. 50-51. I think there is a mistake here. Higher values of $\Delta a_w$ should yield higher values of $J_V$.

6. l. 61-63. Would the authors take care to please check the ambient ice saturation ratios for these studies? The authors language gives the impression that high $RH_i$ at or above 200% happens all the time at temperatures colder than 200 K. This is misleading. It is directly stated in the abstract Krämer et al.[10] that the highest $RH_i$ for clear sky is about 150%. Krämer et al.[10] shows a distribution of $RH_i$ and there are very rarely any measurements at or above 200%. I count

about 7 yellow squares in Fig. 7(e) of Krämer *et al.*[10] at or above 200%, but practically all data is bounded by or scattered around homogeneous freezing. This statement misrepresents the findings of Krämer *et al.*[10], and I would encourage them to be more specific and representative of the previous research they are citing. Please check all citation here. Ambient in-cloud and clear-sky $RH_i > 150\%$ occurs mostly $< 2\%$ of the time.

7. l. 334-335. If the $\Delta a_w$ values would be used together with Nachbar *et al.*[2] to plot a new onset curve, would everything be within error bars? They claim that difference would be reduced, but why not show these differences and if they can completely explain the deviations they observe.

8. l. 341 and 350. The reason why the authors show 2 different fit parameters and procedures here is not clear. Would the authors please choose one, and remove the one you do not want your readers to use from the manuscript?

9. l. 350-351. It is not necessary to state the same parameters and errors twice in adjacent sentences. Please remove.

10. WAC freezing curves in figures. It is not clear that the freezing curves for constant $J_V$ are correctly determined. In a later paper, Koop and Zobrist[11] altered the homogeneous freezing curve of Koop *et al.*[1] by an offset in $\Delta a_w$ of 0.008. It appears this is not accounted for in this manuscript.

11. Figure 3. Why doesn't the activated fraction go to 1.0? I expect that homogeneous ice nucleation is so fast that all particles should turn to ice? Is there that much vapor depletion due to the first few ice crystals that form that the authors cannot nucleate all aqueous droplets?

12. Figure 4 and 6. There is a bit of a bias here (some systematic uncertainty that is not explained?) that the majority of ice saturation data points at temperatures warmer than 210 K are lower than homogeneous freezing estimates. Then, data is mostly higher than homogeneous freezing estimates when temperatures are colder than 210K. Would the authors care to comment on this somewhat systematic uncertainty? In addition, if there is no theory or physical explanation to back up their measurements (see major comment), their data is more suspect to unknown experimental artifacts or errors.

**References**

[1] T. Koop, B. P. Luo, A. Tsias and T. Peter, *Nature*, 2000, **406**, 611–614.

[2] M. Nachbar, D. Duft and T. Leisner, *J. Chem. Phys.*, 2019, **151**, 064504.

[3] D. M. Murphy and T. Koop, *Q. J. R. Meteorol. Soc.*, 2005, **131**, 1539–1565.

[4] T. Koop, *Z. Phys. Chemie-Int. J.*, 2004, **218**, 1231–1258.

[5] R. Wagner, S. Benz, H. Bunz, O. Möhler, H. Saathoff, M. Schnaiter, T. Leisner and V. Ebert, *J. Phys. Chem. A*, 2008, **112**, 11661–11676.

[6] T. Berkemeier, M. Shiraiwa, U. Pöschl and T. Koop, *Atmos. Chem. Phys.*, 2014, **14**, 12513–12531.

[7] K. S. Carslaw, S. L. Clegg and P. Brimblecombe, *J. Phys. Chem.*, 1995, **99**, 11557–11574.

[8] M. Massucci, S. L. Clegg and P. Brimblecombe, *J. Phys. Chem. A*, 1999, **103A**, 4209–4226.

[9] S. L. Clegg and P. Brimblecombe, *J. Phys. Chem. A*, 2005, **109**, 2703–2706.

[10] M. Krämer, C. Rolf, N. Spelten, A. Afchine, D. Fahey, E. Jensen, S. Khaykin, T. Kuhn, P. Lawson, A. Lykov, L. L. Pan, M. Riese, A. Rollins, F. Stroh, T. Thornberry, V. Wolf, S. Woods, P. Spichtinger, J. Quaas and O. Sourdeval, *Atmos. Chem. Phys.*, 2020, **20**, 12569–12608.

[11] T. Koop and B. Zobrist, *Phys. Chem. Chem. Phys.*, 2009, **11**, 10839–10850.

---

## Author Comment (AC2)

Referee comment on "High Homogeneous Freezing Onsets of Sulfuric Acid Aerosol at Cirrus Temperatures" by Julia Schneider et al., Atmos. Chem. Phys. Discuss., https://doi.org/10.5194/acp-2021-319-RC2, 2021

We thank referee #2 for his or her thoughtful comments and feedback. Please find below our responses and suggestions for the manuscript revision, with the referee comments in black, our answers in green, and suggested changes or additions to the manuscript in blue.

**General Comment**

This manuscript titled "High Homogeneous Freezing Onsets of Sulfuric Acid Aerosol at Cirrus Temperatures" by Schneider et al. reports homogeneous freezing of aqueous sulfuric acid aerosol using the AIDA chamber at conditions relevant for cirrus cloud regime. The highlight of this study is the significant deviation of the onsets of ice formation from the water activity criterion (WAC) (Koop et al., 2000) at temperatures below about 200 K. The manuscript is well-written, clearly discussing the uncertainties in their measurements and the underlying assumptions.

However, the discussion/conclusions rely entirely on a single type of aerosol particle, with the absence of any physical interpretation for deviations from WAC, and in-depth discussion of mixing time of investigated aerosol particle at temperatures below 200 K. Nonetheless, I recommend the manuscript for publication after the authors have addressed the following major questions and revised the manuscript accordingly.

**Major Comments**

1. Kinetics of sulfuric acid aerosol particles at cirrus temperatures: I understand that not much work has been done related to the activation kinetics and particle phase at such low temperatures. The authors handled this issue with their test for kinetic limitations at 197 K. However, the majority of their ice supersaturation results still comply with the WAC at around that temperature. I am curious why a similar kinetic test was not considered at, e.g. 190 K, where the deviations are suggested to be significantly higher?

Please note that the experiment shown in Fig. 5 was started at 197 K, but the actual ice onset temperature was observed at a lower temperature of about 193 K with a corresponding ice saturation ratio of about 1.95. Therefore, this experiment was done in the temperature regime, where the observed deviation in onset supersaturation is already significant and we consider it as representative for the other experiments with significant deviation. We are planning more AIDA homogeneous freezing experiments with other solutes and will take up your suggestion to carry out more experiments with variable pumps speeds or rate of pressure change.

2. Thermodynamic equilibrium: Based on previous work (Williams and Long, 1995), it won't be surprising that these sulfuric acid aerosol particles are extremely viscous at such low temperatures, likely approaching glassy state (if not already). This could really slow down the mixing. This casts some doubts on whether the particles attained thermodynamic equilibrium condition. I strongly recommend adding a discussion related to this issue and the implications it can have on the presented results and its interpretation.

We agree that the particle phase and the related mixing time at low temperatures are an important aspect to consider in the measurements of this study. We also discussed those aspects in the response to major comment 3 of referee #1, which is attached below:

As a first step, we included the start relative humidity with respect to ice $RH_{i,0}$ and liquid water $RH_{w,0}$ to the tables in the appendix, as suggested.

To illustrate the start conditions and the sulfuric acid phase at the ice onset, we adapted Fig. 4 in Koop, 2004 and added the data points for the start composition of the sulfuric acid aerosol particles in wt% $H_2SO_4$ for our experiments and the data points for the composition at ice onset (see Fig. 1 in this response, red and blue dots). The weight percentage composition was determined by using Model I of the E-AIM model with the measured temperature and relative humidity inside the chamber. As described in the referee's comment above, a glassy state of the sulfuric acid aerosol particles in our experiments would strongly influence the observed freezing process due to slow mixing processes. In Fig. 1, we can see that all experiments are above the conditions of ultra-viscous and glassy particles, according to the phase diagram in Fig. 4 in Koop, 2004 and the references therein. Especially the glass transition conditions given in this figure are at significantly lower temperatures compared to our starting and ice onset conditions. This is supported by our SIMONE measurements, which show an increasing signal in the forward scattering intensity between pump start and ice onset, for all the experiments. This increase shows that the aerosol particles are able to take up water and dilute. Only from this observation, it cannot be completely ruled out that an enhanced viscosity at low temperatures decelerates the water uptake, so that it might be not sufficient to maintain particles in thermodynamic equilibrium with the environment. For the investigation of a potential decelerated water uptake, we refer again to the experiment shown in Fig. 5 in the manuscript. In this experiment, the relative humidity was kept nearly constant above Koop2000 for about five minutes. Firstly, we observe no ice formation in this time period, which indicates that the AIDA ice onset is not higher than the Koop2000 line due to a delayed ice detection. Secondly, the forward scattering intensity of the SIMONE instrument (dark red line, third panel) shows a significant increase in the first two minutes of constant pumping, which is related to the water uptake of the particles. In the following five minutes of constant relative humidity (slower pumping), no further uptake of the particles is observed. To illustrate this more precisely, we added a second panel to Fig. 5 in the manuscript (see Fig. 2 in this response), which shows a direct comparison of the relative humidity and the forward scattering intensity on a smaller scale. In this panel, we clearly observe the particle diluting in the first two minutes of the experiment, which generally decreases the particles' viscosity. After that period, the particles could have continued taking up water, but no further increase in the forward scattering is observed, as soon as the pump speed is lowered. Rather, the forward scattering intensity follows nicely the slight variations in the relative humidity, but a decelerated or delayed water uptake is not indicated.

[revised manuscript text omitted]

3. Physical interpretation of the deviations from WAC: There is no attempt made at understanding the microphysical picture underlying the reported deviations from WAC. It is unclear why sulfuric acid system would behave in such a fashion. I recommend that the authors state this lack of understanding clearly in the manuscript.

First of all, we would like to state that we present here experimental results after careful discussion and uncertainty analysis, with direct measurements of the temperature and ice saturation ratio at the point of homogeneous freezing onset for aqueous sulfuric acid aerosol particles. That our data deviates from those derived from the WAC-based parameterization by Koop et al. (2000) is an experimental result, and does not require any physical reasoning. Physical reasoning is important and may be helpful when discussing possible reasons for this disagreement, and some are included in the discussion in section "3. Results and discussion", but unfortunately we do not yet have a definite explanation for the observed deviation.

We therefore included the following statement into Section "3. Results and Discussion" to emphasize that we are not questioning the underlying physical principles of the WAC, but rather its application to the AIDA parameter space at low T:

**Finally, we do not have a solid explanation for the deviation of the $H_2SO_4/H_2O$ homogeneous freezing thresholds observed in AIDA experiments from the WAC-**

**based homogeneous freezing lines. This deviation may be related to uncertainties in the formulation of physicochemical properties at low temperatures, which are required for the conversion between the Koop2000 parameter space ($a_w$-T-space) and the AIDA parameter space ($S_{ice}$-T-space), as described above.**

4. Atmospheric models: Sulfuric acid aerosol particles certainly dominate in the stratosphere, however, there are several studies showing the presence of various other components such as organics and inclusions of aluminum and silicon (Murphy et al., 2014), with even more variety of components present in the upper troposphere. While the presented study focuses only on sulfuric acid aerosol particles. Keeping this in mind as well as the comments stated in 1-3 above, I find the authors' suggestion to use their parameterization in atmospheric models over WAC a bit far fetched. I do not agree with this suggestion and recommend removing this part. In addition, the authors should state this caveat in their abstract, discussion and conclusions. There needs to be more work done on other atmospherically relevant aerosol particles at such low temperatures and high ice supersaturation conditions, to establish whether the deviations from WAC reported here are universal or not.

We understand that a replacement of the WAC and the application in atmospheric models should not be suggested, as it is also discussed in the response to major comment 4 of referee #1. We changed the related sections in the manuscript, as follows:

Abstract:

Based on **the** experimental results **of our direct measurements**, we suggest a new fit line **to formulate**  the onset conditions of homogeneous freezing of sulfuric acid aerosol particles **as an isoline for nucleation rate coefficients between  5 ·10⁸ cm⁻³s⁻¹ and 10¹³ cm⁻³s⁻¹**.  **The potential significant impacts of the higher** homogeneous freezing thresholds, **as directly observed in the AIDA experiments under simulated cirrus formation conditions,**  on the **model** prediction of cirrus cloud occurrence and related cloud radiative effects **are discussed**.

Results and Discussion:

Based on the AIDA results and the discussion above, we  **provide a new fit line** for homogeneous freezing **of H₂SO₄/H₂O aerosol particles directly measured as function of** $S_{ice}$ and T.

 (The section about the constrained fit was completely removed.)

 **Applying** **t**his **new freezing threshold for cloud formation processes in the atmosphere,**  the homogeneous freezing onset in cirrus formation **would be shifted** to ice saturation ratios of about 2.0 at temperatures around 185 K. A higher homogeneous freezing onset may explain the high ice saturation ratios **occasionally** reported for low temperatures in the upper troposphere in some field studies (Jensen et al., 2005; Lawson et al., 2008; Krämer et al., 2009; Krämer et al., 2020). **However, it needs to be considered that the fit line only describes homogeneous freezing of H₂SO₄/H₂O aerosol particles under laboratory conditions. Other aerosol species,**

**which could be relevant for homogeneous freezing processes in the atmosphere, are not taken into account. Application to atmospheric conditions therefore needs to be done with caution.**

Consequently, a precise description of homogeneous freezing processes is crucial to understand cloud radiative effects in the present climate as well as in predictions of climate change.  This may in particular be relevant for cirrus clouds in the cold tropical tropopause layer
(TTL). (This section was moved from Section "4. Conclusions" to Section "3. Results and Discussion")

Conclusions:

A higher homogeneous freezing onset as derived from our experiments may also explain field observations of high clear-sky supersaturation, which should not occur according to the freezing thresholds predicted by Koop2000 (Jensen et al.,2005; Lawson et al., 2008; Krämer et al., 2009; Krämer et al., 2020). **However, the discussed high freezing onsets only consider homogeneous freezing of $H_2SO_4/H_2O$ aerosol particles under laboratory conditions, without involving other atmospherically relevant aerosol species.**  **A**n empirical fit to the AIDA results with the form $\ln(S_{ice}) = a + 1/T \cdot b$ with fit parameters a =  **−1.40±0.05** and b =  **390±10** K **describes the observed homogeneous freezing onsets of $H_2SO_4/H_2O$ aerosol particles in the chamber**. **This fit line is based on direct measurements of the freezing onset conditions at simulated cirrus formation conditions and provides an isoline for homogeneous freezing of sulfuric acid aerosol particles for nucleation rate coefficients between $5 \cdot 10^8$ $cm^{-3}s^{-1}$ and $10^{13}$ $cm^{-3}s^{-1}$.**  **The application of this fit line to atmospheric conditions requires further work on the physical behavior of $H_2SO_4/H_2O$ aerosol particles at low temperatures and on the involvement of other atmospheric aerosol particle types.** Ongoing experiments in the AIDA cloud simulation chamber aim at investigating homogeneous freezing onsets of different solutes and at constraining the descriptions for liquid water saturation pressures to experimental results.

**Minor comments:**

Figure 4: Could the uncertainties in predictions from Koop et al. (2000) be added in this figure for clarity.

In Koop, 2004, the uncertainty range for the freezing line in the $a_w$-T-space is given with 5% uncertainty in the water activity. We included this range of uncertainty in Figs. 4 and 6, as a shaded area around the WAC freezing curves.

L6: WAC not a function of size of the aqueous aerosol particles. Please correct.

We changed the sentence in l. 6 as follows:
The WAC describes the homogeneous nucleation rate coefficients only as a function of the water activity,  which makes this approach well applicable in numerical models.

L255: Please mention/reference the specific E-AIM models used here and for evaluating the data from other studies shown in Fig. 1

We used the Model I. We added the correct references to this line, as suggested on the E-AIM website:
In cases where water activities or relative humidities were not given by the authors, we used the Extended AIM Aerosol Thermodynamics Model (E-AIM), **Inorganic Model I** on http://www.aim.env.uea.ac.uk/aim/aim.php **(Clegg et al., 1992; Carslaw et al., 1995; Massucci et al., 1999; Wexler and Clegg, 2002; Clegg and Brimblecombe, 2005)** to transfer given $H_2SO_4$ concentrations into water activities, which we assume to be equal to relative humidity.

**Technical comments:**

L8: "...laboratory-based homogeneous..."
Corrected.

L11: "Aqueous sulfuric acid aerosol particles of high purity were generated by..."
Corrected.

L47: "...to be equal to relative humilities."
Corrected.

L80: "...and the potential deviation from..."
Corrected.

L107: "...pump starts..."
Corrected.

L122: a period missing at the end of the sentence
Corrected.

Fig3. Caption: "...AIDA chamber as a function of time since pump start ..."
Corrected.

Fig6. Caption: "The fit shown in panel (a) is constrained..."
This sentence was removed completely due to other referee comments.

L353-54: "...freezing onset may explain the high ice saturation ratios..."
Corrected.

---

## Author Comment (AC3)

Referee comment on "High Homogeneous Freezing Onsets of Sulfuric Acid Aerosol at Cirrus Temperatures" by Julia Schneider et al., Atmos. Chem. Phys. Discuss., https://doi.org/10.5194/acp-2021-319-RC3, 2021

We thank referee #3 for his or her thoughtful and detailed comments and feedback. Please find below our responses and suggestions for the manuscript revision, with the referee comments in black, our answers in green, and suggested changes or additions to the manuscript in blue.

In this work the authors investigate the homogenous ice nucleation of aqueous solution droplets in a cloud chamber. The authors carry out a literature review on the freezing thresholds reported for sulfuric acid solutions. They also collect results from prior experiments and perform new ones using the same experimental setup. Their experiments show that at low temperatures there are large deviations in the measured homogeneous freezing thresholds, when compared against a widely used parameterization. Based on this they propose a new parameterization to be used in atmospheric models. Homogeneous ice nucleation is still far from being completely understood, especially at low temperature where it can impact the formation of polar cirrus. This work is relevant to the scientific community. The experiments use established techniques and present interesting results. On the other hand, the analysis of the results is overly simplistic omitting several important factors. The usefulness of the derived parameterization is not clear. These issues should be addressed before the work could be suitable for publication in ACP.

**General Comments:**

Two other reviewers have already made comprehensive comments pointing out some of the major issues of the paper. Hence, I would emphasize some points that may require further discussion.

- The authors should include their data points in Figure 1 with their estimates of aw, or if it turns out too busy, make a separate figure depicting Tf vs aw. This would allow an easier comparison against the Koop et al (2000) results.

[Figure]

Fig. 1: Water activities of the AIDA results of this study in comparison to the summary of homogeneous freezing experiments shown in Fig. 1 in the manuscript.

This is a good suggestion. In the above Fig. 1, we added the AIDA sulfuric acid data to Fig. 1 of the manuscript. For the calculation of $a_w$, we assumed equilibrium conditions and therefore $a_w = RH_w$. The ice onset ice saturation ratios $S_{ice}$ were then transferred to $RH_w$ using the parameterization of saturation pressures for ice and liquid water given by Murphy and Koop, 2005. As this plot provides a supplementary comparison of our measurements in another parameter space, but is not crucial for the discussion, we decided to put this figure in the appendix and added the following figure description:

**Figure C1. Summary of homogeneous freezing measurements of $H_2SO_4/H_2O$ solutions in the $a_w$-T-space. The homogeneous freezing onsets of sulfuric acid solution samples shown in Fig. 1 are complemented by the AIDA results from this study (red points). The results in the $S_{ice}$-T-space were converted into the $a_w$-T-space by assuming equilibrium conditions $a_w=RH_w$ and by using the parameterizations for the water vapor saturation pressures with respect to ice and supercooled liquid water given by Murphy and Koop (2005). The uncertainties of the calculated $a_w$ values vary between ±0.036 and ±0.076.**

We referred to this figure in the Section "3. Results and discussion", as follows:

With decreasing temperature, the AIDA measurements show an increasing deviation from Koop2000 towards higher ice saturation ratios.
**A comparison of the AIDA results to the homogeneous freezing onsets from previous studies in the $a_w$-T-space shown in Fig. 1 can be found in Fig. C1.**

- There is a lot of uncertainty in the estimation of aw at low T. Slow droplet growth is not a sufficient condition to rule out kinetic limitations since the formation of glasses and hydrates is possible. The E-AIM model does not account for curvature effects which may bias the aw estimate as well.

We added the following sentence to the introduction, where the E-AIM model is mentioned the first time. We also added the Koop, 2004 reference, where this kind of uncertainties are further described:

**It needs to be considered that for the low temperature range, the model predictions are based on extrapolations of physiochemical properties towards low temperatures, which remain uncertain (Koop, 2004).**

We also added this reference to the Section "3. Results and Discussion":

The descriptions for the liquid water saturation pressures are rather uncertain **(Koop, 2004)**, and existing parameterizations deviate from each other (e.g. Buck, 1981; Sonntag, 1994; Tabazadeh et al., 1997; Murphy and Koop, 2005; Nachbar et al., 2019).

We fully agree that the uncertainty of the determination of the water activity at low temperature is an important issue. Our study shows that the conversion of data from different experimental set-ups like the AIDA chamber to compare with the WAC brings uncertainties into the description of homogeneous freezing. For this conversion, calculations based on parameterizations for e.g. water activities of different solutes and the water vapor saturation pressure with respect to supercooled liquid water are needed, which are uncertain especially at low temperatures. More details on this discussion are given in the responses to the major comments 1 and 3 of referee #1.

- The single experiment shown ruling out kinetic limitations is performed at 197 K hence does not target the conditions where the discrepancy with Koop2000 is the largest. It is my feeling that all experiments should have been conducted allowing for equilibration time in the same way as depicted in Fig. 5.

Please note that the experiment shown in Fig. 5 was started at 197 K, but the actual ice onset temperature was observed at a lower temperature of about 193 K with a corresponding ice saturation ratio of about 1.95. Therefore, this experiment was done in the temperature regime, where the observed deviation in onset supersaturation is already significant and we consider it as representative for the other experiments with significant deviation. We are planning more AIDA homogeneous freezing experiments with other solutes and will take up your suggestion to carry out more experiments with variable pumps speeds or rate of pressure change.

- One assumption of Koop2000 is that aw is independent of temperature. Deviation from this behavior would be enough to explain the discrepancy against the results of this work.

Only for some of the experiments shown in Koop et al. (2000), the water activity of the solution aerosols was determined by assuming that $a_w$ is independent of the temperature. In Fig. 1b, Koop et al. (2000) distinguish between freezing points, for which the water activity was derived directly by an ion interaction model at the actual freezing temperature (filled circles), and for which the water activity was estimated by equalling $a_w$ to that measured at the melting temperature (open circles), thus assuming that $a_w$ is independent of temperature between the melting and freezing temperature. All data points are well represented by the shown fit line, independent of the method by which the water activity was determined. We therefore cannot state that the temperature dependence of $a_w$ is generally not taken into account, but, as outlined above, have pointed to the overall uncertainty of the E-AIM in estimating $a_w$ at low T in the revised manuscript version.

- What would happen with the sulfuric acid upon freezing in highly concentrated droplets? This is significant since a fundamental assumption of the equilibrium approach to ice nucleation is that aw=1 in ice. Does the acid remain at the center of the droplet or does it get pushed to the surface (as another review notes, there is some evidence for the later)? If it is incorporated in the ice lattice, then the equilibrium calculations must be corrected accordingly.

Unfortunately, we cannot derive from our measurements what exactly happens to the acid when the ice forms and how it is distributed in or around the ice lattice. There are observations and theories claiming that the remaining highly concentrated sulfuric acid completely covers the ice crystal during the initial growth of the critical ice nucleus, which may decelerate ice crystal growth and could therefore explain the observed high freezing onset supersaturations (e.g. Bogdan et al., 2010). This way of kinetic limitation is also discussed in the manuscript in Section "3. Results and discussion". This discussion was improved and extended with more details, please see our response to major comment 3 of referee#1 for the changes and additions to the manuscript.

- The authors assert that their new correlation should be used instead of the Koop2000 However homogeneous ice nucleation does not admit a "singular" description, and the definition of a freezing threshold is of limited use for atmospheric modeling. Enough data is available in the experiments to calculate the nucleation rate. This would be more meaningful and useful. It would also refute/corroborate the Koop2000 approach of estimating the nucleation rate of solutions at low T based on that of pure water at much higher T. Does the Koop2000 parameterization for J match the measured nucleation rate? This would be a much better test of the Koop2000 hypothesis than the mere calculation of the freezing threshold.

We understand that the application of the new fit line for the sulfuric acid AIDA data in atmospheric models should not be used suggested. We therefore removed any statement suggesting this and added some notes of caution when applying the fit line to atmospheric conditions. For details on the changes in the manuscript, please see the responses to referees #1 (major comment 4) and #2 (major comment 4).

Our experiments at the AIDA chamber are not designed to explicitly calculate nucleation rates with a high accuracy, which would be needed to describe the dependence of water activity on the nucleation rate coefficient, as it is done in Koop et al. (2000). Möhler et al. (2003) give an estimate of homogeneous freezing nucleation rates to which the determination of the freezing onset in AIDA experiment is sensitive. The article also discusses the uncertainties of this estimation, including the estimation of the freezing probability, which is prone to uncertainties of the ice crystal number determination, the estimation of the volume of particles which freeze, and the uncertainty of the time determined for the occurrence of the freezing onset. Therefore, we can only provide an upper and lower limit of nucleation rates to which the AIDA measurements are sensitive to. Based on this estimation in Möhler et al. (2003), we compared our AIDA results to the WAC-lines for the given range of nucleation rates. Due to the mentioned uncertainties, a more accurate determination of nucleation rates is not possible with our experimental set-up.

**Technical comments**

Line 13. Please spell out "several thousand"
Corrected.

Line 31. Organic monolayers also promote freezing, not only solid ice-nucleating particles.
We changed the sentence as follows:
In regions with low concentrations of  ice-nucleating particles, [...].

Line 50. This is probably backwards. Higher aw leads to higher J in the Koop2000 model.
Corrected as follows:
Also shown are homogeneous freezing onset lines calculated according to Koop2000 for nucleation rate coefficients of $J_V = 10^{13}$ cm$^{-3}$s$^{-1}$ ($\Delta a_w = 0.3$**2**, dashed line) and $J_V = 5 \cdot 10^8$ cm$^{-3}$s$^{-1}$ ($\Delta a_w = 0.3$, dotted line) (Möhler et al., 2003).

Line 129. At least for these experiments the FTIR data should give some insight on whether the equilibrium assumption is correct. Does the aw calculated with the E-AIM model using the measured mass concentration match the relative humidity in the cloud chamber?
An accurate analysis of the FTIR data would in principle even be much more insightful, as it could provide the composition of the solution droplets (wt% $H_2SO_4$) upon freezing, i.e. the same quantity as inferred from the freezing experiments by Koop et al. (1998) with deposited sulfuric acid solution droplets. However, due to the lack of accurate infrared optical constants of dilute sulfuric acid solution droplets at such very low temperatures, this analysis is affected by a large uncertainty. It can be seen in the compilation of Fig. 1 in the manuscript that previous studies that have employed infrared spectroscopy to analyze the freezing of $H_2SO_4/H_2O$ aerosol particles (e.g. Bertram et al., 1996 and Clapp et al., 1997) show large deviations from the WAC line, probably due to uncertainties in the determination of the composition of the solution droplets at the freezing onset (Koop et al., 1998). Since 1997, some new low-temperature refractive index data sets for $H_2SO_4/H_2O$ have been published, and we have used them in Wagner et al. (2008) to develop an approach to quantitatively analyze the dilution of the sulfuric acid solution droplets during expansion cooling in the AIDA chamber and determine their composition prior to freezing. However, these new data sets still do not fully capture the range of temperatures and compositions covered by our expansion cooling experiments, so that the analysis remains highly uncertain. Generally, the $a_w$ calculated with the E-AIM model using the droplet composition estimated from the FTIR measurements is 20-30% smaller than the measured relative humidity. But this might again be due to uncertainties in the E-AIM calculations and the saturation water vapor pressure parameterization and cannot be taken as an argument that the equilibrium condition is incorrect.

Line 137. Typo in "aerosol".
Corrected.

Line 165. How important is the Kelvin effect overall? The E-AIM model does not account for it.
For our AIDA experiments, we assume that the Kelvin effect is not contributing to the observed freezing behavior, as the majority of the injected aerosol particles generally were larger than 100 nm in diameter. From the mentioned process modelling studies (Haag et al., 2003 and Wagner et al., 2008), we learned that smaller particles, which are affected by the Kelvin effect, do not maintain the same equilibrium compositions as the larger particles in the aerosol population. Consequently, their freezing behavior should be affected if the water activity is crucial for the homogeneous freezing onsets.

Line 176. Is there an induction time between the onset of ice and the observation a rozen droplet? In other words, how long does it take for ice to propagate inside a droplet?
In general, the diffusive growth of an ice particle is very fast. Of course, it is important to show that this is also the case for low temperature experiments. In the response to major comment 3 of referee #1, we calculated the diffusion length for a water molecule on a time scale of 6s. This diffusion length is significantly larger than the mean particle

diameter in all the experiments. This indicates that the diffusion of water in the aerosol and therefore the growth of the initial ice nucleus is fast. An additional indication that the ice growth is very fast also at low temperatures is the experiment we show in Fig. 5 in the manuscript. Here, we can see that in the minutes of constant but high relative humidity, no ice is detected. This would be the case, if the ice onset already happened at or below this relative humidity, but the detection of the formed ice particles would be delayed due to a significant induction time. In addition, the relative increase in the activated fraction and depolarization (third panel in Fig. 5) is strong and comparable to the increase observed in the experiments at higher temperatures. Regarding our OPCs, their detection range starts at about 0.7 μm. As discussed in Järvinen et al. (2014) the OPCs overestimate the size of aspherical ice particles by a factor of about 2.2. This means that an initial ice embryo needs to grow to an ice particle with a diameter of about 0.32 μm. In our uncertainty analysis, we considered the change in relative humidity and temperature in a specific time interval. Based on the diffusion lengths estimated above, this time interval is expected to cover the time the ice crystal needs to grow in the detection range of the OPC. This is also supported by the observation that the ice onsets determined by the OPCs agree well with the onsets determined by the light scattering measurements of the SIMONE instrument (see Fig. A1 in the manuscript). Therefore, a potential induction time is not expected to change the determined ice onsets in the given uncertainty range.

We added a statement on the potential induction time to Section "2.4 Analysis of uncertainties":

We determined an uncertainty of ±3 s for the onset time. **Based on estimates for the diffusion length of water molecules in the $H_2SO_4/H_2O$ aerosol particles at low temperature, which are discussed in greater detail in Sect. 3, we assume that this uncertainty covers the time the initial ice nucleus needs to grow into the detection range of the OPCs.** The variability of the gas temperature T in this 6s time interval is a first factor contributing to the uncertainty of $T_{ice}$.

Line 203. Please define the freezing probability.
We added the following to the text in the manuscript:
These nucleation rate coefficients were determined for sulfuric acid aerosol particles with diameters between 0.5 and 2 μm with an estimated freezing probability between 0.02 and 0.5. **For the definition of the freezing probability, see Eq. 2 in Möhler et al. (2003).**

Line 204. Given that the whole nucleation "pulse" lasts about 100 s this time delay could be significant.
In the AIDA experiments, we do not have a kind of "nucleation pulse". Temperature is decreasing during the experiment run and the relative humidity is increasing accordingly, until a specific nucleation rate is reached, which causes the observed ice onset in the chamber. This ice onset is characterized by the current temperature and relative humidity inside the chamber. At lower temperatures, the relative humidity changes more slowly over time, so that the uncertainty of ice onset relative humidity is reduced.

Line 252. Please spell out "several thousand" or rephrase it.
Done.

Line 272. This is a key line. Do the authors suggest that such is not the case? If aw-awi is not constant, then the Koop2000 approach is not valid at these conditions. If this is what the authors meant, please spell it out.
No, we do not want to suggest that it is not the case, that the nucleation rate coefficients can be described only in dependence on water activity and temperature. As already stated in our response to referee #1, we do not want to suggest that the WAC and the

underlying assumptions are not correct. Instead, we want to show that the conversion of data from different experimental set-ups like the AIDA chamber to compare with the WAC brings uncertainties into the description of homogeneous freezing. For this conversion, calculations based on parameterizations for e.g. the water vapor saturation pressure with respect to supercooled liquid water are needed, which are uncertain especially at low temperatures.

Line 288. This is a large uncertainty. What is the equivalent in aw space?
The uncertainties of $S_{ice}$ converted into the $a_w$-space gives uncertainties of $a_w$ between ±0.036 and ±0.076. This is also plotted in Fig. C1, which has been added to the manuscript appendix, and mentioned in the figure description.

Line 316-320. I don't see how these results indicate no internal kinetic limitations or that there is no induction time. In fact, in Line 204 it was indicated that it could be as much as 10 s. All that the experiment does is to show that droplet growth is slow (which could be due to glass formation) before nucleation, not that the equilibration time scale is much smaller than the nucleation time scale. Please explain.

We added a more detailed discussion on the kinetic limitations and phase state of the sulfuric acid aerosol particles to Section "3. Results and discussion". Please see response to major comment 3 of referee#1.
We also added a more detailed description of the mentioned experiment shown in Fig. 5 in the manuscript including a new additional panel showing a direct comparison of the relative humidity and the forward scattering intensity measured by SIMONE. We do not agree that the experiment shows that the droplet growth is slow. Rather, the additional panel emphasizes that the particles react instantaneously to the changes in relative humidity. As soon as the pumping speed is reduced and the relative humidity is controlled for a duration of about 300 s to a constant value clearly above the Koop et al. (2000) line, there is no further increase of the forward scattering intensity, which could be ascribed to a delayed water uptake. Rather, the scattering intensities instantly follow the smooth variations in the relative humidity. This observation (which is now much better described in the revised manuscript version), together with the estimates for the diffusion length of the water molecules in the H2SO4/H2O aerosol particles, indicates that kinetic limitations of the particles growth due to water uptake are not expected during the experiment.

Line 321-323. How does the aw calculated with the measured mass concentration compare against the relative humidity? Does the equilibrium assumption hold?
Please see the response to the comment to line 129 above.

Line 340. Freezing thresholds are not very useful for atmospheric modeling. The authors should report nucleation rates instead.
Please see the response to the last general comment about the difficulty of calculating nucleation rates with high accuracy from the AIDA experiments.

Line 345-355. This is confusing. Please just report the recommended correlation.
We decided to keep the unconstrained version of the fit and removed the constrained one from Fig. 6 and the corresponding text. For details on the changes made in the manuscript, please see the response to the minor comment 8 of referee#1.

Line 385. There is the implicit assumption that the nucleation rate is still a function of aw only, which seems to contradict the premise of this work.
At first, we decided to remove the constrained fit, which is described in this line, and to only keep the unconstrained one, as it was suggested to only keep one of the fit

versions. However, in general, the assumption that the nucleation rate coefficient is still a function of $\Delta a_w$ as suggested by Koop et al. (2000), does not contradict the conclusion and discussion of this study. Rather, we show that we observe deviation from the WAC-derived homogeneous freezing lines, but relate this deviation to the conversion from different parameter spaces into each other, and not to an invalidity of the ideas behind the WAC. See also our answer to line 272 above.

Line 392. There is not enough data here to assert this. Koop2000 also parameterizes nucleation rates which is much more useful and completely omitted in this work.
We understand that it should not be suggested to replace the well-established Koop2000 freezing lines. We therefore adjusted this section in the manuscript as follows:

Consequently, a precise description of homogeneous freezing processes is crucial to understand cloud radiative effects in the present climate as well as in predictions of climate change.  This may in particular be relevant for cirrus clouds in the cold tropical tropopause layer
(TTL). (This section was moved from Section "4. Conclusions" to Section "3. Results and discussion")

**References:**

Bertram, A. K., Patterson, D. D., and Sloan, J. J.: Mechanisms and temperatures for the freezing of sulfuric acid aerosols measured by FTIR extinction spectroscopy, J. Phys. Chem., 100, 2376–2383, https://doi.org/10.1021/jp952551v, 1996.

Bogdan, A. and Molina, M. J.: Aqueous aerosol may build up an elevated upper tropospheric ice supersaturation and form mixed-phase particles after freezing, J. Phys. Chem. A, 114, 2821–2829, https://doi.org/10.1021/jp9086656, 2010.

Buck, A. L.: New equations for computing vapour pressure and enhancement factor, J. Appl. Meteorol., 20, 1527–1532, https://doi.org/10.1175/1520-0450(1981)020<1527:nefcvp>2.0.co;2, 1981.

Clapp, M. L., Niedziela, R. F., Richwine, L. J., Dransfield, T., Miller, R. E., and Worsnop, D. R.: Infrared spectroscopy of sulfuric acid/water aerosols: Freezing characteristics, J. Geophys. Res. Atmos., 102, 8899–8907, https://doi.org/10.1029/97jd00012, 1997.

Haag, W., Kärcher, B., Schaefers, S., Stetzer, O., Möhler, O., Schurath, U., Krämer, M., and Schiller, C.: Numerical simulations of homogeneous freezing processes in the aerosol chamber AIDA, Atmos. Chem. Phys., 3, 195–210, https://doi.org/10.5194/acp-3-195-2003, 2003.

Järvinen, E., Vochezer, P., Möhler, O., and Schnaiter, M.: Laboratory study of microphysical and scattering properties of corona-producing cirrus clouds, Appl. Opt. 53, 7566-7575, https://doi.org/10.1364/AO.53.007566, 2014.

Koop, T., Ng, H. P., Molina, L. T., and Molina, M. J.: A new optical technique to study aerosol phase transitions: The nucleation of ice from $H_2SO_4$ aerosols, J. Phys. Chem. A, 102, 8924–8931, https://doi.org/10.1021/jp9828078, 1998.

Koop, T., Luo, B., Tsias, A., and Peter, T.: Water activity as the determinant for homogeneous ice nucleation in aqueous solutions, Nature, 406, 611–614, https://doi.org/10.1038/35020537, 2000.

Koop,T.: Homogeneous ice nucleation in water and aqueous solutions, Z. Phys. Chem., 218,1231–1258, https://doi.org/10.1524/zpch.218.11.1231.50812, 2004.

Mangold, A., Wagner, R., Saathoff, H., Schurath, U., Giesemann, C., Ebert, V., Krämer, M., and Möhler, O.: Experimental investigation of ice nucleation by different types of aerosols in the aerosol chamber AIDA: Implications to microphysics of cirrus clouds, Meteorol. Zeitschrift, 14, 485–497, https://doi.org/10.1127/0941-2948/2005/0053, 2005.

Möhler, O., Stetzer, O., Schaefers, S., Linke, C., Schnaiter, M., Tiede, R., Saathoff, H., Krämer, M., Mangold, A., Budz, P., Zink, P., Schreiner, J., Mauersberger, K., Haag, W., Kärcher, B., and Schurath, U.: Experimental investigation of homogeneous freezing of sulphuric acid particles in the aerosol chamber AIDA, Atmos. Chem. Phys., 3, 211–223, https://doi.org/10.5194/acp-3-211-2003, 2003.

Murphy, D. M. and Koop, T.: Review of the vapour pressures of ice and supercooled water for atmospheric applications, Q. J. R. Meteorol. Soc., 131, 1539–1565, https://doi.org/10.1256/qj.04.94, 2005.

Nachbar, M., Duft, D., and Leisner, T.: The vapor pressure of liquid and solid water phases at conditions relevant to the atmosphere, J. Chem. Phys., 151, 064 504, https://doi.org/10.1063/1.5100364, 2019.

Sonntag, D.: Advancements in the field of hygrometry, Meteorol. Zeitschrift, 3, 51–66, https://doi.org/10.1127/metz/3/1994/51, 1994.

Tabazadeh, A., Toon, O. B., Clegg, S. L., and Hamill, P.: A new parameterization of H2SO4/H2O aerosol composition: Atmospheric implications, Geophys. Res. Lett., 24, 1931–1934, https://doi.org/10.1029/97GL01879, 1997.

Wagner, R., Benz, S., Bunz, H., Möhler, O., Saathoff, H., Schnaiter, M., Leisner, T., and Ebert, V.: Infrared optical constants of highly diluted sulfuric acid solution droplets at cirrus temperatures, J. Phys. Chem. A, 112, 11 661–11 676, https://doi.org/10.1021/jp8066102, 2008.

---

## Author Response (AR1)

**Review for "High Homogeneous Freezing Onsets of Sulfuric Acid Aerosol at Cirrus Temperatures" by Schneider et al.**

We thank referee #1 for his or her thoughtful and detailed comments and feedback. Please find below our responses and suggestions for the manuscript revision, with the referee comments in black, our answers in green, and suggested changes or additions to the manuscript in blue.

The manuscript submitted to Atmospheric Chemistry and Physics titled "High Homogeneous Freezing Onsets of Sulfuric Acid Aerosol at Cirrus Temperatures" by Schneider et al. presents new and reanalyzed data on homogeneous freezing of aqueous sulfuric acid aerosol in the AIDA chamber. Homogeneous ice nucleation studies at such large supercoolings is certainly relevant topic and important for atmospheric science. The authors report the onset of ice nucleation to be below water saturation and follow the water activity criterion (WAC) from Koop et al.[1]. However, their results deviate from Koop et al.[1] at 185-205 K. After a thorough uncertainty analysis and clearly stating their assumptions, they conclude that this deviation is significant. They claim that the deviation may be because water saturation curves with respect to ice and water are uncertain and suggest that the estimate by Nachbar et al.[2] instead of Murphy and Koop[3] would cause deviations to decrease. Finally, the authors make a claim about the consequence of their results to ambient cirrus clouds. Overall, the intro, methods and results of this manuscript are well written. The methods are described well and the error analysis is sound. However the discussion needs great improvement. There are major comments that cast the authors conclusions in serious doubt. These have to do with the lack of discussion of the physical evidence for the WAC, the uncertainty for the WAC, the mixing time of the particles, and finally, their suggestion of treating homogeneous freezing for cirrus clouds as only from sulfuric acid. A few minor comments exist. Overall, I cannot recommend publication at this time without significant revision.

Major comments

1. There is a lack of any physical reasoning. The WAC is not an empirical parametrization of aqueous sulfuric acid onset freezing temperatures. It is a physical description of freezing of a variety of solutes at ambient pressure, and of pure water at high pressure[1,4]. In Koop[4], physical evidence is presented that freezing and melting temperatures of pure water at high pressure and highly concentrated aqueous solution at ambient pressure are similar and are the result of similar affects
on the water hydrogen bonding structure. To be fair and balanced, if the authors claim their data deviates from the WAC, then they must claim a physical reason for this and independent evidence to support their reasoning. For example, if WAC is solute independent (l. 38-39), why do they suspect sulfuric acid is such a special case? Does the hydrogen-hydrogen radial distribution function[4] of sulfuric acid aqueous solutions deviate from high-pressure water at the same water activity? Does it deviate from other solutes below 205 K at the same water activity, but not deviate at warmer temperatures? If they cannot explain their results physically or come up with a realistic quantitative measure, it is acceptable that the authors include a statement that they do not know a physical reason why such a deviation would occur.

First of all, we would like to state that we present here experimental results after careful discussion and uncertainty analysis, with direct measurements of the temperature and ice saturation ratio at the point of homogeneous freezing onset for aqueous sulfuric acid aerosol particles. That our data deviates from those derived from the WAC-based parameterization by Koop et al. (2000) is an experimental result, and does not require any physical reasoning. Physical reasoning is important and may be helpful when discussing possible reasons for this disagreement, and some are included in the discussion in section "3. Results and discussion", but unfortunately we do not yet have a definite explanation for the observed deviation.

Please note that it is not the aim of this paper to question the WAC itself and its physical explanation, but our data provide strong evidence that there is an increasing high bias of the homogeneous freezing onset to lower temperatures, at least for aqueous sulfuric acid aerosol we used in our experiments.

Indeed, we compare the freezing experiments of a single solute, $H_2SO_4$, with a freezing line established for a variety of different solutes. However, the results of a previous study by Koop et al., 1998 also focusing on the homogeneous freezing of sulfuric acid aerosol particles shows a good agreement with the WAC-based lines with an universal validity independent on the nature of the solute published later in Koop et al., 2000. We therefore also compare the AIDA sulfuric acid measurements to the WAC-based lines. In summary, with our measurements, we cannot and do not want to show that the WAC and the underlying assumptions are not correct. Instead, we want to show that the conversion of data from different experimental setups like the AIDA chamber to compare with the WAC brings uncertainties into the description of homogeneous freezing. For this conversion, calculations based on parameterizations for e.g. the water vapor saturation pressure with respect to supercooled liquid water are needed, which are uncertain especially at low temperatures.

In order to focus the discussion more on the observed disagreement with the WAC-based homogeneous freezing line, and to make it more clear that we do not doubt the WAC for homogeneous freezing in general, we add the following statement to "3. Results and Discussion":

**Finally, we do not have a solid explanation for the deviation of the $H_2SO_4/H_2O$ homogeneous freezing thresholds observed in AIDA experiments from the WAC-based homogeneous freezing lines. This deviation may be related to uncertainties in the formulation of physicochemical properties at low temperatures, which are required for the conversion between the Koop2000 parameter space ($a_w$-T-space) and the AIDA parameter space ($S_{ice}$-T-space), as described above.**

2. There is uncertainty of the WAC that should be included. I appreciate the authors experimental uncertainty analysis, however, they lack the uncertainty analysis for the WAC. They must include the uncertainty in the WAC lines for a fair comparison. Koop[4] claims a uncertainty up to 5% in temperature from the freezing line, which translates to an error close to ±15 K in temperature and ±0.08 in water activity, $a_w$, at homogeneous freezing temperatures of 185 K. Please check this. In order for a fair comparison with WAC, the

authors must calculate errors on the WAC lines at all temperatures and show them in their figures.

Koop (2004) gives an uncertainty range of +/- 5 % for the water activity in the freezing line in the $a_w$-T-space. We included this range of uncertainty in Fig. 4 and 6, as a shaded area around the WAC freezing curves.

3. Mixing time of high concentrated aqueous solutions at low temperature. On l. 294, the authors claim their assumption that particles are well-mixed and in equilibrium with their humidified environment. Support for this assumption is given[5] for temperatures > 205 K and for the experiment performed by the authors at 194 K in Fig. 6. However, there remains some doubt about this assumption, and the particles may be highly viscous to the point of limiting mixing within them due to slow molecular transport. Whether a particle is or is not well-mixed can depend on the entire relative humidity history, even before the start of experiments. If experiments began at a relative humidity with respect to ice, $RH_i$, and temperature in which aqueous aerosol particles were initially in a glassy state, it would take time for a glassy and inhomogeneously mixed particle to transition to mixed and satisfy the authors assumption. For example, Berkemeier et al.[6] has shown that for glassy organic aerosol, a humidity induced transition to well-mixed particles can take 1600 s starting at 215K and cooling to 212 K and consequently humidity increasing from 60% to 87%. Although glassy inorganic solutions may behave differently than glassy organics, experiments by the authors here were 3 to 4 times faster and therefore, a kinetic limitation cannot be ruled out. If the authors began their experiment in Fig. 6 at a lower $RH_i$ as they did for their experiments in Fig. 3, would kinetic limitations be observed? Evidence for a kinetic limitation comes from the sulfuric acid phase diagram[4]. When $RH_i$ = 95% and T = 185 K, the weight percent of sulfuric acid solution in equilibrium is roughly 50%[7[9] and this is exactly at the boundary of ultra-viscous solutions. The authors should include the starting $RH_i$ in the appendix tables.
The uncertainty here is large, of course, due to extrapolation and seen by the scatter in crystallization temperatures of glassy particles upon warming in Fig. 4 of Koop[4]. What I expect is that the authors include a value of viscosity, molecular diffusion coefficients, or mixing time scales at their exact experimental conditions measure in literature. The authors have not shown evidence for this assumption for temperatures down to 185 K. I do not know of existing viscosity or diffusion coefficient measurements in this temperature and humidity range. If they exist, what is the variability. I do not recommend extrapolating from common measurements. A lack of measurements would cast doubt on this assumption, and thus their conclusions.

We agree that the particle phase and the related mixing time at low temperatures are an important aspect to consider in the measurements of this study and that further and more detailed discussion on that is needed.
As a first step, we included the start relative humidity with respect to ice $RH_{i,0}$ and liquid water $RH_{w,0}$ to the tables in the appendix, as suggested.
To illustrate the start conditions and the sulfuric acid phase at the ice onset, we adapted Fig. 4 in Koop, 2004 and added the data points for the start composition of the sulfuric acid aerosol particles in wt% $H_2SO_4$ for our experiments and the data points for the composition

at ice onset (see Fig. 1 in this response, red and blue dots). The weight percentage composition was determined by using Model I of the E-AIM model with the measured temperature and relative humidity inside the chamber. As described in the referee's comment above, a glassy state of the sulfuric acid aerosol particles in our experiments would strongly influence the observed freezing process due to slow mixing processes. In Fig. 1, we can see that all experiments are above the conditions of ultra-viscous and glassy particles, according to the phase diagram in Fig. 4 in Koop, 2004 and the references therein. Especially the glass transition conditions given in this figure are at significantly lower temperatures compared to our starting and ice onset conditions. This is supported by our SIMONE measurements, which show an increasing signal in the forward scattering intensity between pump start and ice onset, for all the experiments. This increase shows that the aerosol particles are able to take up water and dilute. Only from this observation, it cannot be completely ruled out that an enhanced viscosity at low temperatures decelerates the water uptake, so that it might be not sufficient to maintain particles in thermodynamic equilibrium with the environment. For the investigation of a potential decelerated water uptake, we refer again to the experiment shown in Fig. 5 in the manuscript. In this experiment, the relative humidity was kept nearly constant above Koop2000 for about five minutes. Firstly, we observe no ice formation in this time period, which indicates that the AIDA ice onset is not higher than the Koop2000 line due to a delayed ice detection. Secondly, the forward scattering intensity of the SIMONE instrument (dark red line, third panel) shows a significant increase in the first two minutes of constant pumping, which is related to the water uptake of the particles. In the following five minutes of constant relative humidity (slower pumping), no further uptake of the particles is observed. To illustrate this more precisely, we added a second panel to Fig. 5 in the manuscript (see Fig. 2 in this response), which shows a direct comparison of the relative humidity and the forward scattering intensity on a smaller scale. In this panel, we clearly observe the particle diluting in the first two minutes of the experiment, which generally decreases the particles' viscosity. After that period, the particles could have continued taking up water, but no further increase in the forward scattering is observed, as soon as the pump speed is lowered. Rather, the forward scattering intensity follows nicely the slight variations in the relative humidity, but a decelerated or delayed water uptake is not indicated.

[Figure]

Fig 1: Phase diagram of sulfuric acid adapted from Koop (2004). The phase diagram of sulfuric acid as given by Koop (2004) and references therein is shown and complemented by the calculated weight percentage concentrations of the sulfuric acid aerosol particles before the start of the AIDA experiments of this study (red dots) and at the observed ice onset during the experiments (blue dots).

[Figure]

Fig. 2: Investigation of kinetic limitations with respect to water uptake in the AIDA chamber. Panel (a): The figure is composed in the same way as Fig. 3. For the experiment shown, started at about 197K, the pump rate was controlled in such a way that the relative humidity

with respect to ice stayed relatively constant for about 5 minutes at about 170%, hence above the homogeneous freezing threshold suggested by Koop et al. (2000). For details on this experiment, see Sect. 3. Panel (b): Enlarged view of the relative humidity and the forward scattering intensity at the beginning of the experiment.

To our knowledge, studies or measurements of the viscosity of sulfuric acid aerosol particles at very low temperatures are not available in the literature. Measurements and suggested fits for viscosity in dependence on temperature and $H_2SO_4$ weight percentage concentration by Williams and Long (1995) extend to 200 K and are given for any concentration between 30 and 80 wt%. Fig. 1 in Williams and Long (1995) shows that the viscosity is strongly increasing with decreasing temperature. As the AIDA experiments have weight percentage concentrations < 30% at the ice onset, we can only use the Williams and Long parameterization to determine the viscosity before experiment start when wt% > 30% and for experiments starting at temperatures > 200 K. In Fig. 3a in this response, we show the calculated viscosities $\eta$ using the Williams and Long parameterization for the AIDA experiments at experiment start (black dots) with start temperatures > 200 K. The viscosities for experiments starting at temperature < 200 K are shown in the grey shaded box, as they need to be treated with caution. In addition, we used these viscosities to determine the diffusion coefficient D of water molecules in the sulfuric acid solution aerosol particles using the Stokes-Einstein-equation (red triangles). With the diffusion coefficients, the diffusion length $D_L$ for a water molecule on a time scale of 6s is calculated and shown in Fig. 3b in comparison to the measured mean diameter of the $H_2SO_4/H_2O$ aerosol particle populations. The diffusion length is significantly larger than the mean particle diameter in all experiments, which indicates that the diffusion of water in the aerosol particles is fast enough to keep them in thermodynamic equilibrium with the environment. Note again that these calculations are only strictly valid for experiments starting at temperatures > 200 K and only for the experiment start conditions.

[Figure]

Fig 3: Viscosity and diffusion coefficients for the start conditions of the AIDA experiments. Panel (a): Viscosity (black dots) and diffusion coefficients (red triangles) calculated using the parameterization of Williams and Long (1995) and the Stokes-Einstein-equation. Panel (b): Comparison of the mean particle diameter of the sulfuric acid particle population (blue diamonds) to the diffusion length of water molecules (black squares) on a time scale of 6s. Note that the shown data points represent the starting conditions of the AIDA experiments. As the parameterization of Williams and Long (1995) is only valid for T > 200 K, the data points of experiments at lower temperatures are displayed in a grey shaded box.

We added a similar discussion to Section "3. Results and discussions" and added the Figs. 1 and 3 to the appendix, as additional information, and changed Fig. 5 in the manuscript by adding panel b (see Fig. 2 in this response). Changes in the manuscript are as follows:

[revised manuscript text omitted]

4. The authors want their fitted line in atmospheric models and replace the WAC (l. 391-392). Inherent in this is that only sulfuric acid aerosol particles nucleate homogeneously to form cirrus clouds in models, is to discard the presence of other solutes such as secondary organic aerosol; nitrates or sea salts, is to treat homogeneous freezing only at the authors' measured onset, and is to not account for homogeneous ice nucleation rate coefficients (as function of water activity and

temperature). As there is no given physical explanation for their data, this suggestion is a large leap backward for understanding atmospheric physics and chemistry. The authors certainly make a line go through data points, however it is not appropriate to use this line to predict the formation for cirrus clouds. Up scaling a purely empirical parametrization from the AIDA chamber to real atmospheric conditions is an extrapolation outside of their experimental conditions. If the authors want to replace Koop et al.[1], then more work needs to be done to quantify and understand the physics of homogeneous ice nucleation and apply that understanding to the range of temperature, water activity and nucleation rate coefficients valid for their measurements and consistent with the over 20 years of observation supporting the WAC. Please remove any mention of suggesting to use this parametrization in atmospheric models on l. 344, l. 351-352, l. 384-385 and l. 391-392 and in the last 2 sentences of the **abstract**. These are the instances I have found.

We understand that a replacement of the WAC and the application in atmospheric models should not be suggested. We changed the related sections in the manuscript, as follows:

Abstract:

Based on **the** experimental results **of our direct measurements**, we suggest a new fit line **to formulate**  the onset conditions of homogeneous freezing of sulfuric acid aerosol particles **as an isoline for nucleation rate coefficients between 5 ·10$^8$ cm$^{-3}$s$^{-1}$ and 10$^{13}$ cm$^{-3}$s$^{-1}$**.  **The potential significant impacts of the higher** homogeneous freezing thresholds, **as directly observed in the AIDA experiments under simulated cirrus formation conditions,**  on the **model** prediction of cirrus cloud occurrence and related cloud radiative effects **are discussed**.

Results and Discussion:

Based on the AIDA results and the discussion above, we  **provide a new fit line** for homogeneous freezing **of $H_2SO_4/H_2O$ aerosol particles directly measured as function of** $S_{ice}$ **and T.**

 (The section about the constrained fit was completely removed.)

 **Applying** **t**his **new freezing threshold for cloud formation processes in the atmosphere,**  the homogeneous freezing onset in cirrus formation **would be shifted** to ice saturation ratios of about 2.0 at temperatures around 185 K. A higher homogeneous freezing onset may explain the high ice saturation ratios **occasionally** reported for low temperatures in the upper troposphere in some field studies (Jensen et al., 2005; Lawson et al., 2008; Krämer et al., 2009; Krämer et al., 2020). **However, it needs to be considered that the fit line only describes homogeneous freezing of $H_2SO_4/H_2O$ aerosol particles under laboratory conditions. Other aerosol species, which could be relevant for homogeneous freezing processes in the atmosphere, are not taken into account. Application to atmospheric conditions therefore needs to be done with caution.**

Consequently, a precise description of homogeneous freezing processes is crucial to understand cloud radiative effects in the present climate as well as in predictions of climate change.  This may in particular be relevant for cirrus clouds in the cold tropical tropopause layer
(TTL). (This section was moved from Section "4. Conclusions" to Section "3. Results and Discussion")

Conclusions:

A higher homogeneous freezing onset as derived from our experiments may also explain field observations of high clear-sky supersaturation, which should not occur according to the freezing thresholds predicted by Koop2000 (Jensen et al.,2005; Lawson et al., 2008; Krämer et al., 2009; Krämer et al., 2020). **However, the discussed high freezing onsets only consider homogeneous freezing of $H_2SO_4/H_2O$ aerosol particles under laboratory conditions, without involving other atmospherically relevant aerosol species.**  **A**n empirical fit to the AIDA results with the form ln($S_{ice}$) = a+ 1/T·b with fit parameters a =  **−1.40±0.05** and b =  **390±10** K **describes the observed homogeneous freezing onsets of $H_2SO_4/H_2O$ aerosol particles in the chamber**. **This fit line is based on direct measurements of the freezing onset conditions at simulated cirrus formation conditions and provides an isoline for homogeneous freezing of sulfuric acid aerosol particles for nucleation rate coefficients between $5 \cdot 10^8$ cm$^{-3}$s$^{-1}$ and $10^{13}$ cm$^{-3}$s$^{-1}$.**  **The application of this fit line to atmospheric conditions requires further work on the physical behavior of $H_2SO_4/H_2O$**

**aerosol particles at low temperatures and on the involvement of other atmospheric aerosol particle types.** Ongoing experiments in the AIDA cloud simulation chamber aim at investigating homogeneous freezing onsets of different solutes and at constraining the descriptions for liquid water saturation pressures to experimental results.

Minor comments

1. l. 6. The WAC is not a function of aerosol particle size. Likewise, it is not a function of time either.

We changed the sentence in l. 6 to:
The WAC describes the homogeneous nucleation rate coefficients only as a function of the water activity,  which makes this approach well applicable in numerical models.

2. l. 23-24 and 386-407. The authors certainly review and discuss cirrus cloud formation and radiative effects, however, these are not conclusions. No cloud model or any calculations of radiative forcing were made here to support these statements. In the abstract I suggest the following rewrite or something similar, "Our results are discussed in the context of predicting the formation of cirrus clouds and related cloud radiative effects." In addition, these conclusions need to be moved to the results and discussion section.

We changed the last sentence of the abstract to:
 **The potential significant impacts of the higher** homogeneous freezing thresholds, **as directly observed in the AIDA experiments under simulated cirrus formation conditions,**  on the **model** prediction of cirrus cloud occurrence and related cloud radiative effects **are discussed**.

We further moved the section in l. 186-407 from "3. Conclusions" to "4. Results and Discussions".

3. l. 44-47. and 327-329. It was already stated by Koop[4] that thermodynamic models (to calculate water activity of solutions or saturation vapor pressures) extrapolated to these low temperatures can be large sources of errors. I suggest to add this reference here.

Thanks for bringing this reference to our attention. We added the following sentence after lines 44-47:
**It needs to be considered that for the low temperature range, the model predictions are based on extrapolations, which remain uncertain (Koop, 2004).**

We also added this reference to the other text passages, as suggested:
The descriptions for the liquid water saturation pressures are rather uncertain **(Koop, 2004)**, and  **existing** parameterizations deviate from each other (e.g. Buck, 1981; Sonntag, 1994; Tabazadeh et al., 1997; Murphy and Koop, 2005; Nachbar et al., 2019).

4. l. 46. What E-AIM model did the authors use? I suppose Model I7{9. Please check the correct references on the E-AIM website.

Indeed, we used the Model I. We added the correct references, as suggested on the E-AIM website:
In cases where water activities or relative humidities were not given by the authors, we used the Extended AIM Aerosol Thermodynamics Model (E-AIM), **Inorganic Model I** on http://www.aim.env.uea.ac.uk/aim/aim.php **(Clegg et al., 1992; Carslaw et al., 1995; Massucci et al., 1999; Wexler and Clegg, 2002; Clegg and Brimblecombe, 2005)** to transfer given $H_2SO_4$ concentrations into water activities, which we assume to be equal to relative humidity.

5. l. 50-51. I think there is a mistake here. Higher values of $\Delta a_w$ should yield higher values of $J_v$ .

That is right. We corrected this sentence accordingly:
Also shown are homogeneous freezing onset lines calculated according to Koop2000 for nucleation rate coefficients of $J_v = 10^{13}$ cm$^{-3}$s$^{-1}$ ($\Delta a_w = 0.3$**2**, dashed line) and $J_v = 5 \cdot 10^8$ cm$^{-3}$s$^{-1}$ ($\Delta a_w = 0.3$, dotted line) (Möhler et al., 2003).

6. l. 61-63. Would the authors take care to please check the ambient ice saturation ratios for these studies? The authors language gives the impression that high $RH_i$ at or above 200% happens all the time at temperatures colder than 200 K. This is misleading. It is directly stated in the abstract Krämer et al.10 that the highest $RH_i$ for clear sky is about 150%. Krämer et al.10 shows a distribution of $RH_i$ and there are very rarely any measurements at or above 200%. I count about 7 yellow squares in Fig. 7(e) of Krämer et al.10 at or above 200%, but practically all data is bounded by or scattered around homogeneous freezing. This statement misrepresents the findings
of Krämer et al.10, and I would encourage them to be more specific and representative of the previous research they are citing. Please check all citation here. Ambient in-cloud and clear-sky $RH_i$ > 150% occurs mostly < 2% of the time.

We checked the references for the atmospheric observations again and adjusted the text as follows to be more precise:

For T < 200 K, Koop2000 predicts homogeneous freezing thresholds ranging between ice saturation ratios of 1.6 and 1.7. However, **in an aircraft campaign in 2004,**  enhanced ice saturation ratios up to 2.**3 at about 187 K were observed in the upper troposphere (Jensen et al., 2005), clearly exceeding the Koop2000 freezing thresholds. Also in a later aircraft campaign in 2006, more than half of the relative humidity measurements showed values exceeding Koop2000 at upper tropospheric temperatures (Lawson et al., 2008). Other atmospheric observations at T < 200 K reported atmospheric relative humidities predominantly below Koop2000, but in a few cases the threshold was also exceeded (Krämer et al., 2009, Krämer et al., 2020). These atmospheric observations**

Krämer et al., 2009; Krämer et al., 2020) seemingly contradicting Koop2000 if the assumption holds that ice saturation ratios are unlikely to exceed the homogeneous freezing thresholds in the atmosphere.

7. l. 334-335. If the Δa$_w$ values would be used together with Nachbar et al.[2] to plot a new onset curve, would everything be within error bars? They claim that difference would be reduced, but why not show these differences and if they can completely explain the deviations they observe.

We added the homogeneous freezing thresholds according to Koop et al., 2000, but with using the parameterization of Nachbar et al., 2019 for the liquid water saturation pressure to the Figs. 4 and 6. We also included the related description and discussion in the text:

**This is shown in Fig. 4, where Koop2000 is shown in combination with the liquid water saturation pressure parameterization of Murphy and Koop (2005) (black dashed and dotted lines, MK2005) and additional in combination with the more recent line by Nachbar et al. (2019) (blue dashed and dotted lines, N2019). To fully explain the differences between the AIDA and the Koop2000 onsets, the liquid water saturation pressure would need to be even higher at low temperatures than suggested by Nachbar et al. (2019).**

Description of the new Fig. 4 in the manuscript:

Homogeneous freezing onsets of $H_2SO_4/H_2O$ aerosol particles. The freezing onset conditions, $T_{ice}$ and $S_{ice,fr}$, are displayed in comparison with the homogeneous freezing thresholds suggested by the WAC-based predictions by Koop et al. (2000) (dashed and dotted black lines) **(dashed and dotted lines) using two different parameterizations for the water saturation pressure with respect to supercooled liquid water from Murphy and Koop (2005) (MK2005, black) and Nachbar et al. (2019) (N2019, blue).** and the **The used** water saturation pressures with respect to supercooled liquid water according to **MK2005** Murphy and Koop (2005) (solid black line) and according to **N2019** Nachbar et al. (2019) (solid blue line) **are also shown**. The colors **of the measurement data points** represent the different AIDA campaigns in the corresponding years. The oldest campaigns are presented in reddish, whereas the more recent campaigns are shown in yellowish colors.

8. l. 341 and 350. The reason why the authors show 2 different fit parameters and procedures here is not clear. Would the authors please choose one, and remove the one you do not want your readers to use from the manuscript?

We decided to only keep the unconstraint fit, as this better represents our measurement data. We adjusted Fig. 6 and the related descriptions and discussions accordingly.

Description of Fig. 6:

**Figure 6**. New fit line for homogeneous freezing onsets of $H_2SO_4/H_2O$ aerosol particles. Panel (a): The freezing onsets determined by OPC and SIMONE data (red dots) are shown in an Arrhenius plot and fitted by an ordinary least square (OLS) fit with the form ln(Sice)

=a+1/T·b. The parameters are a=−1.40±0.05 and b= 390±10 K and the goodness of the fit is R² = 0.92. The shaded area is indicative for the uncertainty of the fit parameters. Panel (b): The OLS fit shown in panel (a) is transferred into the Sice-T-space and compared to Koop2000 and the water saturation lines suggested by Murphy and Koop (2005) and Nachbar et al. (2019).

Description and discussion in Section "3. Results and Discussion":

The coefficients of the fit shown in Fig. 6a are a = −1.40±0.05 and b = 390±10 K. This fit transferred into the $S_{ice}$-T-space and compared to Koop2000 and the water saturation lines is shown in 6b ( **red** line) with the range of fit uncertainty ( **red** shaded area). The goodness of fit is R²= 0.92. ~~In a next step, this fit is constrained to the well-known homogeneous freezing temperature of pure water droplets, so that when applied to atmospheric models it matches the freezing point of pure water droplets (see Fig. 6c and d, red line). The fit was fixed to a temperature of 235 K and Sice= 1.45, which is the point of water saturated conditions according to the parameterizations of Murphy and Koop (2005). This point corresponds to a nucleation rate coefficient of about Jv= 4.5·10¹⁰ cm⁻³s⁻¹ according to the parameterization of Koop et al.(2000) for the nucleation rate coefficient in dependence on the temperature (T= 235 K) and water activity (aw = 1 for pure water). This nucleation rate coefficient is in the range the AIDA experiments are sensitive to (Jv = 5·10⁸ to 10¹³ cm⁻³s⁻¹) (Möhler et al., 2003). The coefficients of the constrained fit are a = −0.75±0.04 and b = 263±8 K. The goodness of the fit is also R²= 0.92.~~
 **Applying** this **new freezing threshold for cloud formation processes in the atmosphere,**  the homogeneous freezing onset in cirrus formation **would be shifted** to ice saturation ratios of about 2.0 at temperatures around 185 K. A higher homogeneous freezing onset may also contribute to explain high ice saturation ratios reported for low temperatures in the upper troposphere in some field studies (Jensen et al., 2005; Lawson et al., 2008; Krämer et al., 2009; Krämer et al., 2020).

Conclusions:

an empirical fit to the AIDA results with the form $\ln(S_{ice})$ = a+ 1/T·b with fit parameters a =  **−1.40±0.05** and b =  **390±10** K **describes the observed homogeneous freezing onsets of $H_2SO_4/H_2O$ aerosol particles in the chamber. This fit line is based on direct measurements of the freezing onset conditions at simulated cirrus formation conditions and provides an isoline for homogeneous freezing of sulfuric acid aerosol particles for nucleation rate coefficients between $5·10^8$ cm⁻³s⁻¹ and $10^{13}$ cm⁻³s⁻¹.**

9. l. 350-351. It is not necessary to state the same parameters and errors twice in adjacent sentences. Please remove.

We removed a part of this sentence, as suggested (see changes in the manuscript given in the response to comment 8).

10. WAC freezing curves in figures. It is not clear that the freezing curves for constant $J_v$ are correctly determined. In a later paper, Koop and Zobrist[11] altered the homogeneous freezing curve of Koop et al.[1] by an offset in $\Delta a_w$ of 0.008. It appears this is not accounted for in this manuscript.

The homogeneous freezing curves resulting from the fit to the homogeneous freezing data of micrometer-sized droplets in Koop et al. (2000) yielded a $\Delta a_w$ of about 0.305, which is stated to refer to a nucleation rate coefficient of $J_V = 10^{10}$ cm$^{-3}$s$^{-1}$. In the more recent paper of Koop and Zobrist (2009), this $\Delta a_w$ is corrected to about 0.313, as they used the more recent parameterization of Murphy and Koop (2005) to calculate $a_{w,i}$. For calculating the WAC-based freezing curves in our manuscript, we used the parameterization of $J(\Delta a_w)$ given in Koop et al. (2000). As the correction in Koop and Zobrist (2009) is more recent, we corrected our resulting values of $\Delta a_w$ by 0.008, as suggested.

We mentioned the application of the correction in "1. Introduction", as follows:

Also shown are homogeneous freezing onset lines calculated according to Koop2000 for nucleation rate coefficients of $J_V = 10^{13}$ cm$^{-3}$s$^{-1}$ ($\Delta a_w = 0.3\mathbf{2}$, dashed line) and $J_V = 5 \cdot 10^8$ cm$^{-3}$s$^{-1}$ ($\Delta a_w = 0.3\cancel{2}$, dotted line) (Möhler et al., 2003) **and corrected according to the revised version of the homogeneous freezing lines by Koop and Zobrist (2009).**

11. Figure 3. Why doesn't the activated fraction go to 1.0? I expect that homogeneous ice nucleation is so fast that all particles should turn to ice? Is there that much vapor depletion due to the first few ice crystals that form that the authors cannot nucleate all aqueous droplets?

Yes, we assume that is exactly the case. As one can see in Fig. 3 in the manuscript, the relative humidity decreases quickly after the ice onset, although we were still pumping. This shows that the water vapor depletion by the formation and growth of the first ice crystals is indeed very efficient. As the larger particles in the aerosol population have a higher freezing probability due to their larger volumes, they freeze first and the rapid depletion of the supersaturation in the gas phase prevents smaller aerosol particles from freezing.

12. Figure 4 and 6. There is a bit of a bias here (some systematic uncertainty that is not explained?) that the majority of ice saturation data points at temperatures warmer than 210 K are lower than homogeneous freezing estimates. Then, data is mostly higher than homogeneous freezing estimates when temperatures are colder than 210K. Would the authors care to comment on this somewhat systematic uncertainty? In addition, if there is

no theory or physical explanation to back up their measurements (see major comment), their data is more suspect to unknown experimental artifacts or errors.

In the AIDA chamber we have several temperature sensors distributed in the chamber volume. As explained in Section "2.4 Analysis of uncertainties", the deviation between the measurements of the different sensors is usually below 0.1 K. During an expansion, this deviation increases and we have a wider temperature distribution inside the chamber. The freezing onset temperatures given in the manuscript are referring to the mean temperature calculated from the different sensors. Assuming that parts of the volume are colder than others and first ice formation is expected to occur in the coldest parts, the mean temperature might overestimate the actual freezing onset temperature. This uncertainty is considered in the calculation of error bars of the ice onsets, as explained in Section 2.4. Considering the upper limits of these error bars, the ice onsets agree with the homogeneous freezing onsets predicted by the WAC according to Koop et al., 2000.

First of all, we would like to state that we present here experimental results after careful discussion and uncertainty analysis, with direct measurements of the temperature and ice saturation ratio at the point of homogeneous freezing onset for aqueous sulfuric acid aerosol particles. That our data deviates from those derived from the WAC-based parameterization by Koop et al. (2000) is an experimental result, and does not require any physical reasoning. Physical reasoning is important and may be helpful when discussing possible reasons for this disagreement, and some are included in the discussion in section "3. Results and discussion", but unfortunately we do not yet have a definite explanation for the observed deviation.

We therefore included the following statement into Section "3. Results and Discussion" to emphasize that we are not questioning the underlying physical principles of the WAC, but rather its application to the AIDA parameter space at low T:

**Finally, we do not have a solid explanation for the deviation of the $H_2SO_4/H_2O$ homogeneous freezing thresholds observed in AIDA experiments from the WAC-**

**based homogeneous freezing lines. This deviation may be related to uncertainties in the formulation of physicochemical properties at low temperatures, which are required for the conversion between the Koop2000 parameter space ($a_w$-T-space) and the AIDA parameter space ($S_{ice}$-T-space), as described above.**

4. Atmospheric models: Sulfuric acid aerosol particles certainly dominate in the stratosphere, however, there are several studies showing the presence of various other components such as organics and inclusions of aluminum and silicon (Murphy et al., 2014), with even more variety of components present in the upper troposphere. While the presented study focuses only on sulfuric acid aerosol particles. Keeping this in mind as well as the comments stated in 1-3 above, I find the authors' suggestion to use their parameterization in atmospheric models over WAC a bit far fetched. I do not agree with this suggestion and recommend removing this part. In addition, the authors should state this caveat in their abstract, discussion and conclusions. There needs to be more work done on other atmospherically relevant aerosol particles at such low temperatures and high ice supersaturation conditions, to establish whether the deviations from WAC reported here are universal or not.

We understand that a replacement of the WAC and the application in atmospheric models should not be suggested, as it is also discussed in the response to major comment 4 of referee #1. We changed the related sections in the manuscript, as follows:

Abstract:

Based on **the** experimental results **of our direct measurements**, we suggest a new fit line **to formulate**  the onset conditions of homogeneous freezing of sulfuric acid aerosol particles **as an isoline for nucleation rate coefficients between $5 \cdot 10^8$ cm$^{-3}$s$^{-1}$ and $10^{13}$ cm$^{-3}$s$^{-1}$**.  **The potential significant impacts of the higher** homogeneous freezing thresholds, **as directly observed in the AIDA experiments under simulated cirrus formation conditions,**  on the **model** prediction of cirrus cloud occurrence and related cloud radiative effects **are discussed**.

Results and Discussion:

Based on the AIDA results and the discussion above, we  **provide a new fit line** for homogeneous freezing **of H$_2$SO$_4$/H$_2$O aerosol particles directly measured as function of** $S_{ice}$ and T.

 (The section about the constrained fit was completely removed.)

 **Applying** **t**his **new freezing threshold for cloud formation processes in the atmosphere,**  the homogeneous freezing onset in cirrus formation **would be shifted** to ice saturation ratios of about 2.0 at temperatures around 185 K. A higher homogeneous freezing onset may explain the high ice saturation ratios **occasionally** reported for low temperatures in the upper troposphere in some field studies (Jensen et al., 2005; Lawson et al., 2008; Krämer et al., 2009; Krämer et al., 2020). **However, it needs to be considered that the fit line only describes homogeneous freezing of H$_2$SO$_4$/H$_2$O aerosol particles under laboratory conditions. Other aerosol species,**

**which could be relevant for homogeneous freezing processes in the atmosphere, are not taken into account. Application to atmospheric conditions therefore needs to be done with caution.**

Consequently, a precise description of homogeneous freezing processes is crucial to understand cloud radiative effects in the present climate as well as in predictions of climate change.  This may in particular be relevant for cirrus clouds in the cold tropical tropopause layer
(TTL). (This section was moved from Section "4. Conclusions" to Section "3. Results and Discussion")

Conclusions:

A higher homogeneous freezing onset as derived from our experiments may also explain field observations of high clear-sky supersaturation, which should not occur according to the freezing thresholds predicted by Koop2000 (Jensen et al.,2005; Lawson et al., 2008; Krämer et al., 2009; Krämer et al., 2020). **However, the discussed high freezing onsets only consider homogeneous freezing of $H_2SO_4/H_2O$ aerosol particles under laboratory conditions, without involving other atmospherically relevant aerosol species.**  **A** empirical fit to the AIDA results with the form $\ln(S_{ice}) = a + 1/T \cdot b$ with fit parameters a =  **−1.40±0.05** and b =  **390±10** K **describes the observed homogeneous freezing onsets of $H_2SO_4/H_2O$ aerosol particles in the chamber**. **This fit line is based on direct measurements of the freezing onset conditions at simulated cirrus formation conditions and provides an isoline for homogeneous freezing of sulfuric acid aerosol particles for nucleation rate coefficients between $5 \cdot 10^8$ cm$^{-3}$s$^{-1}$ and $10^{13}$ cm$^{-3}$s$^{-1}$.**  **The application of this fit line to atmospheric conditions requires further work on the physical behavior of $H_2SO_4/H_2O$ aerosol particles at low temperatures and on the involvement of other atmospheric aerosol particle types.** Ongoing experiments in the AIDA cloud simulation chamber aim at investigating homogeneous freezing onsets of different solutes and at constraining the descriptions for liquid water saturation pressures to experimental results.

**Minor comments:**

Figure 4: Could the uncertainties in predictions from Koop et al. (2000) be added in this figure for clarity.

In Koop, 2004, the uncertainty range for the freezing line in the $a_w$-T-space is given with 5% uncertainty in the water activity. We included this range of uncertainty in Figs. 4 and 6, as a shaded area around the WAC freezing curves.

L6: WAC not a function of size of the aqueous aerosol particles. Please correct.

We changed the sentence in l. 6 as follows:
The WAC describes the homogeneous nucleation rate coefficients only as a function of the water activity,  which makes this approach well applicable in numerical models.

L255: Please mention/reference the specific E-AIM models used here and for evaluating the data from other studies shown in Fig. 1

We used the Model I. We added the correct references to this line, as suggested on the E-AIM website:
In cases where water activities or relative humidities were not given by the authors, we used the Extended AIM Aerosol Thermodynamics Model (E-AIM)**, Inorganic Model I** on http://www.aim.env.uea.ac.uk/aim/aim.php **(Clegg et al., 1992; Carslaw et al., 1995; Massucci et al., 1999; Wexler and Clegg, 2002; Clegg and Brimblecombe, 2005)** to transfer given $H_2SO_4$ concentrations into water activities, which we assume to be equal to relative humidity.

**Technical comments:**

L8: "...laboratory-based homogeneous..."
Corrected.

L11: "Aqueous sulfuric acid aerosol particles of high purity were generated by..."
Corrected.

L47: "...to be equal to relative humilities."
Corrected.

L80: "...and the potential deviation from..."
Corrected.

L107: "...pump starts..."
Corrected.

L122: a period missing at the end of the sentence
Corrected.

Fig3. Caption: "...AIDA chamber as a function of time since pump start ..."
Corrected.

Fig6. Caption: "The fit shown in panel (a) is constrained..."
This sentence was removed completely due to other referee comments.

L353-54: "...freezing onset may explain the high ice saturation ratios..."
Corrected.

[Figure]

Fig. 1: Water activities of the AIDA results of this study in comparison to the summary of homogeneous freezing experiments shown in Fig. 1 in the manuscript.

This is a good suggestion. In the above Fig. 1, we added the AIDA sulfuric acid data to Fig. 1 of the manuscript. For the calculation of $a_w$, we assumed equilibrium conditions and therefore $a_w = RH_w$. The ice onset ice saturation ratios $S_{ice}$ were then transferred to $RH_w$ using the parameterization of saturation pressures for ice and liquid water given by Murphy and Koop, 2005. As this plot provides a supplementary comparison of our measurements in another parameter space, but is not crucial for the discussion, we decided to put this figure in the appendix and added the following figure description:

**Figure C1. Summary of homogeneous freezing measurements of $H_2SO_4/H_2O$ solutions in the $a_w$-T-space. The homogeneous freezing onsets of sulfuric acid solution samples shown in Fig. 1 are complemented by the AIDA results from this study (red points). The results in the $S_{ice}$-T-space were converted into the $a_w$-T-space by assuming equilibrium conditions $a_w=RH_w$ and by using the parameterizations for the water vapor saturation pressures with respect to ice and supercooled liquid water given by Murphy and Koop (2005). The uncertainties of the calculated $a_w$ values vary between ±0.036 and ±0.076.**

We referred to this figure in the Section "3. Results and discussion", as follows:

With decreasing temperature, the AIDA measurements show an increasing deviation from Koop2000 towards higher ice saturation ratios.
**A comparison of the AIDA results to the homogeneous freezing onsets from previous studies in the $a_w$-T-space shown in Fig. 1 can be found in Fig. C1.**

- There is a lot of uncertainty in the estimation of aw at low T. Slow droplet growth is not a sufficient condition to rule out kinetic limitations since the formation of glasses and hydrates is possible. The E-AIM model does not account for curvature effects which may bias the aw estimate as well.

We added the following sentence to the introduction, where the E-AIM model is mentioned the first time. We also added the Koop, 2004 reference, where this kind of uncertainties are further described:

**It needs to be considered that for the low temperature range, the model predictions are based on extrapolations of physiochemical properties towards low temperatures, which remain uncertain (Koop, 2004).**

We also added this reference to the Section "3. Results and Discussion":

The descriptions for the liquid water saturation pressures are rather uncertain **(Koop, 2004)**, and existing parameterizations deviate from each other (e.g. Buck, 1981; Sonntag, 1994; Tabazadeh et al., 1997; Murphy and Koop, 2005; Nachbar et al., 2019).

We fully agree that the uncertainty of the determination of the water activity at low temperature is an important issue. Our study shows that the conversion of data from different experimental set-ups like the AIDA chamber to compare with the WAC brings uncertainties into the description of homogeneous freezing. For this conversion, calculations based on parameterizations for e.g. water activities of different solutes and the water vapor saturation pressure with respect to supercooled liquid water are needed, which are uncertain especially at low temperatures. More details on this discussion are given in the responses to the major comments 1 and 3 of referee #1.

- The single experiment shown ruling out kinetic limitations is performed at 197 K hence does not target the conditions where the discrepancy with Koop2000 is the largest. It is my feeling that all experiments should have been conducted allowing for equilibration time in the same way as depicted in Fig. 5.

Please note that the experiment shown in Fig. 5 was started at 197 K, but the actual ice onset temperature was observed at a lower temperature of about 193 K with a corresponding ice saturation ratio of about 1.95. Therefore, this experiment was done in the temperature regime, where the observed deviation in onset supersaturation is already significant and we consider it as representative for the other experiments with significant deviation. We are planning more AIDA homogeneous freezing experiments with other solutes and will take up your suggestion to carry out more experiments with variable pumps speeds or rate of pressure change.

- One assumption of Koop2000 is that aw is independent of temperature. Deviation from this behavior would be enough to explain the discrepancy against the results of this work.

Only for some of the experiments shown in Koop et al. (2000), the water activity of the solution aerosols was determined by assuming that $a_w$ is independent of the temperature. In Fig. 1b, Koop et al. (2000) distinguish between freezing points, for which the water activity was derived directly by an ion interaction model at the actual freezing temperature (filled circles), and for which the water activity was estimated by equalling $a_w$ to that measured at the melting temperature (open circles), thus assuming that $a_w$ is independent of temperature between the melting and freezing temperature. All data points are well represented by the shown fit line, independent of the method by which the water activity was determined. We therefore cannot state that the temperature dependence of $a_w$ is generally not taken into account, but, as outlined above, have pointed to the overall uncertainty of the E-AIM in estimating $a_w$ at low T in the revised manuscript version.

- What would happen with the sulfuric acid upon freezing in highly concentrated droplets? This is significant since a fundamental assumption of the equilibrium approach to ice nucleation is that aw=1 in ice. Does the acid remain at the center of the droplet or does it get pushed to the surface (as another review notes, there is some evidence for the later)? If it is incorporated in the ice lattice, then the equilibrium calculations must be corrected accordingly.

Unfortunately, we cannot derive from our measurements what exactly happens to the acid when the ice forms and how it is distributed in or around the ice lattice. There are observations and theories claiming that the remaining highly concentrated sulfuric acid completely covers the ice crystal during the initial growth of the critical ice nucleus, which may decelerate ice crystal growth and could therefore explain the observed high freezing onset supersaturations (e.g. Bogdan et al., 2010). This way of kinetic limitation is also discussed in the manuscript in Section "3. Results and discussion". This discussion was improved and extended with more details, please see our response to major comment 3 of referee#1 for the changes and additions to the manuscript.

- The authors assert that their new correlation should be used instead of the Koop2000 However homogeneous ice nucleation does not admit a "singular" description, and the definition of a freezing threshold is of limited use for atmospheric modeling. Enough data is available in the experiments to calculate the nucleation rate. This would be more meaningful and useful. It would also refute/corroborate the Koop2000 approach of estimating the nucleation rate of solutions at low T based on that of pure water at much higher T. Does the Koop2000 parameterization for J match the measured nucleation rate? This would be a much better test of the Koop2000 hypothesis than the mere calculation of the freezing threshold.

We understand that the application of the new fit line for the sulfuric acid AIDA data in atmospheric models should not be used suggested. We therefore removed any statement suggesting this and added some notes of caution when applying the fit line to atmospheric conditions. For details on the changes in the manuscript, please see the responses to referees #1 (major comment 4) and #2 (major comment 4).

Our experiments at the AIDA chamber are not designed to explicitly calculate nucleation rates with a high accuracy, which would be needed to describe the dependence of water activity on the nucleation rate coefficient, as it is done in Koop et al. (2000). Möhler et al. (2003) give an estimate of homogeneous freezing nucleation rates to which the determination of the freezing onset in AIDA experiment is sensitive. The article also discusses the uncertainties of this estimation, including the estimation of the freezing probability, which is prone to uncertainties of the ice crystal number determination, the estimation of the volume of particles which freeze, and the uncertainty of the time determined for the occurrence of the freezing onset. Therefore, we can only provide an upper and lower limit of nucleation rates to which the AIDA measurements are sensitive to. Based on this estimation in Möhler et al. (2003), we compared our AIDA results to the WAC-lines for the given range of nucleation rates. Due to the mentioned uncertainties, a more accurate determination of nucleation rates is not possible with our experimental set-up.

**Technical comments**

Line 13. Please spell out "several thousand"
Corrected.

Line 31. Organic monolayers also promote freezing, not only solid ice-nucleating particles.
We changed the sentence as follows:
In regions with low concentrations of  ice-nucleating particles, […].

Line 50. This is probably backwards. Higher aw leads to higher J in the Koop2000 model.
Corrected as follows:
Also shown are homogeneous freezing onset lines calculated according to Koop2000 for nucleation rate coefficients of $J_V = 10^{13}$ cm$^{-3}$s$^{-1}$ ($\Delta a_w = 0.3\mathbf{2}$, dashed line) and $J_V = 5 \cdot 10^{8}$ cm$^{-3}$s$^{-1}$ ($\Delta a_w = 0.3\sout{2}$, dotted line) (Möhler et al., 2003).

Line 129. At least for these experiments the FTIR data should give some insight on whether the equilibrium assumption is correct. Does the aw calculated with the E-AIM model using the measured mass concentration match the relative humidity in the cloud chamber?
An accurate analysis of the FTIR data would in principle even be much more insightful, as it could provide the composition of the solution droplets (wt% $H_2SO_4$) upon freezing, i.e. the same quantity as inferred from the freezing experiments by Koop et al. (1998) with deposited sulfuric acid solution droplets. However, due to the lack of accurate infrared optical constants of dilute sulfuric acid solution droplets at such very low temperatures, this analysis is affected by a large uncertainty. It can be seen in the compilation of Fig. 1 in the manuscript that previous studies that have employed infrared spectroscopy to analyze the freezing of $H_2SO_4/H_2O$ aerosol particles (e.g. Bertram et al., 1996 and Clapp et al., 1997) show large deviations from the WAC line, probably due to uncertainties in the determination of the composition of the solution droplets at the freezing onset (Koop et al., 1998). Since 1997, some new low-temperature refractive index data sets for $H_2SO_4/H_2O$ have been published, and we have used them in Wagner et al. (2008) to develop an approach to quantitatively analyze the dilution of the sulfuric acid solution droplets during expansion cooling in the AIDA chamber and determine their composition prior to freezing. However, these new data sets still do not fully capture the range of temperatures and compositions covered by our expansion cooling experiments, so that the analysis remains highly uncertain. Generally, the $a_w$ calculated with the E-AIM model using the droplet composition estimated from the FTIR measurements is 20-30% smaller than the measured relative humidity. But this might again be due to uncertainties in the E-AIM calculations and the saturation water vapor pressure parameterization and cannot be taken as an argument that the equilibrium condition is incorrect.

Line 137. Typo in "aerosol".
Corrected.

Line 165. How important is the Kelvin effect overall? The E-AIM model does not account for it.
For our AIDA experiments, we assume that the Kelvin effect is not contributing to the observed freezing behavior, as the majority of the injected aerosol particles generally were larger than 100 nm in diameter. From the mentioned process modelling studies (Haag et al., 2003 and Wagner et al., 2008), we learned that smaller particles, which are affected by the Kelvin effect, do not maintain the same equilibrium compositions as the larger particles in the aerosol population. Consequently, their freezing behavior should be affected if the water activity is crucial for the homogeneous freezing onsets.

Line 176. Is there an induction time between the onset of ice and the observation a rozen droplet? In other words, how long does it take for ice to propagate inside a droplet?
In general, the diffusive growth of an ice particle is very fast. Of course, it is important to show that this is also the case for low temperature experiments. In the response to major comment 3 of referee #1, we calculated the diffusion length for a water molecule on a time scale of 6s. This diffusion length is significantly larger than the mean particle

diameter in all the experiments. This indicates that the diffusion of water in the aerosol and therefore the growth of the initial ice nucleus is fast. An additional indication that the ice growth is very fast also at low temperatures is the experiment we show in Fig. 5 in the manuscript. Here, we can see that in the minutes of constant but high relative humidity, no ice is detected. This would be the case, if the ice onset already happened at or below this relative humidity, but the detection of the formed ice particles would be delayed due to a significant induction time. In addition, the relative increase in the activated fraction and depolarization (third panel in Fig. 5) is strong and comparable to the increase observed in the experiments at higher temperatures. Regarding our OPCs, their detection range starts at about 0.7 µm. As discussed in Järvinen et al. (2014) the OPCs overestimate the size of aspherical ice particles by a factor of about 2.2. This means that an initial ice embryo needs to grow to an ice particle with a diameter of about 0.32 µm. In our uncertainty analysis, we considered the change in relative humidity and temperature in a specific time interval. Based on the diffusion lengths estimated above, this time interval is expected to cover the time the ice crystal needs to grow in the detection range of the OPC. This is also supported by the observation that the ice onsets determined by the OPCs agree well with the onsets determined by the light scattering measurements of the SIMONE instrument (see Fig. A1 in the manuscript). Therefore, a potential induction time is not expected to change the determined ice onsets in the given uncertainty range.

We added a statement on the potential induction time to Section "2.4 Analysis of uncertainties":

We determined an uncertainty of ±3 s for the onset time. **Based on estimates for the diffusion length of water molecules in the $H_2SO_4/H_2O$ aerosol particles at low temperature, which are discussed in greater detail in Sect. 3, we assume that this uncertainty covers the time the initial ice nucleus needs to grow into the detection range of the OPCs.** The variability of the gas temperature T in this 6s time interval is a first factor contributing to the uncertainty of $T_{ice}$.

Line 203. Please define the freezing probability.
We added the following to the text in the manuscript:
These nucleation rate coefficients were determined for sulfuric acid aerosol particles with diameters between 0.5 and 2 µm with an estimated freezing probability between 0.02 and 0.5. **For the definition of the freezing probability, see Eq. 2 in Möhler et al. (2003).**

Line 204. Given that the whole nucleation "pulse" lasts about 100 s this time delay could be significant.
In the AIDA experiments, we do not have a kind of "nucleation pulse". Temperature is decreasing during the experiment run and the relative humidity is increasing accordingly, until a specific nucleation rate is reached, which causes the observed ice onset in the chamber. This ice onset is characterized by the current temperature and relative humidity inside the chamber. At lower temperatures, the relative humidity changes more slowly over time, so that the uncertainty of ice onset relative humidity is reduced.

Line 252. Please spell out "several thousand" or rephrase it.
Done.

Line 272. This is a key line. Do the authors suggest that such is not the case? If aw-awi is not constant, then the Koop2000 approach is not valid at these conditions. If this is what the authors meant, please spell it out.
No, we do not want to suggest that it is not the case, that the nucleation rate coefficients can be described only in dependence on water activity and temperature. As already stated in our response to referee #1, we do not want to suggest that the WAC and the

underlying assumptions are not correct. Instead, we want to show that the conversion of data from different experimental set-ups like the AIDA chamber to compare with the WAC brings uncertainties into the description of homogeneous freezing. For this conversion, calculations based on parameterizations for e.g. the water vapor saturation pressure with respect to supercooled liquid water are needed, which are uncertain especially at low temperatures.

Line 288. This is a large uncertainty. What is the equivalent in aw space?
The uncertainties of $S_{ice}$ converted into the $a_w$-space gives uncertainties of $a_w$ between ±0.036 and ±0.076. This is also plotted in Fig. C1, which has been added to the manuscript appendix, and mentioned in the figure description.

Line 316-320. I don't see how these results indicate no internal kinetic limitations or that there is no induction time. In fact, in Line 204 it was indicated that it could be as much as 10 s. All that the experiment does is to show that droplet growth is slow (which could be due to glass formation) before nucleation, not that the equilibration time scale is much smaller than the nucleation time scale. Please explain.

We added a more detailed discussion on the kinetic limitations and phase state of the sulfuric acid aerosol particles to Section "3. Results and discussion". Please see response to major comment 3 of referee#1.
We also added a more detailed description of the mentioned experiment shown in Fig. 5 in the manuscript including a new additional panel showing a direct comparison of the relative humidity and the forward scattering intensity measured by SIMONE. We do not agree that the experiment shows that the droplet growth is slow. Rather, the additional panel emphasizes that the particles react instantaneously to the changes in relative humidity. As soon as the pumping speed is reduced and the relative humidity is controlled for a duration of about 300 s to a constant value clearly above the Koop et al. (2000) line, there is no further increase of the forward scattering intensity, which could be ascribed to a delayed water uptake. Rather, the scattering intensities instantly follow the smooth variations in the relative humidity. This observation (which is now much better described in the revised manuscript version), together with the estimates for the diffusion length of the water molecules in the H2SO4/H2O aerosol particles, indicates that kinetic limitations of the particles growth due to water uptake are not expected during the experiment.

Line 321-323. How does the aw calculated with the measured mass concentration compare against the relative humidity? Does the equilibrium assumption hold?
Please see the response to the comment to line 129 above.

Line 340. Freezing thresholds are not very useful for atmospheric modeling. The authors should report nucleation rates instead.
Please see the response to the last general comment about the difficulty of calculating nucleation rates with high accuracy from the AIDA experiments.

Line 345-355. This is confusing. Please just report the recommended correlation.
We decided to keep the unconstrained version of the fit and removed the constrained one from Fig. 6 and the corresponding text. For details on the changes made in the manuscript, please see the response to the minor comment 8 of referee#1.

Line 385. There is the implicit assumption that the nucleation rate is still a function of aw only, which seems to contradict the premise of this work.
At first, we decided to remove the constrained fit, which is described in this line, and to only keep the unconstrained one, as it was suggested to only keep one of the fit

versions. However, in general, the assumption that the nucleation rate coefficient is still a function of $\Delta a_w$ as suggested by Koop et al. (2000), does not contradict the conclusion and discussion of this study. Rather, we show that we observe deviation from the WAC-derived homogeneous freezing lines, but relate this deviation to the conversion from different parameter spaces into each other, and not to an invalidity of the ideas behind the WAC. See also our answer to line 272 above.

Line 392. There is not enough data here to assert this. Koop2000 also parameterizes nucleation rates which is much more useful and completely omitted in this work.
We understand that it should not be suggested to replace the well-established Koop2000 freezing lines. We therefore adjusted this section in the manuscript as follows:

Consequently, a precise description of homogeneous freezing processes is crucial to understand cloud radiative effects in the present climate as well as in predictions of climate change.  This may in particular be relevant for cirrus clouds in the cold tropical tropopause layer
(TTL). (This section was moved from Section "4. Conclusions" to Section "3. Results and discussion")

---

## Author Response (AR2)

**Responses to editor and referee comments:**

**We thank the editor and the referees again for their thoughtful comments and feedback. Please find below our responses and suggestions for the manuscript revision, with the referee comments in black, our answers in green, and suggested changes or additions to the manuscript in blue.**

**Editor report:**

Dear Authors,

I received the comments of the referees on your revised version of the manuscript. Those are listed below with further comments of mine. Please address my and the referees' minor comments before proceeding with the publication of this work.

With kindest regards,

Daniel Knopf

Dear Daniel Knopf,

Thank you for carefully reading again the revised manuscript and for your comments. Please find our answers below.

Best regards, on behalf of all authors,

Julia Schneider

Editor minor comments:
You added uncertainties to the freezing line in Figs. 4 and 6. I just want to clarify that this is not an uncertainty of +-5% but is +-0.05 error in water activity. Hence, it will scale slightly with temperature, I believe. Could you please indicate the meaning of the shading in the respective figure captions. Plotting this uncertainty in Fig. 1 as parallel lines to the Koop et al. (2000)1 freezing curve would be beneficial/fair showing that this includes many of the experimental data points.

We corrected the shown uncertainties in Fig. 4 and 6 in the manuscript so that they now actually represent a relative uncertainty of ± 5% in water activity as given in Koop (2004), and added an explanation on the shaded areas to the figure descriptions. Please see revised figures and changes in the figure descriptions below.

We also added the uncertainty range in water activity of +-5% to the Koop lines shown in Fig. 1 and in Fig. C1 according to Fig. 9 in Koop (2004) and adjusted the figure descriptions in the manuscript (see changes below).

[Figure]

**Figure 4.** Homogeneous freezing onsets of $H_2SO_4/H_2O$ aerosol particles. The freezing onset conditions, $T_{ice}$ and $S_{ice,fr}$, are displayed in comparison with the homogeneous freezing thresholds suggested by the WAC-based predictions by Koop et al. (2000) (dashed and dotted lines) using two different parameterizations for the water saturation pressure with respect to supercooled liquid water from Murphy and Koop (2005) (MK2005, black) and Nachbar et al. (2019) (N2019, blue). **The blue and grey shaded areas are indicative for the uncertainties of the WAC-based predictions (±5% in water activity) as given in Koop (2004).** The used water saturation pressures with respect to supercooled liquid water according to MK2005 (solid black line) and N2019 (solid blue line) are also shown. The colors of the measurement data points represent the different AIDA campaigns in the corresponding years. The oldest campaigns are presented in reddish, whereas the more recent campaigns are shown in yellowish colors.

[Figure]

**Figure 6.** New fit line for homogeneous freezing onsets of $H_2SO_4/H_2O$ aerosol particles. Panel (a): The freezing onsets determined by OPC and SIMONE data (red dots) are shown in an Arrhenius plot and fitted by an ordinary least square (OLS) fit with the form $\ln(S_{ice}) = a + 1/T \cdot b$. The parameters are a = −1.40 ± 0.05 and b = 390 ± 10 K and the goodness of the fit is $R^2$= 0.92. The shaded area is indicative

for the uncertainty of the fit parameters. **Panel (b): The OLS fit shown in panel (a) is transferred into the $S_{ice}$-T-space and compared to Koop2000 and the water saturation lines suggested by Murphy and Koop (2005) and Nachbar et al. (2019). The blue and grey shaded areas are indicative for the uncertainties of Koop2000 (±5% in water activity) as given in Koop (2004).**

[Figure]

**Figure 1.** Review of homogeneous freezing measurements of $H_2SO_4/H_2O$ solutions. The homogeneous freezing onsets of sulfuric acid solution samples reported in different studies are shown and compared. Most of the studies report onset temperatures and weight percentage of $H_2SO_4$ in the solution samples. We used Model I of the E-AIM model (Clegg et al., 1992; Carslaw et al., 1995; Massucci et al., 1999; Wexler and Clegg, 2002; Clegg and Brimblecombe, 2005) to transfer this weight percentage data into water activity, which is assumed to be equal to the relative humidity (assumption of thermodynamic equilibrium). If ice saturation ratios were given, the parameterizations of Murphy and Koop (2005) for the saturation pressures of supercooled liquid water and ice were used to calculate water activities. Additionally, the melting point line according to Murphy and Koop (2005) (solid line) and the homogeneous freezing thresholds for two different nucleation rate coefficients according to Koop et al. (2000) (dashed and dotted lines) are shown. **The grey shaded area is indicative for the uncertainties of these homogeneous freezing thresholds (±5% in water activity) as given in Koop (2004).**

[Figure]

**Figure C1.** Summary of homogeneous freezing measurements of $H_2SO_4/H_2O$ solutions in the $a_w$-T-space. The homogeneous freezing onsets of sulfuric acid solution samples shown in Fig. 1 are complemented by the AIDA results from this study (red points). The results in the $S_{ice}$-T-space were converted into the $a_w$-T-space by assuming equilibrium conditions $a_w$=$RH_w$ and by using the parameterizations for the water vapor saturation pressures with respect to ice and supercooled liquid water given by Murphy and Koop (2005). The uncertainties of the calculated $a_w$ values vary between ±0.036 and ±0.076.

Something not discussed by the referees: The homogenous ice nucleation rate coefficient is likely overpredicted by Koop et al. (2000)1. Knopf and Rigg (2011)2 suggested that Jhom is likely about 2 orders of magnitude lower. This was also observed in a study by the Koop group (Riechers et al. 2013 3). This could delay ice nucleation or decrease the number of ice crystals to be observed. There is no need to go into detail with this but it may be worthwhile to briefly mention the role of Jhom.

We added a statement on the homogeneous nucleation rate coefficient and its impact on the Koop2000 lines to Section "3. Results and discussion" including the suggested references. Changes in the manuscript as follows:

The Koop2000 lines are based on homogeneous freezing experiments of 18 different aqueous solutions with a known and constant composition. The homogeneous freezing temperature was measured as a function of the concentration of the solute. Converting the solute concentration into water activity resulted in a close match of the freezing temperatures for the different solutes. It was therefore suggested to formulate the freezing nucleation rate coefficients of aqueous solution particles only as function of the water activity and the temperature. **More recent studies suggested that the homogeneous nucleation rate coefficients are about 2 orders of magnitude**

**lower than those given by Koop et al. (2000) (Knopf and Rigg, 2011; Riechers et al., 2013). In terms of water activity, the experimentally derived freezing curve by Knopf and Rigg (2011) is about 0.01 lower than that predicted by Koop et al. (2000). This slightly delayed ice nucleation onset, however, does not account for the much larger deviation of the AIDA results to Koop2000.**

Reviewer comments (I have an additional comment to reviewer #3):

Reviewer #1:

The authors have responded well to the comments and I applaud them for their efforts. I recommend publication after a few technical comments listed below are addressed.

1) l. 328-329 revised text. Again, this statement is not entirely accurate because hysteresis is not completely considered. Glassy particles will transition over time to a more well-mixed dilute particle. This could last 1600 s starting at 215 K and cooling to 212 K as reported in my previous comment. Their experiments last < 600 s and maybe I missed this, but how long do the particles wait at the "starting" conditions? During the glass-aqueous solution transition, the particle will still not be in equilibrium as the glassy core is not transitioning. Strictly speaking, just because water uptake occurs does not mean particles are not glassy.

The aerosol particles were exposed to the starting conditions for at least 20 min. We agree that the mere water uptake does not rule out the possibility that the particle core could still be glassy. But we have now better described the time series of the SIMONE data during the considered experiment (see new panel 5b), and explicitly pointed out that there was also no delayed water uptake when the RH was temporarily controlled to a constant value, which contradicts the assumption of a gradual glass-aqueous solution transition.

We therefore propose to delete the corresponding sentence and to combine it with the previous one as follows:

This is also supported by the aerosol particle forward scattering intensity measurements with the SIMONE instrument, which **showed no evidence of delayed water uptake that would be expected if initially glassy particles were gradually transformed to aqueous solution droplets.**

2) References: Many have chemical formulas that require subscripts. Please check.

Corrected.

3) References: Please check Cziczo et al. 2013. "(80-.)".

Corrected.

4) References: Please check the last author in the list for Vortisch et al. 2000.

Corrected.

Reviewer #3:

I thank the authors for addressing my comments and clarify the uncertainties of their approach.

However there is still one thing that is bothering me quite a bit.

I asked the authors to plot their results in the same way as Koop et al. 2000 (that is Tf vs aw). They did it as Figure 1 in their response (AC3). But something seems quite odd. The other studies tend to underestimate aw against Koop2000, however this study overestimates it. Moreover, below 200 K all their aw values are above 1. Honestly I can't think of a scenario where a highly concentrated solution of sulfuric acid would have aw>1. I would like the authors to please clarify this, just to make sure their analysis is fundamentally consistent.

Editor comment: Reviewer #3 refers to Figs. 1 and C1. To put it in other words and to avoid confusion that persists in the community and literature: Obviously, RH is a parameter that describes the gas phase and aw the condensed phase. If gas and condensed phase are in equilibrium, RH=aw. Although in experiments one can achieve RH > 100%, aw can never be larger than 1. aw=1 indicates the presence of pure water, not bound to other ions etc. Even if the gas phase rises above 100% RH and more water condenses, aw stays equal to 1.

We agree with referee #3 and the editor that a water activity > 1 cannot occur. We have therefore adjusted Fig. 1 and Fig. C1 by setting those data points, at which the measured $S_{ice}$ onset corresponded to a relative humidity >100%, to a water activity equal to one. As noted above, we also included the uncertainty range of the Koop et al., 2000 lines. For changes in the manuscript, please see the revised Fig. 1 and C1 and descriptions above. Please note again that the water activities for the new AIDA experiments plotted in Fig. C1 only result from the conversion of our directly measured nucleation onsets, given in terms of $S_{ice}(T)$. We also do not expect that the water activities of the aqueous sulfuric acid solution droplets (which have a concentration of about 25 wt% $H_2SO_4$ when freezing at 190 K) have values close or equal to one, as suggested by the AIDA data for temperatures below 200 K. Rather, we discuss in our article that the uncertainty with respect to the parameterization for the saturation water vapor pressure over supercooled water at T <= 200 K could be a reason for the apparently "inconsistent" picture when our $S_{ice}(T)$ freezing data are converted to the $a_w(T)$ space in Fig. C1 (see also the point below). For that reason, we provided our parameterization directly in the measured $S_{ice}$ – T space.

The other point the reviewer hints to when asking for "fundamentally consistent" is: If you plot a data point indicating pure water (aw=1) below ~235 K (Fig. C1), you imply the existence of water/ice in the physical not allowed region, also called "no man's land" (see your Fig. D1). With the experimental conditions applied here, you cannot produce ice under these thermodynamic conditions. This in turn implies that the water saturation parameterizations (both Nachbar et al. and Murphy and Koop) would need to make a huge jump to higher RHice values below 200 K to make this somehow consistent with known physics.

We agree that it is difficult to show and discuss freezing experiments in the "no man's land" region. However, we think a discussion of measurements at these conditions is important, as these occur in the upper troposphere/lower stratosphere. We agree that the water saturation parameterizations are a critical aspect in this discussion, as we also state in Section "3. Results and discussion". Nachbar et al., 2019 do not give a parameterization for water saturation pressure below 200 K and the Murphy and Koop, 2005 parameterization is based on interpolations through the "no man's land" region. This shows how uncertain the parameterizations at these conditions are. Nachbar et al. (2019) actually suggested that there could be a discontinuity in the water vapor pressure in the "no man's land" region between 230 and 200 K, because their findings indicate that amorphous solid water (ASW) and supercooled liquid water (SLW) are distinct phases of water, potentially involving a

phase transition somewhere in the "no man's land". As shown in their Fig. 2, there is a large gap between the water vapor pressures at 200 K (the warmest known temperature of existence of ASW) and 236 K (the coldest known temperature of existence of SLW). They proposed studies of the condensation of gas phase water on hydrophobic surfaces to detect a potential phase transition from SLW to ASW when lowering the temperature, and as stated in the article we plan to focus on this aspect in future experiments.

Clearly, there are processes in the aqueous sulfuric acid system going on below 200 K that we do not completely understand. The only other point that has not been mentioned is that the aqueous sulfuric acid aerosol may transform into a sulfuric acid monohydrate which happens slightly above 190 K (considering the uncertainty in RH, there might be a range where this could happen). See, e.g., discussion of Fig. 6 in Koop et al. (2011)4. Then some water would be bound in the crystal with a remaining aqueous solution outside the crystal yielding a much decreased aw value. This could potentially reduce the disagreements observed below 200 K.

The potential crystallization of sulfuric acid hydrates (e.g. the monohydrate, SAM, or the tetrahydrate SAT) and their impact on the ice nucleation experiments at temperatures below 200 K is indeed an interesting question. Our measurement data do not indicate that such crystallization has occurred in a significant (> 5-10%) number fraction of the injected aerosol particles. Firstly, sulfuric acid hydrate infrared spectra have features distinctly different from aqueous $H_2SO_4/H_2O$ solution droplets (Nash et al., 2001), but such signatures were not detected in our FTIR measurements. Secondly, the presence of crystalline or partly crystalline particles would have resulted in an increase of the backscattering linear depolarization ratio compared to the background value measured for purely aqueous solution droplets, which was also not detected in the SIMONE records. Thirdly, as shown specifically for SAT (Fortin et al., 2003), sulfuric acid hydrates could act as ice nucleating particles and induce heterogeneous ice formation at lower $S_{ice}$ values compared to homogeneous freezing conditions, which was also not detected in the experiments presented in this article. We therefore strongly suppose that our experiments were unaffected by the formation of hydrates, and we will add a short statement on this issue in Section "3. Results and discussion" of the article:

[...] **In addition, the depolarization measurements with the SIMONE instrument and the signatures of the recorded FTIR spectra show no indication for the formation of sulfuric acid hydrates (Nash et al., 2001), which could occur under the experimental conditions in the AIDA chamber (Koop et al., 1997). An impact of hydrate formation on the observed ice onsets, as shown e.g. in the case of sulfuric acid tetrahydrate (Fortin et al., 2003), is therefore not expected.**

References
1. Koop, T.; Luo, B. P.; Tsias, A.; Peter, T., Water activity as the determinant for homogeneous ice nucleation in aqueous solutions. Nature 2000, 406 (6796), 611-614.
2. Knopf, D. A.; Rigg, Y. J., Homogeneous Ice Nucleation From Aqueous Inorganic/Organic Particles Representative of Biomass Burning: Water Activity, Freezing Temperatures, Nucleation Rates. J. Phys. Chem. A 2011, 115 (5), 762-773.
3. Riechers, B.; Wittbracht, F.; Hutten, A.; Koop, T., The homogeneous ice nucleation rate of water droplets produced in a microfluidic device and the role of temperature uncertainty. Phys. Chem. Chem. Phys. 2013, 15 (16), 5873-5887.
4. Koop, T.; Bookhold, J.; Shiraiwa, M.; Pöschl, U., Glass transition and phase state of organic

compounds: dependency on molecular properties and implications for secondary organic aerosols in the atmosphere. Phys. Chem. Chem. Phys. 2011, 13 (43), 19238-19255.

Additional references in author's response:

Carslaw, K. S., Clegg, S. L., and Brimblecombe, P.: A thermodynamic model of the system HCl-HNO3-H2SO4-H2O, including solubilities of HBr, from <200 to 328 K, J. Phys. Chem., 99, 11 557–11 574, https://doi.org/10.1021/j100029a039, 1995.

Clegg, S. L. and Brimblecombe, P.: Comment on the "Thermodynamic dissociation constant of the bisulfate ion from Raman and ion interaction modeling studies of aqueous sulfuric acid at low temperatures", The Journal of Physical Chemistry A, 109, 2703–2706,https://doi.org/10.1021/jp0401170, 2005.

Clegg, S. L., Pitzer, K. S., and Brimblecombe, P.: Thermodynamics of multicomponent, miscible, ionic solutions. Mixtures including un-symmetrical electrolytes, The Journal of Physical Chemistry, 96, 9470–9479, https://doi.org/10.1021/j100202a074, 1992.

Fortin, T. J., Drdla, K., Iraci, L. T., and Tolbert, M. A.: Ice condensation on sulfuric acid tetrahydrate: Implications for polar stratospheric ice clouds, Atmos. Chem. Phys., 3, 987-997, 2003.

Koop, T., Carslaw, K. S., and Peter, T.: Thermodynamic stability and phase transitions of PSC particles, Geophys. Res. Lett., 24, 2199-2202, https://doi.org/10.1029/97GL02148, 1997.

Koop,T.: Homogeneous ice nucleation in water and aqueous solutions, Z. Phys. Chem., 218,1231–1258, https://doi.org/10.1524/zpch.218.11.1231.50812, 2004.

Massucci, M., Clegg, S. L., and Brimblecombe, P.: Equilibrium partial pressures, thermodynamic properties of aqueous and solid phases, and Cl2 production from aqueous HCl and HNO3 and their mixtures, The Journal of Physical Chemistry A, 103, 4209–4226,https://doi.org/10.1021/jp9847179, 1999.

Murphy, D. M. and Koop, T.: Review of the vapour pressures of ice and supercooled water for atmospheric applications, Q. J. R. Meteorol. Soc., 131, 1539–1565, https://doi.org/10.1256/qj.04.94, 2005.

Nachbar, M., Duft, D., and Leisner, T.: The vapor pressure of liquid and solid water phases at conditions relevant to the atmosphere, J. Chem. Phys., 151, 064 504, https://doi.org/10.1063/1.5100364, 2019.

Nash, K. L., Sully, K. J., and Horn, A. B.: Observations on the Interpretation and Analysis of Sulfuric Acid Hydrate Infrared Spectra, The Journal of Physical Chemistry A, 105, 9422-9426, 10.1021/jp0114541, 2001.

Wexler, A. S. and Clegg, S. L.: Atmospheric aerosol models for systems including the ions H+, NH+4, Na+, SO2−4, NO−3, Cl−, Br−, and H2O, J. Geophys. Res. Atmos., 107, https://doi.org/10.1029/2001JD000451, 2002.